# Causal Representation Learning for Instantaneous and Temporal Effects in Interactive Systems

**Phillip Lippe**
QUVA Lab
University of Amsterdam
p.lippe@uva.nl

**Sara Magliacane**
MIT-IBM Watson AI Lab
University of Amsterdam

**Sindy Löwe**
UvA-Bosch Delta Lab
University of Amsterdam

**Yuki M. Asano**
QUVA Lab
University of Amsterdam

**Taco Cohen**
Qualcomm AI Research[*]
Amsterdam

**Efstratios Gavves**
QUVA Lab
University of Amsterdam

## ABSTRACT

Causal representation learning is the task of identifying the underlying causal variables and their relations from high-dimensional observations, such as images. Recent work has shown that one can reconstruct the causal variables from temporal sequences of observations under the assumption that there are no instantaneous causal relations between them. In practical applications, however, our measurement or frame rate might be slower than many of the causal effects. This effectively creates "instantaneous" effects and invalidates previous identifiability results. To address this issue, we propose iCITRIS, a causal representation learning method that allows for instantaneous effects in intervened temporal sequences when intervention targets can be observed, e.g., as actions of an agent. iCITRIS identifies the potentially multidimensional causal variables from temporal observations, while simultaneously using a differentiable causal discovery method to learn their causal graph. In experiments on three datasets of interactive systems, iCITRIS accurately identifies the causal variables and their causal graph.

## 1 INTRODUCTION

Recently, there has been a growing interest in *causal representation learning* (Schölkopf et al., 2021), which aims at learning representations of causal variables in an underlying system from high-dimensional observations like images. Several works have considered identifying causal variables from time series data, assuming that the variables are independent of each other conditioned on the previous time step (Gresele et al., 2021; Khemakhem et al., 2020a; Lachapelle et al., 2022a;b; Lippe et al., 2022b; Yao et al., 2022a;b). This assumes that within each discrete, measured time step, intervening on one causal variable does not affect any other variable instantaneously. However, in real-world systems, this assumption is often violated, as there might be causal effects that act faster than the measurement or frame rate (Faes et al., 2010; Hyvärinen et al., 2008; Moneta et al., 2006; Nuzzi et al., 2021). Consider the example of a light switch and a light bulb. When flipping the switch, there is an almost immediate effect on the light by turning it on or off, changing the appearance of the whole room instantaneously. In this case, an intervention on a variable (*e.g.*, the switch) also affects other variables (*e.g.*, the bulb) in the same time step, violating the assumption that each variable is independent of the others in the same time step, conditioned on the previous time step. In biology, some protein-protein interactions also occur nearly-instantaneously (Acuner Ozbabacan et al., 2011).

To overcome this limitation, we consider the task of identifying causal variables and their causal graphs from temporal sequences, even in case of instantaneous cause-effect relations. This task contains two main challenges: identifying the causal variables from observations, and learning the causal relations between those variables. We show that, as opposed to temporal sequences without instantaneous

---

[*]Qualcomm AI Research is an initiative of Qualcomm Technologies, Inc.

effects, neither of these two tasks can be completed without the other: without knowing the variables, we cannot identify the graph; but without knowing the graph, we cannot identify the causal variables, since they are not conditionally independent. In particular, in contrast to causal relations across time steps, the orientations of instantaneous edges are not determined by the temporal ordering, hence requiring to jointly solve the tasks of causal representation learning and causal discovery.

As a starting point, we consider the setting of CITRIS (Causal Identifiability from Temporal Intervened Sequences; Lippe et al. (2022b)). In CITRIS, potentially multidimensional causal variables interact over time, and interventions with known targets may have been performed. While in that work all causal relations were assumed to be *temporal*, *i.e.*, from variables in one time step to variables in the next time step, we generalize this setting to include instantaneous causal effects. In particular, we show that in general, causal variables are not identifiable if we do not have access to *partially-perfect interventions*, *i.e.*, interventions that remove the instantaneous parents. If such interventions are available, we prove that we can identify the minimal causal variables (Lippe et al., 2022b), *i.e.*, the parts of the causal variables that are affected by the interventions, and their temporal and instantaneous causal graph. Our results *generalize* the identifiability results of Lippe et al. (2022b), since if there are no instantaneous causal relations, any intervention is partially-perfect by definition. As a practical implementation, we propose *instantaneous CITRIS* (iCITRIS). iCITRIS maps high-dimensional observations, *e.g.*, images, to a lower-dimensional latent space on which it learns an instantaneous causal graph by integrating differentiable causal discovery methods into its prior (Lippe et al., 2022a; Zheng et al., 2018). In experiments on three different video datasets, iCITRIS accurately identifies the causal variables as well as their instantaneous and temporal causal graph. Our contributions are:

- We show that causal variables in temporal sequences with instantaneous effects are not identifiable without interventions that remove instantaneous parents.
- We prove that when having access to such interventions with known targets, the minimal causal variables can be identified along with their causal graph under mild assumptions.
- We propose iCITRIS, a causal representation learning method that identifies minimal causal variables and their causal graph even in the case of instantaneous causal effects.

**Related Work** We provide an extended discussion on related work in Appendix C. Early works on causal representation learning focused on identifying independent factors of variations (Klindt et al., 2021; Kumar et al., 2018; Locatello et al., 2019; 2020b; Träuble et al., 2021), in settings similar to Independent Component Analysis (ICA) (Comon, 1994; Hyvärinen et al., 2001; 2019). In particular, Lachapelle et al. (2022a;b); Yao et al. (2022a;b) discuss the identifiability of causal variables from temporal sequences. Yet, in all of these ICA-based setups, causal variables are required to be conditionally independent. For causally-dependent variables, Yang et al. (2021) learn causal variables from labeled images in a supervised manner. Ahuja et al. (2022); Brehmer et al. (2022) identify causal variables with unknown causal relations from pairs of observations that only differ in a subset of causal factors influenced by an intervention, *i.e.*, having counterfactual observations. As discussed by Pearl (2009), however, knowing counterfactuals is not realistic in most scenarios. Instead, CITRIS (Lippe et al., 2022b) focuses on temporal sequences, in which also the variables that are not intervened upon can still continue evolving over time. On the other hand, in this setting the intervention targets need to be known. Moreover, within a time step, the causal variables are assumed to be independent conditioned on the variables of the previous time step, hence not allowing for instantaneous effects. To the best of our knowledge, iCITRIS is the first method to identify causal variables and their causal graph from temporal, intervened sequences even for potentially instantaneous causal effects, without requiring counterfactuals or data labeled with the true causal variables.

## 2 RELEVANT BACKGROUND AND DEFINITIONS

In this work, we start from the setting of Temporal Intervened Sequences (TRIS) (Lippe et al., 2022b). For clarity, we provide a brief overview of TRIS and discuss previous identifiability results, before extending and generalizing the theory to instantaneous effects.

### 2.1 TEMPORAL INTERVENED SEQUENCES

Temporal intervened sequences (TRIS) (Lippe et al., 2022b) are a latent temporal causal process $\mathcal{S}$ with $K$ *causal variables* $(C_1^t, C_2^t, ..., C_K^t)_{t=1}^T$ (*e.g.*, the light switch and bulb), representing a dynamic

Bayesian network (DBN) (Dean et al., 1989; Murphy, 2002). Each causal variable $C_i$ is instantiated at each time step $t$, denoted by $C_i^t$, and its causal parents $\mathrm{pa}(C_i^t)$ are a subset of the variables at time $t-1$.

To represent interventions, the causal graph is augmented with binary intervention variables $I^t \in \{0,1\}^K$, where $I_i^t = 1$ refers to the causal variable $C_i^t$ having been intervened at time step $t$. This setting can model *soft* interventions (Eberhardt, 2007), *i.e.*, interventions that change the conditional distribution $p(C_i^t|\mathrm{pa}(C_i^t), I_i^t = 1) \neq p(C_i^t|\mathrm{pa}(C_i^t), I_i^t = 0)$ (*e.g.*, flipping the light switch). This trivially includes also *perfect* interventions, $\mathrm{do}(C_i = c_i)$ (Pearl, 2009), which cause the target variable to be independent of its parents. To allow for arbitrary sets of interventions, the intervention variables are considered to be confounded by an unobserved variable $R^t$. While the intervention variables $I^t$ are assumed to be observed, the actual values of the intervened variables, *e.g.*, the state of the light switch, are not. The graph and its parameters are assumed to be time-invariant (*i.e.*, repeat across time steps), causally sufficient (*i.e.*, no latent confounders besides the variables mentioned before), and faithful (*i.e.*, no additional independences w.r.t. the ones encoded in the graph).

In this setting, causal variables can be scalar or span over multiple dimensions, *i.e.*, $C_i \in \mathbb{D}_i^{M_i}$ with the dimensionality $M_i \geq 1$ and $\mathbb{D}_i$ being the domain. For example, a 3d position can be represented as $C_i \in \mathbb{R}^3$. The causal variable space is defined as $\mathcal{C} = \mathbb{D}_1^{M_1} \times \mathbb{D}_2^{M_2} \times ... \times \mathbb{D}_K^{M_K}$.

Instead of observing the causal variables directly, we measure a high-dimensional observation $X^t$, *e.g.*, an image, representing a noisy, entangled view of all causal variables $C^t = (C_1^t, C_2^t, ..., C_K^t)$ at time step $t$. The observation function is defined as $h(C_1^t, C_2^t, ..., C_K^t, E^t) = X^t$, where $E^t \in \mathcal{E}$ is any noise on the observation $X^t$ that is independent of $C^t$ (*e.g.*, pixel or color shifts), and $h : \mathcal{C} \times \mathcal{E} \to \mathcal{X}$ is a function from the causal variable space $\mathcal{C}$ and the space of the observation noise $\mathcal{E} = \mathbb{R}^L$ to the observation space $\mathcal{X} \subseteq \mathbb{R}^N$. To allow unique identification of causal variables from observations, the observation function $h$ is assumed to be bijective, *i.e.*, there exist a unique inverse of $h$ for all $X \in \mathcal{X}$.

## 2.2 MINIMAL CAUSAL VARIABLES AND IDENTIFIABILITY CLASS

Multidimensional causal variables are not always fully identifiable in TRIS, when interventions only affect a subset of the variables' dimensions (Lippe et al., 2022b), *e.g.*, only $x$ in a 3d position $[x, y, z]$. To account for these interventions, each causal variable $C_i^t$ can be split into an *intervention-dependent* part $s_i^{\mathrm{var}}(C_i^t)$, *e.g.*, $[x]$, and an *intervention-independent* part $s_i^{\mathrm{inv}}(C_i^t)$, *e.g.*, $[y, z]$, where $s_i(C_i^t) = (s_i^{\mathrm{var}}(C_i^t), s_i^{\mathrm{inv}}(C_i^t))$ is an invertible function. Under this split, the distribution of $C_i^t$ becomes:

$$p\left(s_i(C_i^t)|\mathrm{pa}(C_i^t), I_i^t\right) = p\left(s_i^{\mathrm{var}}(C_i^t)|\mathrm{pa}(C_i^t), I_i^t\right) \cdot p\left(s_i^{\mathrm{inv}}(C_i^t)|\mathrm{pa}(C_i^t)\right) \tag{1}$$

With this setup, Lippe et al. (2022b) define a *minimal causal variable* as follows:

**Definition 2.1.** *The **minimal causal variable** of a causal variable $C_i^t$ w.r.t. its intervention variable $I_i^t$ is the intervention-dependent part $s_i^{\mathrm{var}}(C_i^t)$ of the split $s_i(C_i^t) = (s_i^{\mathrm{var}}(C_i^t), s_i^{\mathrm{inv}}(C_i^t))$, such that the split maximizes the information content $H(s_i^{\mathrm{inv}}(C_i^t)|\mathrm{pa}(C_i^t))$ in terms of the limiting density of discrete points (LDDP) (Jaynes, 1957; 1968).*

To identify the minimal causal variables from data triplets $\{x^t, x^{t+1}, I^{t+1}\}$, CITRIS approximates the observation function $h$ by learning an invertible map $g_\theta : \mathcal{X} \to \mathcal{Z}$, with $\mathcal{Z} \in \mathbb{R}^M$ being a latent space with $M$ dimensions. For this latent space, CITRIS learns an assignment function $\psi : [\![1..M]\!] \to [\![0..K]\!]$, mapping each dimension of $\mathcal{Z}$ to a causal variable $C_1, ..., C_K$. The index 0 is used for the observation noise or intervention-independent variables. We denote the set of latents assigned to the causal variable $C_i$ with $z_{\psi_i} = \{z_j | j \in [\![1..M]\!], \psi(j) = i\}$. On this latent space, CITRIS models a prior $p_{\phi,\psi}(z^{t+1}|z^t, I^{t+1})$ where each group of latent variables, $z_{\psi_i}^{t+1}$, is conditioned on the previous time step $z^t$ and its intervention variable $I_i^{t+1}$, but independent among each other within the same time step. The causal graph can be found by pruning the temporal dependencies in this prior.

Under this model, Lippe et al. (2022b) consider a causal system $\mathcal{S}$ to be identified by a model $\mathcal{M}$ if its minimal causal variables are identified up to an invertible transformation, and $\mathcal{M}$ recovers the true causal graph in $\mathcal{S}$. CITRIS is shown to identify $\mathcal{S}$ if it maximizes the information content of $z_{\psi_0}$, under the constraint of maximizing the likelihood of data points $\{X^t, X^{t+1}, I^{t+1}\}$, and no intervention variable $I_i^t$ is a deterministic function of any other intervention variable. However, if causal effects occur faster than the observation rate, an intervention influences also other variables in the same time step in addition to its target, leading CITRIS to potentially identify incorrect variables. In this paper, we generalize this identifiability result to systems where instantaneous causal relations may exist.

# 3 IDENTIFYING CAUSAL VARIABLES WITH INSTANTANEOUS EFFECTS

In this section, we generalize Temporal Intervened Sequences (TRIS) to settings where causal relations can be potentially instantaneous. First, we discuss the challenges arising from instantaneous effects, and then present solutions to overcome these challenges. Finally, we derive our identifiability results.

## 3.1 iTRIS AND CHALLENGES OF INSTANTANEOUS EFFECTS

We extend TRIS to a setting we call instantaneous Temporal Intervened Sequences (iTRIS) which allows for instantaneous causal effects. In iTRIS, causal variables within the same time step can cause each other, as long as the graph remains acyclic. This means that, for example, $C_i^{t+1}$ can cause $C_j^{t+1}$ for $i \neq j$, as long as there is no directed path $C_i^{t+1} \to \cdots \to C_i^{t+1}$. Figure 1 summarizes this setting.

While the addition of instantaneous effects may seem like a small change, it violates the key assumption of most previous works (Khemakhem et al., 2020a; Lachapelle et al., 2022a;b; Lippe et al., 2022b; Yao et al., 2022a;b), namely that causal variables within a time step are independent conditioned on some external variable. As a consequence, we have to differentiate between causal models in a much larger function space than before, making identifiability a considerably harder task.

To formalize this intuition, consider the following example. Assume we have two latent causal variables $C_1$ and $C_2$, and, for simplicity, no temporal relations. The causal variables $C_1$ and $C_2$ do not cause each other, and we have an arbitrary observation function $h(C_1, C_2) = X$ and distributions $p_1(C_1), p_2(C_2)$. In this example, if our method allows for instantaneous effects, we cannot identify the causal variables or their graph from $p_x(X)$ alone, since there are multiple representations that model $p_x(X)$ equally well. For instance, the representation $\hat{C}_1 = C_1, \hat{C}_2 = C_1 + C_2$ with the causal graph $\hat{C}_1 \to \hat{C}_2$ model the same observation distribution since $\hat{p}_2(\hat{C}_2|\hat{C}_1) = \hat{p}_2(C_1 + C_2|C_1) = p_2(C_2)$ and hence $\hat{p}(\hat{C}_2|\hat{C}_1)\hat{p}(\hat{C}_1) = p(C_1)p(C_2) = p_x(X)$. This happens even under soft interventions, because the causal graph can remain unchanged. However, if we have interventions on $C_i$ that remove its instantaneous parents, then the learned representation of $C_i$ must be independent of any instantaneous effects. This eliminates $\hat{C}_1, \hat{C}_2$, since $\hat{C}_2 \not\perp \hat{C}_1 | I_2 = 1$. We refer to these interventions as *partially-perfect*:

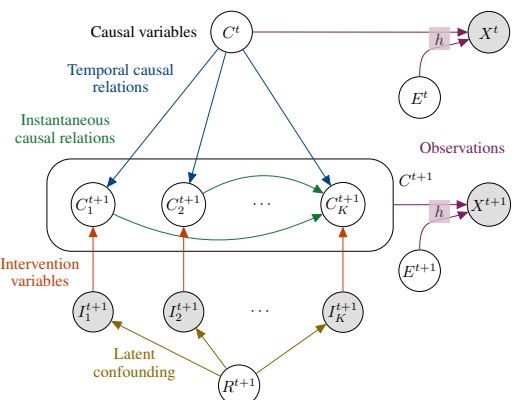

Figure 1: An example causal graph in iTRIS. A latent causal variable $C_i^{t+1}$ can have as potential parents a subset of the causal variables at the previous time step $C^t = (C_1^t, \ldots, C_K^t)$, instantaneous parents $C_j^{t+1}, i \neq j$, and its intervention variable $I_i^{t+1}$. All causal variables $C^{t+1}$ and observation noise $E^{t+1}$ cause the observation $h(C^{t+1}, E^{t+1}) = X^{t+1}$. $R^{t+1}$ is a latent confounder allowing for dependencies between intervention variables.

**Definition 3.1.** *A **partially-perfect intervention** on a causal variable $C_i$ is a soft intervention that removes all parents in the instantaneous graph: $p(C_i^t|pa(C_i^t), I_i^t = 1) = p_I(C_i^t|pa^{temp}(C_i^t))$ where $pa^{temp}(C_i^t) = \{C_j^{t-1}|j \in [\![1..K]\!], C_j^{t-1} \in pa(C_i^t)\}$ and $p_I$ is the post-interventional distribution.*

As an example, consider an intervention that sets $C_i^t = C_i^{t-1} + \epsilon$, where $\epsilon$ is noise. While this intervention breaks instantaneous relations, the target still depends on the previous time step, making it partially-perfect. Using the partially-perfect interventions, we can prove the following:

**Lemma 3.2.** *In iTRIS, a causal variable $C_i$ cannot be identified up to an invertible transformation $T_i$ that is independent of all other causal variables, if $C_i$ can have instantaneous parents and no partially-perfect interventions on $C_i$ are provided.*

We provide the proof for this lemma and an example with temporal relations in Appendix D.2.1. In the following, we assume that all interventions in iTRIS are partially-perfect. This subsumes the soft intervention setup in TRIS by definition, since the instantaneous graph is empty in this case.

## 3.2 IDENTIFYING THE MINIMAL CAUSAL VARIABLES IN iTRIS

Using partially-perfect interventions, we introduce our identifiability results in iTRIS. For simplicity of the theoretical analysis, we follow Lippe et al. (2022b) and consider a continuous domain for causal variables, *i.e.*, $\mathbb{D} = \mathbb{R}$, and that distributions have full support. In Section 5, we empirically extend the results to variables with categorical and circular domains, and distributions with limited support. Further, we assume that the temporal dependencies and interventions break all symmetries in the variables' distributions such that distributional implies functional independence. For a Gaussian, this entails that the mean cannot be a constant (see Appendix D.2 for details on the assumptions).

Similar to CITRIS, we learn an invertible mapping, $g_\theta : \mathcal{X} \to \mathcal{Z}$, and an assignment function $\psi : [\![1..M]\!] \to [\![0..K]\!]$ to align latent to causal variables. Differently, however, we also learn a directed, acyclic graph $G$ on the $K + 1$ latent variable groups $z_{\psi_0}, ..., z_{\psi_K}$ to model the instantaneous causal relations. The graph $G$ induces a parent structure denoted by $z_{\psi_i^{pa}} = \{z_j | j \in [\![1..M]\!], \psi(j) \in \mathrm{pa}_G(i)\}$ where $\mathrm{pa}_G(0) = \emptyset$, *i.e.*, the variables in $z_{\psi_0}$ have no instantaneous parents. Meanwhile, the temporal causal graph between $C^t$ and $C^{t+1}$ is implicitly learned by conditioning $z^{t+1}$ on all latents of the previous time step, $z^t$, and can be pruned after training. This results in the following prior:

$$p_{\phi,\psi,G}\left(z^{t+1}|z^t, I^{t+1}\right) = p_\phi\left(z_{\psi_0}^{t+1}|z^t\right) \cdot \prod_{i=1}^{K} p_\phi\left(z_{\psi_i}^{t+1}|z^t, z_{\psi_i^{pa}}^{t+1}, I_i^{t+1}\right). \tag{2}$$

With this setup, we can now formally define our identifiability class that contains the one in CITRIS:

**Definition 3.3.** *A model $\mathcal{M} = \langle \theta, \phi, \psi, G \rangle$ **identifies** a causal system $\mathcal{S} = \langle C, E, h \rangle$ iff for each causal variable $C_i, i \in [\![1..K]\!]$, the following two conditions hold:*
*(1) each minimal causal variable is identified up to an invertible transformation $T_i$, i.e., for all observations $x \in \mathcal{X} : s_i^{var}(c_i) = T_i(z_{\psi_i})$, where $c_i = [h^{-1}(x)]_i$ is the value of the true causal variable and $z_{\psi_i} = [g_\theta(x)]_{\psi_i}$ is the value of the estimated minimal causal variable, and*
*(2) the estimated parents $\mathrm{pa}(z_{\psi_i}^t)$ are the same as the true parents of the minimal causal variable $s_i^{var}(C_i^t)$, i.e., the estimated parent set contains $z_{\psi_j}^\tau, j \in [\![1..K]\!], \tau \in \{t-1, t\}$ iff $s_j^{var}(C_j^\tau)$ is a parent of $s_i^{var}(C_i^t)$, and it contains $z_{\psi_0}^\tau$ iff there exist $l \in [\![1..K]\!]$ for which $s_l^{inv}(C_l^\tau)$ is a parent of $s_i^{var}(C_i^t)$.*

For the light switch example, this means that the latent variable $z_{\psi_1}$ must model the switch's state, $z_{\psi_2}$ the bulb's state, and the instantaneous graph is $z_{\psi_1} \to z_{\psi_2}$. Compared to ICA-based results, this identifiability class explicitly aligns the latent variables with the causal variables and, thus, we do not rely on a permutation equivalence class. To identify $\mathcal{S}$ from observations, we consider a dataset of triplets $\{x^t, x^{t+1}, I^{t+1}\}$ with observations $x^t, x^{t+1} \in \mathcal{X}$ and intervention variables $I^{t+1}$. This dataset could be created interactively or, for example, recorded by an expert. With this, the objective becomes:

$$p_{\phi,\psi,\theta,G}\left(x^{t+1}|x^t, I^{t+1}\right) = \left|\det J_{g_\theta}(x^{t+1})\right| \cdot p_{\phi,\psi,G}\left(z^{t+1}|z^t, I^{t+1}\right) \tag{3}$$

where the Jacobian of $g_\theta$, $\left|\det J_{g_\theta}(x^{t+1})\right|$, is introduced due to the change of variables of $x$ to $z$. If $\dim(\mathcal{X}) > \dim(\mathcal{C} \times \mathcal{E})$, we consider that $g_\theta$ contains an arbitrary, fixed map from $\mathcal{X}$ to the lower dimensionality. Under the assumption that $g_\theta, p_\phi$ are universal function approximators and the dataset is unlimited in size, we derive the following identifiability result:

**Theorem 3.4.** *In iTRIS, a model $\mathcal{M}^* = \langle \theta^*, \phi^*, \psi^*, G^* \rangle$ identifies a causal system $\mathcal{S} = \langle C, E, h \rangle$ (Definition 3.3) if $\mathcal{M}^*$, under the constraint of maximizing the likelihood $p_{\phi,\theta,G}(X^{t+1}|X^t, I^{t+1})$:*
*(1) maximizes the information content $H(z_{\psi_0}^{t+1}|z^t)$ in terms of the LDDP (Jaynes, 1957; 1968),*
*(2) minimizes the number of edges in $G^*$, and*
*(3) no intervention variables $I_i^t, I_j^t$ are deterministically related, i.e., $\forall j \neq i : \neg(\exists f, \forall t : I_i^t = f(I_j^t))$.*

Intuitively, this theorem shows that we can identify the minimal causal variables, even when instantaneous effects are present, under the same constraints as CITRIS. The proof in Appendix D follows three main steps. First, we show that the true observation function constitutes a global optimum of Equation (3), but is not necessarily unique. Second, we derive that any global optimum must identify the minimal causal variables. Finally, we show that optimizing the data likelihood identifies the complete causal graph, *i.e.*, instantaneous and temporal, between the minimal causal variables.

# 4 CAUSAL REPRESENTATION LEARNING WITH INSTANTANEOUS EFFECTS

Based on our theoretical results, we propose iCITRIS, a generalization of CITRIS (Lippe et al., 2022b). We first review the original CITRIS architecture, and then describe our extensions in iCITRIS.

## 4.1 BASELINE: CITRIS

CITRIS is build upon a variational autoencoder (VAE) (Kingma et al., 2014), where the (convolutional) encoder and decoder approximate the invertible map $g_\theta$. To promote the identification of the causal variables in latent space, the prior of the VAE follows Equation (2), excluding the instantaneous parents. All latent distributions $p_\phi$ are usually implemented as conditional Gaussians, with the mean and std predicted by a small MLP per latent variable. Finally, the VAE is trained via maximum likelihood on $p(X^{t+1}|X^t, I^{t+1})$. Alternatively, CITRIS can also be trained on the representations of a pretrained autoencoder, where the map $g_\theta$ is replaced by a normalizing flow (Rezende et al., 2015). We follow the same setup in iCITRIS, but extend CITRIS' prior to instantaneous effects.

## 4.2 LEARNING THE INSTANTANEOUS CAUSAL GRAPH

To learn the instantaneous causal graph simultaneously with the causal representation, we incorporate recent differentiable score-based causal discovery methods in iCITRIS. Given a distribution over graphs $p(G)$, the conditional distribution over the latent variables $z^{t+1}$ of Equation (2) becomes:

$$p_{\phi,\psi,G}\left(z^{t+1}|z^t, I^{t+1}\right) = p_\phi\left(z^{t+1}_{\psi_0}|z^t\right) \cdot \mathbb{E}_{G \sim p(G)}\left[\prod_{i=1}^{K} p_\phi\left(z^{t+1}_{\psi_i}|z^t, z^{t+1}_{\psi^{pa}_i}, I^{t+1}_i\right)\right] \tag{4}$$

where the parent sets, $z_{\psi^{pa}_i}$, depend on the graph structure $G$. The goal is to jointly optimize $p_\phi$ and $p(G)$ under maximizing the likelihood objective of Equation (3), such that $p(G)$ is peaked at the correct causal graph. To this end, we experiment with two causal discovery methods that allow for continuous optimization: NOTEARS (Zheng et al., 2018), and ENCO (Lippe et al., 2022a).

**NOTEARS** (Zheng et al., 2018) casts structure learning as a continuous optimization problem by providing a continuous constraint on the adjacency matrix to enforce acyclicity. Following Ng et al. (2022), we model the adjacency matrix with independent edge likelihoods, and differentially sample from it using the Gumbel-Softmax trick (Jang et al., 2017). We use these samples as graphs in the prior $p_\phi\left(z^{t+1}|z^t, I^{t+1}\right)$ to mask the parents of the individual causal variables, and obtain gradients for the graph through the maximum likelihood objective of the prior. In order to promote acyclicity, we use the constraint as a regularizer, and exponentially increase its weight over training.

**ENCO** (Lippe et al., 2022a), on the other hand, uses interventional data and two separate parameter sets: one for the orientation per edge, and one for the existence per edge. By only using interventions to update the orientation parameters, ENCO converges to the true, acyclic graph under single-target interventions in the sample limit. Yet, we found it to also work well under multi-target interventions in iCITRIS. ENCO uses low-variance, unbiased gradients based on REINFORCE (Williams, 1992), potentially providing a more stable optimization than NOTEARS. For efficiency, we merge the two learning stages of ENCO and update both the graph and distribution parameters at each iteration.

## 4.3 STABILIZING THE OPTIMIZATION PROCESS

Simultaneously identifying the causal variables and their graph leads to a chicken-and-egg situation: without knowing the variables, we cannot identify the graph; but without knowing the graph, we cannot identify the causal variables. This can cause the optimization to be unstable and to converge to local minima with incorrect graphs. To stabilize it, we propose the two following approaches.

**Graph learning scheduler** During the first training iterations, the assignment of latent to causal variables is almost random, since the gradients for the graph parameters are very noisy and uninformative. Thus, we use a learning rate schedule for the graph learning parameters such that the graph parameters are frozen for the first couple of epochs. During those training iterations, the model learns to fit the latent variables to the intervention variables under an arbitrary graph, leading to an initial, rough assignment of latent to causal variables. Then, we warm up the learning rate to slowly start the graph learning process while continuing to separate the causal variables in latent space.

**Mutual information estimator** If the provided interventions are fully perfect, *i.e.*, they remove temporal dependencies as well, we can exploit this independence by masking out the temporal parents in the prior distribution under interventions. Furthermore, with perfect interventions, we also enforce the mutual information (MI) (Kullback, 1997) between parents and children under interventions to be zero as an additional regularization. Following work on neural MI estimation (Belghazi et al., 2018; Hjelm et al., 2019; van den Oord et al., 2018), we train a network to distinguish between samples from the joint distribution $p(z_{\psi_i}^t, z_{\psi_i^{\text{pa}}}^t, z^{t-1} | I_i^t = 1)$ and the product of their marginals, $p(z_{\psi_i}^t | I_i^t = 1)p(z_{\psi_i^{\text{pa}}}^t, z^{t-1} | I_i^t = 1)$. While the MI estimator optimizes its accuracy, the latents are optimized to do the opposite, effectively forcing $z_{\psi_i}^t$ and its parents to be independent under interventions.

## 5 EXPERIMENTS

We evaluate iCITRIS on three video datasets with varying difficulties and compare it to common causal representation methods. We include further dataset details in Appendix E, discuss hyperparameters in Appendix F, and provide the code at https://github.com/phlippe/CITRIS.

### 5.1 EXPERIMENTAL SETTINGS

**Baselines** Since iCITRIS is, to the best of our knowledge, the first method to identify causal variables with instantaneous effects in this setting, we compare it to methods for identifying conditionally independent causal variables. Firstly, we use CITRIS (Lippe et al., 2022b) and the Identifiable VAE (iVAE) (Khemakhem et al., 2020a), which both use the previous time step and intervention targets to model conditionally independent variables. Further, to compare to a model with dependencies among latent variables, we evaluate the iVAE with an autoregressive prior, which we denote with iVAE-AR. All methods share the general model setup, *e.g.*, the encoder network architecture, where possible.

**Evaluation metrics** To evaluate the identification of the causal variables, we follow Lippe et al. (2022b) and report the $R^2$ scores for correlations. In particular, $R_{ij}^2$ is the score between the true causal variable $C_i$ and the latents that have been assigned to the causal variable $C_j$ by the learned model, *i.e.*, $z_{\psi_j}$. We denote the average correlation of the predicted variable to its true value with $R^2 \text{ diag} = 1/K \sum_i R_{ii}^2$ (optimal 1), and the maximum correlation besides its true variable with $R^2 \text{ sep} = 1/K \sum_i \max_{j \neq i} R_{ij}^2$ (optimal 0). Furthermore, to investigate the modeling of the temporal and instantaneous relations between the causal variables, we perform causal discovery as a post-processing step on the latent representations since the baselines do not explicitly learn the graph, and report the Structural Hamming Distance (SHD) between the predicted and true causal graph.

### 5.2 2D COLORED CELLS WITH CAUSAL EFFECTS: VORONOI BENCHMARK

We first conduct experiments on synthetically generated causal graphs with various instantaneous structures to investigate the difficulty and challenges of the task. We consider three instantaneous graph structures: `random` has a randomly sampled, acyclic graph structure with a probability of 0.5 of two variables being connected by a direct edge, and `chain` and `full` represent the minimally- and maximally-connected DAGs respectively. For each graph, we sample temporal edges with an edge probability of 0.25 matching the density of the instantaneous causal graph. Based on these graphs, we create the variable's observational distributions as Gaussians parameterized by randomly initialized neural networks, and provide for simplicity single-target, perfect interventions for all variables. The causal variables are mapped to image space $\mathcal{X}$ by firstly applying a randomly initialized, two-layer normalizing flow, and afterwards plotting them as colors in a 32x32 pixels image of a fixed Voronoi diagram as an irregular structure. Thus, the representation learning models need to distinguish between the entanglement by the random normalizing flow and the underlying causal graphs to identify the causal variables, while also performing causal discovery to find the correct causal graph.

In Figure 2, we show the results of all models on graphs of 4, 6, and 9 variables. For the `random` and `chain` graphs, iCITRIS-ENCO identifies the causal variables and their causal graph with only minor errors, even for the largest graphs of 9 variables. Even on the challenging `full` graph, iCITRIS-ENCO considerably outperforms the other models. In contrast, iCITRIS-NOTEARS struggles with the edge orientations and converges to edge probabilities noticeably lower than 1.0, with which the variables cannot be perfectly identified anymore, especially for increasing graph sizes. Meanwhile,

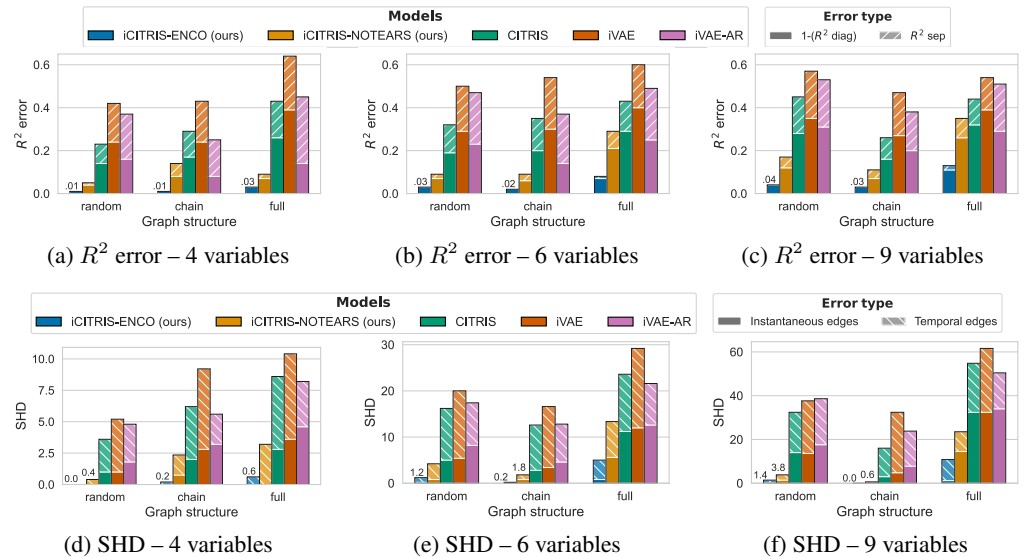

Figure 2: Results on the Voronoi benchmark over three graph structures and sizes with five seeds (error bars in Appendix G.1). For all metrics, lower is better. **Top row** (a-c): Plotting the $R^2$ correlation error as the average distance between predicted and true causal variables (1-"$R^2$ diag", solid bars), plus the maximum correlation to any other variable ("$R^2$ sep", striped bars). iCITRIS-ENCO performs well across graph structures and sizes. **Bottom row** (d-f): The SHD between predicted and ground truth causal graph, divided into instantaneous (solid bars) and temporal (striped bars) edges. iCITRIS-ENCO obtains the lowest error across graphs, with close to zero for the graphs `random` and `chain`.

CITRIS and iVAE find $K$ independent dimensions, conditioned on the previous time step and the intervention targets, instead of the true causal variables, which leads to sparse instantaneous, but wrongly dense temporal graphs. Finally, the autoregressive baseline, iVAE-AR, naturally entangles all dimensions in the latent space, on which the true causal graph cannot be recovered anymore. This underlines the non-triviality of identifying instantaneously-related causal variables. In conclusion, iCITRIS identifies the causal variables and graph well across graph structures and sizes, with ENCO outperforming NOTEARS due to more stable optimization, especially for larger, complex graphs.

## 5.3   3D OBJECT RENDERINGS: INSTANTANEOUS TEMPORAL CAUSAL3DIDENT

As a visually challenging dataset, we use the Temporal Causal3DIdent dataset (Lippe et al., 2022b; von Kügelgen et al., 2021) which contains 3D renderings ($64 \times 64$ pixels) of different object shapes under varying positions, rotations, and lights. The dataset has seven causal variables, including categorical and circular variables, going beyond iCITRIS's theoretical setting. To introduce instantaneous effects, we replace all temporal relations with instantaneous edges, except those on the same variable ($C_i^t \rightarrow C_i^{t+1}$). For instance, a change in the rotation leads to an instantaneous change in the position of the object, which again influences the spotlight. Overall, we obtain an instantaneous graph of eight edges between the seven multidimensional causal variables.

We provide partially-perfect interventions that remove instantaneous parents, but leave the existing temporal dependencies unchanged. Since the dataset is visually complex, we use the normalizing flow variant of iCITRIS and CITRIS applied on a pretrained autoencoder.

Table 1 shows that iCITRIS-ENCO identifies the causal variables well and recovers most instantaneous relations, with up to two errors on average. The temporal graph had more false positive edges due to minor correlations. iCITRIS-

Table 1: Results on the Instantaneous Temporal Causal3DIdent dataset over three seeds (standard deviations in Table 9). iCITRIS-ENCO performs best in identifying the variables and their graph.

| Model | $R^2$ (diag $\uparrow$ / sep $\downarrow$) | SHD (instant $\downarrow$ / temp $\downarrow$) |
|---|---|---|
| iCITRIS-ENCO | **0.96 / 0.07** | **1.67 / 5.67** |
| iCITRIS-NOTEARS | 0.95 / 0.10 | 4.33 / 6.33 |
| CITRIS | 0.90 / 0.23 | 5.67 / 12.67 |
| iVAE | 0.79 / 0.24 | 6.00 / 15.00 |
| iVAE-AR | 0.74 / 0.29 | 10.67 / 12.33 |

NOTEARS incorrectly orients several edges during training, underlining the benefit of ENCO as the graph learning method in iCITRIS. The baselines have a significantly higher entanglement of the causal variables and struggle with finding the true causal graph. Further, in Appendix G.2, we apply iCITRIS to the original Temporal Causal3DIdent dataset, which contains only temporal causal relations and no instantaneous effects. In this setting, iCITRIS performs on par with CITRIS, verifying that iCITRIS generalizes CITRIS across datasets. In summary, iCITRIS-ENCO can identify the causal variables along with their instantaneous graph well, even in a visually challenging dataset.

## 5.4 REAL GAME DYNAMICS: CAUSAL PINBALL

Finally, we consider a simplified version of the game Pinball, which naturally has instantaneous causal effects: if the paddles are activated when the ball is close, the ball is accelerated immediately. Similarly, when the ball hits a bumper, its light turns on and the score increases immediately. This results in instantaneous effects, especially under common frame rates. In this environment, we consider five causal variables: the position of the left paddle, the right paddle, the ball (position and velocity), the state of the bumpers, and the score. Interventions again remove instantaneous, but keep temporal parents. Pinball is closer to a real-world environment than the other two datasets and has two characteristic differences: (1) many aspects of the environment are deterministic, *e.g.*, the ball movement, and (2) the instantaneous effects are

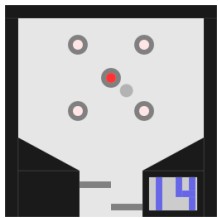

Figure 3: Example of Causal Pinball.

sparse, *e.g.*, the paddles do not influence the ball if it is far away of them. Such a setting violates several assumptions like faithfulness, the full support and potential symmetries in the observational and interventional distributions, questioning whether iCITRIS empirically works here.

The results in Table 2 suggest that iCITRIS still works well on this environment. Besides identifying the causal variables well, iCITRIS-ENCO identifies the instantaneous causal graph with minor errors. In contrast, CITRIS entangles the variables much stronger, while iVAE has difficulties identifying all variables in the environment. This shows that iCITRIS can be applied in challenging environments beyond our theoretical limitations, even with deterministic causal effects, while maintaining strong empirical results.

Table 2: Results on the Causal Pinball dataset over three seeds (see Table 12 for standard deviations).

| Model | $R^2$ (diag ↑ / sep ↓) | SHD (instant ↓ / temp ↓) |
|---|---|---|
| iCITRIS-ENCO | **0.99** / 0.12 | **0.67** / **3.00** |
| iCITRIS-NOTEARS | 0.98 / 0.18 | 3.33 / 4.67 |
| CITRIS | 0.90 / 0.39 | 3.00 / 7.67 |
| iVAE | 0.44 / **0.05** | 4.33 / 4.67 |
| iVAE-AR | 0.47 / 0.15 | 8.00 / 3.67 |

## 6 CONCLUSION AND DISCUSSION

We propose iCITRIS, a causal representation learning framework for temporal intervened sequences with potentially instantaneous effects. From such sequences, iCITRIS identifies the minimal causal variables while jointly learning the instantaneous and temporal causal graph. In experiments, iCITRIS accurately recovers the causal variables and their graph in three video datasets.

Since instantaneous effects are common in real-world settings (Hyvärinen et al., 2008; Nuzzi et al., 2021), we believe that iCITRIS contributes an important step towards practical causal representation learning methods. Still, as with most other theoretical results, our identifiability theorem is limited by the assumptions it takes. The two most crucial assumptions in iCITRIS are having a dataset, potentially recorded by an expert, that has (1) non-deterministically related, known intervention targets and (2) partially-perfect interventions, *i.e.*, interventions that can remove instantaneous parents. Without the first assumption, causal variables may become entangled in the latent space, and without the latter, instantaneous causal relations may be predicted where none truly exist. However, as demonstrated in experiments on Causal3DIdent and Causal Pinball, iCITRIS still achieves a strong empirical performance in settings that violate other assumptions. For instance, in these experiments, the distributions had limited support and some variables had circular or categorical domains.

To extend iCITRIS to even more settings, future work includes investigating a setup where interventions are not directly available, but can be performed by sequences of actions, and targets must be learned in an unsupervised manner. Further, iCITRIS is limited to acyclic graphs, while for instantaneous effects cycles could occur under low frame rates, which is also an interesting future direction.

AUTHOR CONTRIBUTIONS

P. Lippe conceived the idea, derived the theoretical results, implemented the models and datasets, and wrote the paper. S. Magliacane, S. Löwe, Y. M. Asano, T. Cohen, E. Gavves advised during the project and helped in writing the paper.

ACKNOWLEDGMENTS

We thank Johann Brehmer and Pim de Haan for valuable discussions throughout the project. We also thank SURFsara for the support in using the Lisa Compute Cluster. This work is financially supported by Qualcomm Technologies Inc., the University of Amsterdam and the allowance Top consortia for Knowledge and Innovation (TKIs) from the Netherlands Ministry of Economic Affairs and Climate Policy.

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

# SUPPLEMENTARY MATERIAL
# CAUSAL REPRESENTATION LEARNING FOR INSTANTANEOUS AND TEMPORAL EFFECTS

## TABLE OF CONTENTS

## A  BROADER IMPACT

The importance of causal reasoning for machine learning applications, especially reinforcement learning and latent dynamics understanding, has been emphasized by several previous works (De Haan et al., 2019; Lachapelle et al., 2022b; Pearl, 2009; Schölkopf et al., 2021; Seitzer et al., 2021; Zhang et al., 2020). Thereby, starting from low-level information like pixels constitutes a considerable challenge, since we aim at reasoning about objects and abstract concepts instead of low-level pixels. We believe that this work contributes an important step towards tackling this challenge since it goes beyond previous work by considering instantaneous effects, a common property in real-world systems (Faes et al., 2010; Hyvärinen et al., 2008; Moneta et al., 2006; Nuzzi et al., 2021). Besides providing theoretical identifiability results, we also propose a practical algorithm with which one can learn the causal variables and their graph from high-level observations. Furthermore, we envision a reinforcement learning setting as a future application, where a robotic system may be able to interact with an environment. However, the main assumption that prevent us from doing this so far, is the availability of interventions with known targets. In many systems, one might not be able to directly perform such interventions, but rather require several steps of low-level actions. For instance, instead of being provided the intervention targets, future work could consider a robotic setup where one can control a robot arm which can perform several interactions (*e.g.*, flipping a switch), and we believe that our work can constitute the starting point for such extension. Moreover, as we have seen in the experiments on the Causal3DIdent dataset and the Causal Pinball environment, not all assumptions must be strictly fulfilled to identify the variables empirically. Moving towards this empirical goal, recent advances in unsupervised object-centric learning (Engelcke et al., 2020; Kipf et al., 2022; Locatello et al., 2020c) have shown that objects, which can often be considered as groups of causal variables like position and velocity, can be identified from high-dimensional data without labels. A possible combination of such object-centric approaches with our causal representation learning method can relax further assumptions by using the objects as a prior disentanglement of information, opening up further possible applications of iCITRIS. Thus, we believe that this work can form the basis of several future works in this direction.

Since the possible applications of causal representation learning and specifically iCITRIS are fairly wide-ranging, there might be potential impacts we cannot forecast at the current time. This includes misuses of the method for unethical purposes. For instance, an incorrect application of the method can be used to justify false causal relations, such as referencing gender and race as causes for other characteristics of a person. Hence, the obligation to use this method in a correct way within ethical boundaries lies on the user, and the outputs of the method should always be critically evaluated. We will emphasize this responsibility of the user in the public license of our code.

## B  REPRODUCIBILITY STATEMENT

For reproducibility, the code for all models used in this paper is publicly available at `https://github.com/phlippe/CITRIS`. Further, we provide the code for generating the Voronoi benchmark, the Instantaneous Temporal Causal3DIdent dataset, and the Causal Pinball environment. More details on the datasets and visualizations are outlined in Appendix E.

Moreover, for all experiments of Section 5, we have included a detailed overview of the hyperparameters in F.2 and additional implementation details of the evaluation metrics and model architecture components in Appendix F.1. All experiments have been repeated for at least 3 seeds (5 seeds for the Voronoi benchmark) to obtain stable, reproducible results. We provide an overview of the standard deviations, as well as additional results and ablation studies in Appendix G.

Finally, all experiments in this paper were performed on a single NVIDIA TitanRTX GPU with a 6-core CPU. The overall computation time of all experiments together in this paper correspond to approximately 80 GPU days (excluding hyperparameter search and trials during the research).

## C  EXPANDED RELATED WORK

Early works on causal representation learning focused on identifying independent factors of variations (Klindt et al., 2021; Kumar et al., 2018; Locatello et al., 2019; 2020b; Träuble et al., 2021). A related line of work, Independent Component Analysis (ICA) (Comon, 1994; Hyvärinen et al., 2001), tries

to recover independent latent variables that were transformed by some invertible transformation. ICA was extended to non-linear transformations by exploiting auxiliary variables that make latents mutually conditionally independent (Hyvärinen et al., 2016; Hyvärinen et al., 2019), combined with deep learning methods like VAEs (Khemakhem et al., 2020a;b; Reizinger et al., 2022; Sorrenson et al., 2020; Zimmermann et al., 2021) and applied to causality (Gresele et al., 2021; Monti et al., 2019; Shimizu et al., 2006). In particular, Lachapelle et al. (2022a;b); Yao et al. (2022a;b) discuss the identifiability of causal variables from temporal sequences. As forms of interventions, Lachapelle et al. (2022a;b) consider external actions, while Yao et al. (2022a;b) use non-stationary noise. Yet, in all of these ICA-based setups, causal variables are required to be conditionally independent. Alternatively, Yang et al. (2021) learn causal variables from labeled images in a supervised manner.

Given a known causal structure, von Kügelgen et al. (2021) demonstrate that common contrastive learning methods can block-identify causal variables that remain unchanged under augmentations. Locatello et al. (2020a) identify independent latent causal factors from pairs of observations that only differ in a subset of causal factors. Brehmer et al. (2022) extend this setup to variables that are causally related with access to single-target interventions. Similarly, Ahuja et al. (2022) extend the setup of Locatello et al. (2020a) to variables with interdependencies, relying on interventions that only affect their target. All these methods require pairs of counterfactual observations, where only a subset of variables is changed by the intervention, while the rest are frozen, *i.e.*, they keep the same values before and after an intervention. As discussed by Pearl (2009), however, knowing counterfactuals is not realistic in most scenarios. Instead, CITRIS (Lippe et al., 2022b) focuses on temporal sequences, in which also the variables that are not intervened upon at a given time step can still continue evolving over time. On the other hand, in this setting the intervention targets need to be known. Moreover, within a time step, the causal variables are assumed to be independent conditioned on the variables of the previous time step, hence not allowing for instantaneous effects. To the best of our knowledge, iCITRIS is the first method to identify causal variables and their causal graph from temporal, intervened sequences even for potentially instantaneous causal effects, without requiring counterfactuals or data labeled with the true causal variables.

# D   PROOFS

In this section, we provide the proof for the identifiability theorem 3.4 in Section 3 and Lemma 3.2. The section is structured into three main parts. First, in Appendix D.1, we give an overview of the notation and elements that are used in the proof. Next, we discuss the assumptions needed for Theorem 3.4, with a focus on why they are needed and what a violation of these assumptions can cause. Additionally, we provide a proof of Lemma 3.2 in this subsection. Finally, we provide the proof of Theorem 3.4, structured into multiple subsections as different main steps of the proof. A detailed overview of the proof is provided in Appendix D.3.

## D.1   PRELIMINARIES

Throughout the proof, we will the same notation as used in the main paper, and try to align it as much as possible with Lippe et al. (2022b). As a summary, we review here an adapted version of the notation and preliminaries of the proof for CITRIS:

- We denote the $K$ causal factors in the latent causal dynamical system as $C_1, \ldots, C_K$;
- The dimensions and space of a causal variable is denoted as $C_i \in \mathbb{D}_i^{M_i}$ with $M_i \geq 1$. In the remainder of the proof, we consider $\mathbb{D}_i$ to be $\mathbb{R}$, *i.e.*, $C_i$ being a continuous variable;
- We group all causal factors in a single variable $C = (C_1, \ldots, C_K) \in \mathcal{C}$, where $\mathcal{C}$ is the causal factor space $\mathcal{C} = \mathbb{D}_1^{M_1} \times \mathbb{D}_2^{M_2} \times \ldots \times \mathbb{D}_K^{M_K}$;
- The data we base our identifiability on is generated by a latent Dynamic Bayesian network with variables $(C_1^t, C_2^t, \ldots, C_K^t)_{t=1}^T$;
- We assume to know at each time step the binary intervention variables $I^t \in \{0,1\}^{K+1}$ where $I_i^t = 1$ refers to an intervention on the causal factor $C_i^t$. As a special case $I_0^t = 0$ for all $t$;
- For each causal factor $C_i$, there exists a minimal causal split $s_i^{\text{var}}(C_i), s_i^{\text{inv}}(C_i)$ such that $s_i^{\text{var}}(C_i)$ represents *only* the variable/manipulable part of $C_i$, while $s_i^{\text{inv}}(C_i)$ represents the invariable part of $C_i$;
- At each time step, we can access observations $x^t, x^{t+1} \in \mathcal{X} \subseteq \mathbb{R}^N$;
- There exist a bijective mapping between observations and causal/noise space, denoted by $h : \mathcal{C} \times \mathcal{E} \to \mathcal{X}$, where $\mathcal{E}$ is the space of the noise variable. The bijective map implies that the observations, $\mathcal{X}$, live in a lower-dimensional manifold of size $\dim(\mathcal{C} \times \mathcal{E})$ in $\mathbb{R}^N$. For example, in Causal Pinball, we have a limited set of images that can occur. Formally, this means that there exists an inverse to the observation function, $h^{-1}$, such that $h(h^{-1}(X)) = X$ for all $X \in \mathcal{X}$, and $h^{-1}(h([C; E])) = [C; E]$ for all $C \in \mathcal{C}, E \in \mathcal{E}$.
- The noise $E^t \in \mathcal{E}$ at a time step $t$ subsumes all randomness besides the causal model which influences the observations. For example, this could be brightness shifts in Causal3D, or color shifts in the Causal Pinball environment since in these setups, no causal factor is encoded in brightness and color respectively. While this setting is quite general, we still require that the values of the causal factors must be identifiable from single observations. Hence, the joint dimensionality of the observation noise and causal model is limited to the image size.
- For any model learning a latent space, we denote the vector of latent variables by $z^t \in \mathcal{Z} \subseteq \mathbb{R}^M$, where $\mathcal{Z}$ is the latent space of dimension $M = \dim(\mathcal{E}) + \dim(\mathcal{C})$. In practice, we usually overestimate $M$, *i.e.*, $M > \dim(\mathcal{E}) + \dim(\mathcal{C})$;
- In iCITRIS, we learn the inverse of the observation function as $g_\theta : \mathcal{X} \to \mathcal{Z}$. If $\dim(\mathcal{X}) > \dim(\mathcal{C} \times \mathcal{E})$, we consider that $g_\theta$ contains an arbitrary, fixed map from $\mathcal{X}$ to the lower dimensionality. This map can, in theory, be trivially found by ensuring invertibility of all $X \in \mathcal{X}$ while minimizing the number of dimensions. An alternative interpretation is that $g_\theta$ is a deterministic variational autoencoder with zero reconstruction loss. In this limit, Nielsen et al. (2020) showed that the encoder-decoder function as an invertible normalizing flow, which is what we base our analysis on as well. In practice, we train a deterministic autoencoder with a latent dimension greater than $\dim(\mathcal{E}) + \dim(\mathcal{C})$, and work on this larger dimensionality;
- In iCITRIS, we learn an assignment from latent dimensions to causal factors, denoted by $\psi : [\![1..M]\!] \to [\![0..K]\!]$;
- The latent variables assigned to each causal factor $C_i$ by $\psi$ are denoted as $z_{\psi_i} = \{z_j | j \in [\![1..M]\!], \psi(j) = i\} = \{g_\theta(x^t)_j | j \in [\![1..M]\!], \psi(j) = i\}$;
- The remaining latent variables that are not assigned to any causal factor are denoted as $z_{\psi_0}$;
- In iCITRIS, we learn a directed, acyclic graph $G = (V, E)$ where $V = \{z_{\psi_i} | i \in [\![0..K]\!]\}$

and the edges represent directed causal relations;

- The graph $G$ induces a parent structure which we denote by $z_{\psi_i^{\mathrm{pa}}} = \{z_j | j \in [\![1..M]\!], \psi(j) \in \mathrm{pa}_G(i)\}$ where $\mathrm{pa}_G(0) = \emptyset$, *i.e.*, the variables in $z_{\psi_0}$ having no instantaneous parents;

- The parents of a causal variable within the same time step $t+1$ are denoted by $\mathrm{pa}^{t+1}(C_i^{t+1})$, and the parents of the previous time step $t$ by $\mathrm{pa}^t(C_i^{t+1})$;

- As a special case, we denote the function $g_\theta$ with the parameters $\theta$ that precisely model the inverse of the true observation function, $h^{-1}$, as the disentanglement function $\delta^* : \mathcal{X} \to \tilde{\mathcal{C}} \times \tilde{\mathcal{E}}$ with $\tilde{\mathcal{C}} = \mathbb{D}^{\tilde{M}_1} \times ... \times \mathbb{D}^{\tilde{M}_K}$ and $\tilde{M}_i$ being the number of latent dimensions assigned to the causal factor $C_i$ by $\psi^*$. We denote the output of $\delta^*$ for an observation $X$ as $\delta^*(X) = (\tilde{C}_1, \tilde{C}_2, ..., \tilde{E})$. The representation of $\delta^*$ as a learnable function is denoted by $g_\theta^*$ and $\psi^*$;

- In the following proof, we will use entropy as a measure of information content in a random variable. To be invariant to possible invertible transformations, *e.g.*, scaling by 2, we use the notion of the limiting density of discrete points (LDDP) (Jaynes, 1957; 1968). In contrast to differential entropy, LDDP introduces an *invariant measure* $m(X)$, which can be seen as a reference distribution we measure the entropy of $p(X)$ to. The entropy is thereby defined as:

$$H(X) = -\int p(X) \log \frac{p(X)}{m(X)} dx \tag{5}$$

In the following proof, we will consider entropy measures over latent and causal variables. For the latent variables, we consider $m(X)$ to be the push-forward distribution of an arbitrary, but fixed distribution in $\mathcal{X}$ (*e.g.*, random Gaussian if $\mathcal{X} = \mathbb{R}^n$) through $g_\theta$. For the causal variables, we consider it to be the push-forward through $h^{-1}$. For more details on LDDP, see Lippe et al. (2022b, Appendix A.1.2) and Jaynes (1957; 1968).

## D.2 ASSUMPTIONS FOR IDENTIFIABILITY

In this section, we provide a detailed discussion of the assumptions of iCITRIS to enable the identification of an underlying causal graph with instantaneous effects. We thereby focus on why these assumptions are necessary, and how a violation of those can lead to scenarios where the causal variables and graph is not identifiable.

### D.2.1 ASSUMPTION 1: THE INTERVENTIONS ON THE CAUSAL VARIABLES REMOVE INSTANTANEOUS PARENTS

iCITRIS requires interventions on the causal variables that remove instantaneous parents, in order to separate the variables in latent space, as stated in Lemma 3.2 and copied here for completeness:

**Lemma D.1.** *In iTRIS, a causal variable $C_i$ cannot be identified up to an invertible transformation $T_i$ that is independent of all other causal variables, if $C_i$ can have instantaneous parents and no partially-perfect interventions on $C_i$ are provided.*

*Proof.* To prove this Lemma, consider a causal variable $C_i$ that has $C_j$ as an instantaneous parent. The conditional distribution of $C_i$, as defined in iTRIS, can be written as $p_i(C_i^{t+1}|C_j^{t+1}, S, I_i^{t+1})$, where $S \subseteq C^t \cup C^{t+1} \setminus \{C_i^{t+1}, C_j^{t+1}\}$, *i.e.*, any additional parent set without introducing cycles. We do not put any constraints on the distribution $p$ and also on the provided interventions, except that we do *not* have the knowledge whether under $I^{t+1} = 1$, $C_i^{t+1}$ becomes independent of $C_j^{t+1}$ or not. This implies that one must consider the most general form of interventions for $C_i$, *i.e.*, modeling the distribution $p_i(C_i^{t+1}|C_j^{t+1}, S, I_i^{t+1} = 1)$ under interventions with possible unknown independences. To keep this result general, we consider an arbitrary observation function $h(C^t, E^t) = X^t$.

Under this setting, it is sufficient to show that there exist another representation $\hat{C}$ that cannot be distinguished from $C$ solely based on observation triples $\{X^t, X^{t+1}, I^{t+1}\}$, and that there exist no invertible function $f$ such that $f(\hat{C}_i^t) = C_i^t$ for any $t$. Note that we exclude a permutation of variables, since the intervention targets $I^{t+1}$ align the two representations.

As an alternative representation, consider $\hat{C} = \{C_1, ..., C_{i-1}, C_i + C_j, C_{i+1}, ..., C_K\}$ with $K$ being the number of causal variables. Then, the distribution of $\hat{p}(\hat{C})$ only differs in the conditional of $p_i$ as

follows:

$$\hat{p}_i(\hat{C}_i^{t+1}|\hat{C}_j^{t+1}, \hat{S}, I_i^{t+1}) = \hat{p}_i(\hat{C}_i^{t+1}|C_j^{t+1}, S, I_i^{t+1}) \tag{6}$$

$$= \hat{p}_i(C_i^{t+1} + C_j^{t+1}|C_j^{t+1}, S, I_i^{t+1}) \tag{7}$$

Because $\hat{p}_i$ is conditioned on $C_j^{t+1}$, there exist an invertible, volume-preserving transformation $w$ from $C_i^{t+1} + C_j^{t+1}$ to $C_i^{t+1}$, *i.e.,*, $w(c|C_j^{t+1}) = c - C_j^{t+1}$. Hence, it follows that:

$$\hat{p}_i(C_i^{t+1} + C_j^{t+1}|C_j^{t+1}, S, I_i^{t+1}) = p_i(C_i^{t+1}|C_j^{t+1}, S, I_i^{t+1}) \tag{8}$$

and overall that $\hat{p}(\hat{C}) = p(C)$. Furthermore, there exist a function $\hat{h}$ that maps $\hat{C}$ to the same observations as $h$ does for $C$:

$$\hat{h}(\hat{C}^t, E^t) = h(\{\hat{C}_1^t, ..., \hat{C}_{i-1}^t, \hat{C}_i^t - \hat{C}_j^t, \hat{C}_{i+1}^t, ..., \hat{C}_K^t\}, E^t) = h(C^t, E^t) \tag{9}$$

Therefore, both representations, $C$ and $\hat{C}$, can model the same data generation process for $\{X^t, X^{t+1}, I^{t+1}\}$, and are indistinguishable from these observations alone. Finally, it is apparent that there exist no invertible transformation from $\hat{C}_i^t$ to $C_i^t$ that is independent of $C_j^t$. Thus, the causal variable $C_i$ is not identifiable up to invertible, componentwise transformations. □

As an example of how this effects a standard identification problem, consider two random, causal variables $C_1, C_2$ with the causal graph $C_1^t \to C_1^{t+1}, C_2^t \to C_2^{t+1}$. The two causal variables $C_1, C_2$ have therefore no instantaneous relations. Further, consider the (soft-interventional) distributions $p_1(C_1^{t+1}|C_t^1, I_1^{t+1})$ and $p_2(C_2^{t+1}|C_2^1, I_2^{t+1})$ whose form can be arbitrary, but for this example, we choose them to be Gaussian with constant variance:

$$p_1(C_1^{t+1}|C_1^t, I_1^{t+1}) = \begin{cases} \mathcal{N}(C_1^{t+1}|\mu_1(C_1^t), \sigma_1(C_1^t)^2) & \text{if } I_1^{t+1} = 0 \\ \mathcal{N}(C_1^{t+1}|\tilde{\mu}_1(C_2^t), \tilde{\sigma}_1(C_1^t)^2) & \text{if } I_1^{t+1} = 1 \end{cases} \tag{10}$$

$$p_2(C_2^{t+1}|C_2^t, I_2^{t+1}) = \begin{cases} \mathcal{N}(C_2^{t+1}|\mu_2(C_2^t), \sigma_2(C_2^t)^2) & \text{if } I_2^{t+1} = 0 \\ \mathcal{N}(C_2^{t+1}|\tilde{\mu}_2(C_2^t), \tilde{\sigma}_2(C_2^t)^2) & \text{if } I_2^{t+1} = 1 \end{cases} \tag{11}$$

where $\mu_1, \tilde{\mu}_1, \mu_2, \tilde{\mu}_2, \sigma_1, \tilde{\sigma}_1, \sigma_2, \tilde{\sigma}_2$ are arbitrary, potentially non-linear functions of $C_1^t$ and $C_2^t$ respectively. Further, to consider the simplest case, suppose that the observation $X^t$ at a time step $t$ are the causal variables themselves, $X^t = [C_1^t, C_2^t]$, and we observe data points of all intervention settings, *i.e.,* $I_i^{t+1} \sim \text{Bernoulli}(q)$ with $0 < q < 1$.

Under this setup, the true generative model follows the distribution:

$$p(X^{t+1}|X^t, I^{t+1}) = p(C_1^{t+1}, C_2^{t+1}|C_1^t, C_2^t, I_1^{t+1}, I_2^{t+1}) \tag{12}$$

$$= p(C_1^{t+1}|C_1^t, C_2^t, I_1^{t+1}, I_2^{t+1}) \cdot p(C_2^{t+1}|C_1^t, C_2^t, I_1^{t+1}, I_2^{t+1}) \tag{13}$$

$$= p_1(C_1^{t+1}|C_1^t, I_1^{t+1}) \cdot p_2(C_2^{t+1}|C_2^t, I_2^{t+1}) \tag{14}$$

where $C_1^{t+1} \perp\!\!\!\perp C_2^{t+1}|X^t, I^{t+1}$. To show that the causal variables are not uniquely identifiable, we need at least one other representation which can achieve the same likelihood as the true generative model under all intervention settings $I^{t+1}$. For this, consider the following distribution:

$$p(X^{t+1}|X^t, I^{t+1}) = p(C_1^{t+1}, C_2^{t+1}|C_1^t, C_2^t, I_1^{t+1}, I_2^{t+1}) \tag{15}$$

$$= p(C_1^{t+1}|C_1^t, C_2^t, I_1^{t+1}, I_2^{t+1}) \cdot p(C_2^{t+1}|C_1^t, C_2^t, C_1^{t+1}, I_1^{t+1}, I_2^{t+1}) \tag{16}$$

$$= p_1(C_1^{t+1}|C_1^t, I_1^{t+1}) \cdot \hat{p}_2(C_1^{t+1} + C_2^{t+1}|C_2^t, C_1^{t+1}, I_2^{t+1}) \tag{17}$$

$$= p_1(\hat{C}_1^{t+1}|C_1^t, I_1^{t+1}) \cdot \hat{p}_2(\hat{C}_2^{t+1}|C_2^t, \hat{C}_1^{t+1}, I_2^{t+1}) \tag{18}$$

with $\hat{C}_1^{t+1} = C_1^{t+1}, \hat{C}_2^{t+1} = C_1^{t+1} + C_2^{t+1}$. Note the additional dependency of $\hat{C}_2^{t+1}$ on $\hat{C}_1^{t+1}$, which is possible in the space of possible causal models with an additional instantaneous causal edge

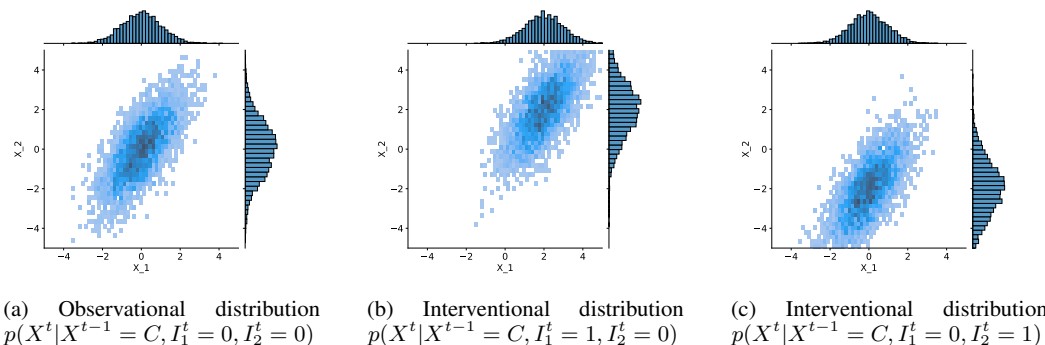

(a) Observational distribution $p(X^t|X^{t-1} = C, I_1^t = 0, I_2^t = 0)$

(b) Interventional distribution $p(X^t|X^{t-1} = C, I_1^t = 1, I_2^t = 0)$

(c) Interventional distribution $p(X^t|X^{t-1} = C, I_1^t = 0, I_2^t = 1)$

Figure 4: Example distribution for showcasing the necessity of partially-perfect interventions for disentangling causal variables with instantaneous effects. Suppose we are given two-dimensional observations $X^t$, for which the observational and interventional distributions are plotted in (a)-(c). The central plot of each subfigure shows a 2D histogram, and the subplots above and on the right show the 1D marginal histograms. For simplicity, we keep the previous time step, $X^{t-1}$, constant here. From the interventional distribution, one might suggest that we have the latent causal graph $C_1 \rightarrow C_2$ since under $I_1^t = 1$, the distribution of both observational distributions change, while $I_2^t = 1$ keeps $X_2$ unchanged. However, the data has been actually generated from two independent causal variables, which have been entangled by having $X^t = [C_1^t, C_1^t + C_2^t]$. We cannot distinguish between these two latent models from interventions that do not reliably break instantaneous causal effects, showing the need for partially-perfect interventions.

$\hat{C}_1^{t+1} \rightarrow \hat{C}_2^{t+1}$. The new distribution $\hat{p}_2$ is identical to the true distribution, since:

$$\hat{p}_2(C_1^{t+1} + C_2^{t+1}|C_2^t, C_1^{t+1}, I_2^{t+1} = 0) = \mathcal{N}(C_1^{t+1} + C_2^{t+1}|C_1^{t+1} + \mu_2(C_2^t), \sigma_2(C_2^t)^2) \tag{19}$$

$$= \frac{1}{\sqrt{2\pi}\sigma_2(C_2^t)} \exp\left(-\frac{1}{2} \frac{\left(C_1^{t+1} + C_2^{t+1} - (C_1^{t+1} + \mu_2(C_2^t))\right)^2}{\sigma_2(C_2^t)^2}\right) \tag{20}$$

$$= \frac{1}{\sqrt{2\pi}\sigma_2(C_2^t)} \exp\left(-\frac{1}{2} \frac{\left(C_2^{t+1} - \mu_2(C_2^t)\right)^2}{\sigma_2(C_2^t)^2}\right) \tag{21}$$

$$= \mathcal{N}(C_2^{t+1}|\mu_2(C_2^t), \sigma_2(C_2^t)^2) \tag{22}$$

$$= p_2(C_2^{t+1}|C_2^t, I_2^{t+1} = 0) \tag{23}$$

Similarly, one can show that $\hat{p}_2(C_1^{t+1} + C_2^{t+1}|C_2^t, C_1^t, I_2^{t+1} = 1) = p_2(C_2^{t+1}|C_2^t, I_2^{t+1} = 1)$. Hence, the alternative representation $\hat{C}_1^{t+1}, \hat{C}_2^{t+1}$ can model the distribution $p(X^{t+1}|X^t, I^{t+1})$ as well as the true causal model. In conclusion, from the samples alone, we cannot distinguish between the two representation $C_1, C_2$ and $\hat{C}_1, \hat{C}_2$, and the model is therefore not identifiable up to invertible transformations.

An alternative example with a non-trivial observation function is visualized in Figure 4, which further underlines the problem.

This shows that with soft interventions, one cannot distinguish between causal relations introduced by the observation function and those that are in the true causal model. (Partially-)Perfect interventions, however, provide an opportunity to do so since if we had known that the intervention on $C_2$ renders it independent of $C_1$, the second causal model could not have modeled the correct distribution under $I_2 = 1$. Thus, we can distinguish between the two, allowing us to identify the correct causal model.

Note that under partially-perfect intervention, the intervention-independent part of a causal variable, $s^{\text{inv}}(C_i^t)$, automatically cannot have any instantaneous parents, since otherwise, the intervention does not remove all instantaneous parents and hence is actually not partially-perfect.

### D.2.2 ASSUMPTION 2: THE INTERVENTION VARIABLES ARE NOT A DETERMINISTIC FUNCTION OF EACH OTHER

iCITRIS builds upon interventions to identify the causal variables. The intervention targets are not necessarily independent of each other, but can be confounded. For instance, we could have a setting where we only obtain single-target interventions, or a certain variable $C_i$ can only be jointly intervened upon with another variable $C_j$. In this large space of possible experimental settings, we naturally cannot guarantee identifiability all the time. In particular, we require that intervention targets for the different causal variables are unique:

**Lemma D.2.** *All information that is strictly dependent on the intervention target* $I_i^t$, *i.e.* $s^{\text{var}}(C_i)$ - *the minimal causal variable of* $C_i$, *cannot be disentangled from another causal variable,* $C_j$ *with* $j \neq i$, *if their intervention targets are identical:* $\forall t, I_i^t = I_j^t$.

*Proof.* Lippe et al. (2022b) have shown that two causal variables $C_i, C_j$ cannot be disentangled from observational data alone if they follow a Gaussian distribution with equal variance over time. Taking this setup, consider that additionally to observational data, we observe samples where both variables have been intervened upon, $I_i^{t+1} = I_j^{t+1} = 1$. If the interventional distribution of $C_i$ and $C_j$ are both Gaussian with the same variance, we have the same non-identifiability as in the observational case. Since the entanglement axes can transfer between the two setups, $C_i$ and $C_j$ cannot be disentangled, and therefore their minimal causal variables. $\square$

In other words, if two variables are always jointly intervened or passively observed, we cannot distinguish whether information belongs to causal variable $C_i$ or $C_j$. Since the causal system is stationary, having one time step $t$ for which $I_i^t \neq I_j^t$ implies that in the sample limit, we will observe samples with $I_i^t \neq I_j^t$ in the limit as well. Further, when we only observe joint interventions on two variables, $C_i, C_j$, the causal graph among the two variable cannot be identified for arbitrary distributions (Eberhardt, 2007), making the identifiability of the graph and variables impossible.

Following Lippe et al. (2022b), we require that the following independence holds for every causal variable $C_i$ with observed interventions:

$$C_i^{t+1} \not\perp\!\!\!\perp I_i^{t+1} | C^t, \text{pa}^{t+1}(C_i^{t+1}), I_j^{t+1} \text{ for any } i \neq j \tag{24}$$

This also implies that there does not exist a variable $C_j$ for which $\forall t, I_i^t = 1 - I_j^t$. As mentioned before, under additional assumptions such that every causal variable has at least one parents, it can be relaxed to unique interventions.

### D.2.3 ASSUMPTION 3: DISTRIBUTIONS HAVE FULL SUPPORT

Following several previous works (Brehmer et al., 2022; von Kügelgen et al., 2021), we consider for the theoretical results that all distributions have full support. If the observational and interventional distribution do not share the same support, there exist data points for which the intervention targets can be determined from the observation $X^t$ alone. In such situation, the encoder can change its encoding depending on the intervention target, as long as the decoder can yet recover the full observation. This can potentially create representation models that ignore the latent structure, since the intervention targets are already known. Furthermore, when intervention targets are known from seeing causal variables, we potentially introduce new independencies from intervention targets. For instance, if we have the graph $C_1, C_2 \to C_3$ where $I_3 = 1$ only if $I_1 = 1, I_2 = 0$, we can induce the intervention targets from other causal factors, making $C_3$ essential independent of $I_3$. To prevent such degenerate solutions, we take the assumption that the observational and intervention distributions share the same support. This assumption implies that any data point could come from either the interventional or observational regime, ensuring that the intervention target cannot deterministically be found from an observation $X^t$.

### D.2.4 ASSUMPTION 4: TEMPORAL CONNECTIONS AND INTERVENTIONS BREAK ALL SYMMETRIES IN THE DISTRIBUTIONS

The temporal and interventional dependencies are an essential part in iCITRIS to guarantee identifiability and disentanglement of the causal variables. Without any of these dependencies, there may exist multiple representations that model the same distribution $p(X^t | X^{t-1}, I^t)$, while following the

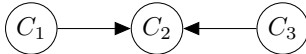

Figure 5: Example instantaneous causal graph between 3 causal variables $C_1, C_2, C_3$. Without temporal dependencies, we could encode information of $C_1$ dependent on $C_3$ without needing an edge in the distribution.

enforced latent structure by iCITRIS. The problem is that variables can functional dependent on each other, where these dependencies exploit symmetries, leaving the distribution unchanged.

For instance, consider the instantaneous causal graph of three variables $C_1, C_2, C_3$ with $C_1, C_3 \rightarrow C_2$, as depicted in Figure 5. Suppose that $C_1$ does not have any temporal parents, and the observational distribution of it follows a Gaussian: $p(C_1^t | I_1^t = 0) = \mathcal{N}(C_1^t | \mu_1, \sigma_1^2)$ with $\mu_1, \sigma_1^2$ being constants. Further, suppose that under interventions, only the standard deviation changes, i.e. $p(C_1^t | I_1^t = 1) = \mathcal{N}(C_1^t | \mu_1, \tilde{\sigma}_1^2)$ with $\tilde{\sigma}_1^2 \neq \sigma_1^2$. Then, for any point $C_1^t = c_1$, there exists a second point, $c_1' = 2\mu_1 - c_1$, which has the same probability for any value of $I_1^t$. This is because both distributions, $p(C_1^t | I_1^t = 0)$ and $p(C_1^t | I_1^t = 1)$, share a symmetry around the mean $\mu_1$.

Now, suppose we have the optimal encoder which maps an observation $X^t$ of this system to the three causal variables with their ground truth values. Then, there exist an alternative encoder, which flips the observed value of $C_1^t$ around the mean $\mu_1$, deterministically conditioned on the remaining variables $C_2^t$ and $C_3^t$. For instance, we could have the following representation $\hat{C}_1^t, \hat{C}_2^t, \hat{C}_3^t$ for the causal variables:

$$\hat{C}_2^t = C_2^t, \hat{C}_3^t = C_3^t, \hat{C}_1^t = \begin{cases} C_1^t & \text{if } \hat{C}_3^t > 0 \\ 2\mu_1 - C_1^t & \text{otherwise} \end{cases} \tag{25}$$

This alternative representation model shares the same likelihood as the optimal encoder in terms of $p(X^t | X^{t-1}, I^t)$, since flipping the value of $C_1^t$ around the mean does not change its probability. Further, despite the flipping, the original observation $X^t$ can be recovered from this alternative representation $\hat{C}^t$ by the decoder, because the possible conditioning factors, i.e. $\hat{C}_3^t$ in this case, are observable to the decoder. Hence, both representations are equally valid for the causal models. Yet, one cannot recover the value of the true causal variable, $C_1^t$, from its alternative representation $\hat{C}_1^t$ alone, since $\hat{C}_3^t$ needs to be known to invert the example condition. This shows that we can have functional dependencies between representations of causal variables while their distributions remain independent. Thus, there exist more than one representation that cannot be distinguished between from having samples of $p(X^t | X^{t-1}, I^t)$ alone.

More generally speaking, functional dependencies between variables can be introduced if there exists a transformation that leaves the probability of a variable $C_i$ unchanged for any possible value of its parents unseen in $X^t$, i.e. its intervention target $I_i^t$ and temporal parents $C^{t-1}$. Whether this transformation is performed or not can now be conditioned on other variables at time step $t$. Meanwhile, this transformation does not introduce additional dependencies in the causal graph, since the distribution does not change.

To prevent such transformations from being possible, the temporal parents and intervention targets need to break all symmetries in the distributions. We can specify it in the following assumption:

**Assumption 4**: *For a causal variable $C_i$ and its causal mechanism $p(C_i^{t+1} | pa^{t+1}(C_i^{t+1}), pa^t(C_i^{t+1}), I_i^{t+1})$, there exist no invertible, smooth transformation $T$ with $T(C_i^{t+1} | C_{-i}^{t+1}) = \tilde{C}_i^{t+1}$ besides the identity, for which the following holds:*

$$\forall C^t, C^{t+1}, I^{t+1} : p(C_i^{t+1} | pa^{t+1}(C_i^{t+1}), pa^t(C_i^{t+1}), I_i^{t+1}) =$$
$$\left| \frac{\partial T(C_i^{t+1} | C_{-i}^{t+1})}{\partial C_i^{t+1}} \right| \cdot p(\tilde{C}_i^{t+1} | pa^{t+1}(C_i^{t+1}), pa^t(C_i^{t+1}), I_i^{t+1}) \tag{26}$$

Intuitively, this means that there does not exist any symmetry that is shared across all possible values of the parents (temporal and interventions) of a causal variable. While this might first sound restricting, this assumption will likely hold in most practical scenarios. For instance, if the distribution is a Gaussian, then the assumption holds as long as the mean is not constant since the intervention breaks any parent dependencies are broken by the perfect interventions. The same holds in higher

dimensions, as the new symmetries, i.e. rotations, are yet broken if the center point is not constant. Note that these symmetries can be smooth transformations, in contrast to the discontinuous flipping operation on the Gaussian (*i.e.*, either we flip the distribution or not, but there is no step in between).

### D.2.5 ASSUMPTION 5: CAUSAL GRAPH STRUCTURE REQUIREMENTS

Besides disentangling and identifying the true causal variables, we are also interested in finding the instantaneous causal graph. This requires us to perform causal discovery, for which we need to take additional assumptions. First, we assume that the causal graph is acyclic, *i.e.*, for any causal variable $C_i^t$, there does not exist a path through the directed causal graph that loops back to it. Note that this excludes different instances over time, meaning that a path from $C_i^t$ to $C_i^{t+\tau}$ is not considered a loop. In real-world setups, there potentially exist instantaneous graphs which are not acyclic, which essentially model a feedback loop over multiple variables. However, to rely on the graph as a distribution factorization, we assume it to be acyclic, and leave extension to cyclic causal graphs for future work. As the second causal graph assumption, we require that the causal graph is faithful, which means that all independences between causal variables are implications of the graph structure, not the specific parameterization of the distributions (Hyttinen et al., 2013; Pearl, 2009). Without faithfulness, the graph might not be fully recoverable. Finally, we assume causal sufficiency, *i.e.*, there do not exist any additional latent confounders that introduce dependencies between variables beyond the ones we model. Note that this excludes the potential latent confounder between the intervention targets, and we rather focus on confounders on the causal variables $C_1, ..., C_K$ besides their intervention targets, the previous time step $C^t$, and instantaneous parents $C^{t+1}$.

### D.3 THEOREM 3.4 - PROOF OUTLINE

The goal of this section is to proof Theorem 3.4: the global optimum of iCITRIS will identify the minimal causal variables and their instantaneous causal graph. The proof follows a similar structure as Lippe et al. (2022b) used for proofing the identifiability in CITRIS, but requires additional steps to integrate the possible instantaneous relations. In summary, we will take the following steps:

1. (Appendix D.4) Firstly, we show that the function $\delta^*$ that finds the true latent variables $C_1, ..., C_K$ and assigns them to the corresponding sets $z_{\psi_1}, ..., z_{\psi_K}$ constitutes a global, but not necessarily unique, optimum for maximizing the likelihood $p(X^{t+1}|X^t, I^{t+1})$.

2. (Appendix D.5) Next, we characterize the class of disentanglement functions $\Delta^*$ which all represent a global maximum of the likelihood, *i.e.*, get the same score as the true function $\delta^*$. We do this by proving that all functions in $\Delta^*$ must identify the minimal causal variables.

3. (Appendix D.6) In a third step, we show that based on the identification of the minimal causal variables, the causal graph on these learned representations must contain at least the same edges as in the ground truth graph.

4. (Appendix D.7) Finally, we put all parts together and derive Theorem 3.4.

We will make use of Figure 6 summarizing the temporal causal graph, and the notation introduced in Appendix D.1. For the remainder of the proof, we assume for simplicity of exposition that:

- The invertible map $g_\theta$ and the prior $p_\phi \left( z^{t+1}|z^t, I^{t+1} \right)$ are sufficiently complex to approximate any possible function and distribution one might consider in iTRIS. In practice, over-parameterized neural networks can approximate most functions with sufficient accuracy.

- The sample size for the provided experimental settings is unlimited. This ensures that dependencies and conditional independencies in the causal graph of Figure 6 transfer to the observed dataset, and no additional relations are introduced by sample biases. In practice, a large sample size is likely to give an accurate enough description of the true distributions.

### D.4 THEOREM 3.4 - PROOF STEP 1: THE TRUE MODEL IS A GLOBAL OPTIMUM OF THE LIKELIHOOD OBJECTIVE

We start the identifiability discussion by proving the following Lemma:

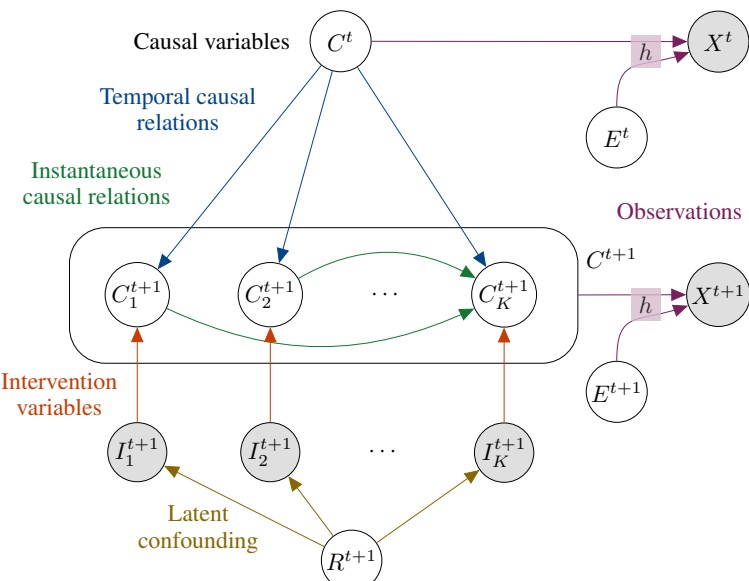

Figure 6: An example causal graph in iTRIS. A latent causal factor $C_i^{t+1}$ can have as potential parents the causal factors at the previous time step $C^t = (C_1^t, \ldots, C_K^t)$, instantaneous parents $C_j^{t+1}, i \neq j$, and its intervention variables $I_i^{t+1}$. All causal variables $C^{t+1}$ and the noise $E^{t+1}$ cause the observation $X^{t+1}$. $R^{t+1}$ is a potential latent confounder between the intervention targets.

**Lemma D.3.** *The true identification function $\delta^*$ that correctly identifies the true causal factors $C_1^{t+1}, \ldots, C_K^{t+1}$ from observations $X^t, X^{t+1}$ using the true $\psi^*$ assignment function on the latent variables $Z^{t+1}$ and the true causal graph $G^*$ is one of the global maxima of the likelihood of $p(X^{t+1}|X^t, I^{t+1})$.*

This lemma ensures that the true model is part of the solution space of maximum likelihood objective on $p(X^{t+1}|X^t, I^{t+1})$.

*Proof.* In order to prove this, we first rewrite the objective in terms of the true causal factors. This can be done by using the causal graph in Figure 6, which represents the true generative model:

$$p(X^t, X^{t+1}, C^t, C^{t+1}, I^{t+1}) = p(X^{t+1}|C^{t+1}) \cdot \left[ \prod_{i=1}^{K} p(C_i^{t+1}|C^t, \text{pa}_G^{t+1}(C_i^{t+1}), I_i^{t+1}) \right] \cdot$$
$$p(X^t|C^t) \cdot p(C^t) \cdot p(I^{t+1}) \tag{27}$$

The context variable $R^{t+1}$ is subsumed in $p(I^{t+1})$, since it is a confounder between the intervention targets and is independent of all other factors given $I^{t+1}$.

In order to obtain $p(X^{t+1}|X^t, I^{t+1})$ from $p(X^t, X^{t+1}, C^t, C^{t+1}, I^{t+1})$, we need to marginalize out $C^t$ and $C^{t+1}$, and condition the distribution on $X^t$ and $I^{t+1}$:

$$p(X^{t+1}|X^t, I^{t+1}) = \int_{C^{t+1}} \int_{C^t} p(X^{t+1}|C^{t+1}) \cdot \left[ \prod_{i=1}^{K} p(C_i^{t+1}|C^t, \text{pa}_G^{t+1}(C_i^{t+1}), I_i^{t+1}) \right] \cdot$$
$$p(C^t|X^t) dC^t dC^{t+1} \tag{28}$$

In the assumptions with respect to the observation function $h$, we have defined $h$ to be bijective, meaning that there exists an inverse $h^{-1}$ that can identify the causal factors $C^t$ and noise variable $E^t$ from $X^t$. Using the invertible map, we can write $p(C^t|X^t) = \delta_{h^{-1}(X^t) = [C^t; \cdot]}$, where $\delta$ is a Dirac delta. We also remove $E^t$ from the conditioning set since it is independent of $X^{t+1}$. This leads us to:

$$p(X^{t+1}|X^t, I^{t+1}) = \int_{C^{t+1}} \left[ \prod_{i=1}^{K} p(C_i^{t+1}|C^t, \text{pa}_G^{t+1}(C_i^{t+1}), I_i^{t+1}), I_i^{t+1}) \right] \cdot p(X^{t+1}|C^{t+1}) dC^{t+1}$$
$$\tag{29}$$

We can use a similar step to relate $X^{t+1}$ with $C^{t+1}$ and $E^{t+1}$. However, since we model a distribution over $X^{t+1}$, we need to respect possible non-volume preserving transformations. Hence, we use the change of variables formula with the Jacobian $J_h = \frac{\partial h(C^{t+1}, E^{t+1})}{\partial C^{t+1} \partial E^{t+1})}$ of the observation function $h$ to obtain:

$$p(X^{t+1}|X^t, I^{t+1}) = |J_h|^{-1} \cdot \left[ \prod_{i=1}^{K} p(C_i^{t+1}|C^t, \mathrm{pa}_G^{t+1}(C_i^{t+1}), I_i^{t+1}) \right] \cdot p(E^{t+1}) \tag{30}$$

Since Equation (30) is a derivation of the true generative model $p(X^t, X^{t+1}, C^t, C^{t+1}, I^{t+1})$, it constitutes a global optimum of the maximum likelihood. Hence, one cannot achieve higher likelihoods by reparameterizing the causal factors or having a different graph, as long as the graph is directed and acyclic.

In the next step, we relate this maximum likelihood solution to iCITRIS, more specifically, the prior of iCITRIS. For this setting, the learnable, invertible map $g_\theta$ is identical to the inverse of the observation function, $h^{-1}$. In terms of the latent variable prior, we have defined our objective of iCITRIS as:

$$p_\phi\left(z^{t+1}|z^t, I^{t+1}\right) = \prod_{i=0}^{K} p_\phi\left(z_{\psi_i}^{t+1}|z^t, z_{\psi_i^{\mathrm{pa}}}^{t+1}, I_i^{t+1}\right) \tag{31}$$

Since we know that $g_\theta^*$ is an invertible function between $\mathcal{X}$ and $\mathcal{Z}$, we know that $z^t$ must include all information of $X^t$. Thus, we can also replace it with $z^t = [C^t, E^t]$, giving us:

$$p_\phi\left(z^{t+1}|C^t, E^t, I^{t+1}\right) = \prod_{i=0}^{K} p_\phi\left(z_{\psi_i}^{t+1}|C^t, E^t, z_{\psi_i^{\mathrm{pa}}}^{t+1}, I_i^{t+1}\right) \tag{32}$$

Next, we consider the assignment function $\psi^*$. The optimal assignment function $\psi^*$ assigns sufficient dimensions to each causal factor $C_1, ..., C_K$, such that we can consider $z_{\psi_i^*}^{t+1} = C_i^{t+1}$ for $i = 1, ..., K$. Further, the same graph $G$ is used in the latent space as in the ground truth, except that we additionally condition $z_{\psi_i^*}, i = 1, ..., K$ on $z_{\psi_0^*}$. With that, Equation (32) becomes:

$$p_\phi\left(z^{t+1}|C^t, E^t, I^{t+1}\right) = \left[ \prod_{i=1}^{K} p_\phi\left(z_{\psi_i^*}^{t+1} = C_i^{t+1}|C^t, z_{\psi_i^{\mathrm{pa}}}^{t+1}, z_{\psi_0^*}^{t+1}, I_i^{t+1}\right) \right] \cdot p(z_{\psi_0^*}^{t+1}|C^t, E^t) \tag{33}$$

where we remove $E^t$ from the conditioning set for the causal factors, since know that $C^{t+1}$ and $E^{t+1}$ is independent of $E^t$. Now, $z_{\psi_0^*}$ must summarize all information of $z^{t+1}$ which is not modeled in the causal graph. Thus, $z_{\psi_0^*}$ represents the noise variables: $z_{\psi_0^*}^{t+1} = E^{t+1}$.

$$p_\phi\left(z^{t+1}|C^t, E^t, I^{t+1}\right) = \left[ \prod_{i=1}^{K} p_\phi\left(z_{\psi_i^*}^{t+1} = C_i^{t+1}|C^t, z_{\psi_i^{\mathrm{pa}}}^{t+1}, z_{\psi_0^*}^{t+1}, I_i^{t+1}\right) \right] \cdot p(z_{\psi_0^*}^{t+1} = E^{t+1}|C^t, E^t) \tag{34}$$

Finally, by using $g_\theta^*$, we can replace the distribution on $z^{t+1}$ by a distribution on $X^{t+1}$ by the change of variables formula:

$$p_\phi\left(X^{t+1}|C^t, E^t, I^{t+1}\right) = \left|\frac{\partial g_\theta^*(z^{t+1})}{\partial z^{t+1}}\right| \cdot \left[ \prod_{i=1}^{K} p_\phi\left(z_{\psi_i^*}^{t+1} = C_i^{t+1}|C^t, z_{\psi_i^{\mathrm{pa}}}^{t+1}, z_{\psi_0^*}^{t+1}, I_i^{t+1}\right) \right] \cdot$$
$$p(z_{\psi_0^*}^{t+1} = E^{t+1}|C^t, E^t) \tag{35}$$

We can simplify this distribution by using the independencies of the noise term $E^{t+1}$ in the causal graph of Figure 6:

$$p_\phi\left(X^{t+1}|C^t, E^t, I^{t+1}\right) = \left|\frac{\partial g_\theta^*(z^{t+1})}{\partial z^{t+1}}\right| \cdot \left[ \prod_{i=1}^{K} p_\phi\left(z_{\psi_i^*}^{t+1} = C_i^{t+1}|C^t, z_{\psi_i^{\mathrm{pa}}}^{t+1}, I_i^{t+1}\right) \right] \cdot$$
$$p(z_{\psi_0^*}^{t+1} = E^{t+1}) \tag{36}$$

With this, Equation (36) represents the exact same distribution as Equation (30). Therefore, we have shown that the function $\delta^*$ that identifies the true latent variables $C_1, ..., C_K$ and assigns them to the corresponding sets $z_{\psi_1}, ..., z_{\psi_K}$ constitutes a global optimum for maximizing the likelihood. However, this solution is not necessarily unique, and additional optima may exist. In the next steps of the proof, we will discuss the class of functions and graphs that lead to the same optimum. $\qquad\square$

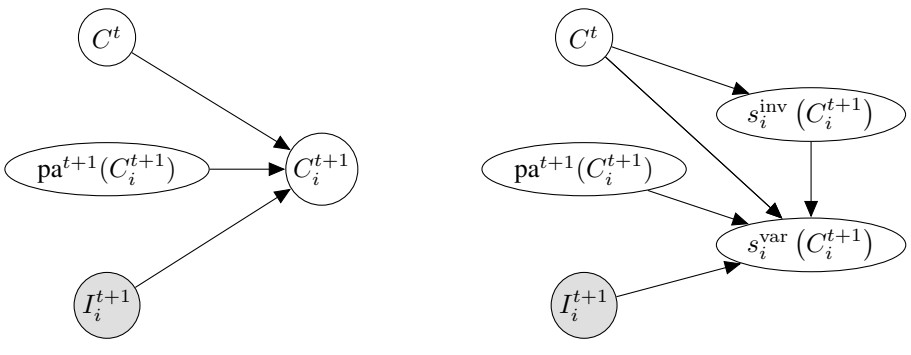

(a) Original causal graph of $C_i$          (b) Minimal causal split graph of $C_i$

Figure 7: The minimal causal variable in terms of a causal graph under iTRIS. (a) In the original causal graph, $C_i^{t+1}$ has as potential parents the causal variables of the previous time step $C^t$ (eventually a subset), its instantaneous parents $\mathrm{pa}^{t+1}(C_i^{t+1})$, and the intervention target $I_i^{t+1}$. (b) The minimal causal variable splits $C_i^{t+1}$ into an invariable part $s_i^{\mathrm{inv}}\left(C_i^{t+1}\right)$ and variable part $s_i^{\mathrm{var}}\left(C_i^{t+1}\right)$. The invariable part $s_i^{\mathrm{inv}}\left(C_i^{t+1}\right)$ is independent of the instantaneous parents and the intervention target. Further, it can be a parent of $s_i^{\mathrm{var}}\left(C_i^{t+1}\right)$ due to the autoregressive distribution modeling.

## D.5    THEOREM 3.4 - PROOF STEP 2: CHARACTERIZING THE DISENTANGLEMENT CLASS

In this section, we discuss the identifiability results of the causal variables in iCITRIS. We first describe the minimal causal variables in iTRIS, and how they differ to TRIS in CITRIS (Lippe et al., 2022b). Next, we identify the information that must be assigned to individual parts of the latent representation. Finally, we discuss the final setup to ensure identification of the variables according to Definition 3.3, including the additional variables in $z_{\psi_0}$.

### D.5.1    MINIMAL CAUSAL VARIABLES

Lippe et al. (2022b) introduced the concept of a minimal causal variable as an invertible split of a causal variable $s_i(C_i) = (s^{\mathrm{var}}(C_i), s^{\mathrm{inv}}(C_i))$ into one part that is strictly dependent on the intervention, $s^{\mathrm{var}}(C_i)$, and a part that is independent of it, $s^{\mathrm{inv}}(C_i)$ (see Definiton 2.1). In other words, the minimal causal variable is the smallest part of a causal variable that strictly depends on the provided intervention.

For iCITRIS, we consider the same concept, but adapt it to the setup of iTRIS. First, iTRIS assumes the presence of interventions that render a variable independent of its instantaneous parents. Hence, when given these interventions, we can ensure that $s^{\mathrm{inv}}(C_i)$ does not have any instantaneous parents. Second, the presence of a causal graph in iCITRIS allows dependencies between different parts of the latent space. Further, $z_{\psi_0}$ can be the parent of any other set of variables, thus allowing for potential dependencies between $s^{\mathrm{inv}}(C_i)$ and $s^{\mathrm{var}}(C_i)$. Note that those for the same time step, however, must also be cut off by the intervention. Hence, the split $s_i(C_i^t) = (s_i^{\mathrm{var}}(C_i^t), s_i^{\mathrm{inv}}(C_i^t))$ must have the following distribution structure:

$$p\left(s_i(C_i^{t+1})|C^t, \mathrm{pa}^{t+1}(C_i^{t+1}), I_i^{t+1}\right) = p\left(s_i^{\mathrm{var}}(C_i^{t+1})|C^t, \mathrm{pa}^{t+1}(C_i^{t+1}), s_i^{\mathrm{inv}}(C_i^{t+1}), I_i^{t+1}\right) \cdot \\ p\left(s_i^{\mathrm{inv}}(C_i^{t+1})|C^t\right) \tag{37}$$

where

$$p\left(s_i^{\mathrm{var}}(C_i^{t+1})|C^t, \mathrm{pa}^{t+1}(C_i^{t+1}), s_i^{\mathrm{inv}}(C_i^{t+1}), I_i^{t+1}\right) = \\ \begin{cases} \tilde{p}\left(s_i^{\mathrm{var}}(C_i^{t+1})|C^t\right) & \text{if } I_i^{t+1} = 1 \\ p\left(s_i^{\mathrm{var}}(C_i^{t+1})|C^t, \mathrm{pa}^{t+1}(C_i^{t+1}), s_i^{\mathrm{inv}}(C_i^{t+1})\right) & \text{otherwise} \end{cases} \tag{38}$$

Thereby, the minimal causal variable with respect to its intervention variable $I_i^{t+1}$ is the split $s_i$ which maximizes the information content $H(s_i^{\mathrm{inv}}(C_i^t)|C^t)$. These relations are visualized in Figure 7.

Causal variables for which the intervention target is constant, *i.e.*, no interventions have been observed,

were modeled by $s^{\text{inv}}(C_i) = C_i, s^{\text{var}}(C_i) = \emptyset$ in CITRIS (Lippe et al., 2022b). Here, this does not naturally hold anymore since $s^{\text{inv}}(C_i)$ is restricted to not having any instantaneous parents. However, as stated in assumption 1, a variable without interventions cannot be an instantaneous child of any variable. Hence, for a causal variable $C_i$, if $I_i^t = 0$ for all $t$, its minimal causal split is defined as $s^{\text{inv}}(C_i) = C_i, s^{\text{var}}(C_i) = \emptyset$, as in CITRIS (Lippe et al., 2022b).

### D.5.2 IDENTIFYING THE MINIMAL CAUSAL VARIABLES

As a first step, we postulate the following lemma:

**Lemma D.4.** *For all representation functions in the class $\Delta^*$, there exist a deterministic map from the latent representation $z_{\psi_i}$ to the minimal causal variable $s^{\text{var}}(C_i)$ for all causal variables $C_i, i = 1, ..., K$.*

This lemma intuitively states that the minimal causal variable $s^{\text{var}}(C_i)$ is modeled in the latent representation $z_{\psi_i}$ for any representation that maximizes the likelihood objective. Note that this does not imply exclusive modeling yet, meaning that $z_{\psi_i}$ can contain more information than just $s^{\text{var}}(C_i)$. We will discuss this aspect in Appendix D.5.3.

*Proof.* In order to prove this lemma, we first review some relations between the conditional and joint entropy. Consider two random variables $A, B$ of arbitrary space and dimension. The conditional entropy between these two random variables is defined as $H(A|B) = H(A, B) - H(B)$ (Cover et al., 2005). Further, the maximum of the joint entropy is the sum of the individual entropy terms, $H(A, B) \leq H(A) + H(B)$ (Cover et al., 2005). Hence, we get that $H(A|B) = H(A, B) - H(B) \leq H(A) + H(B) - H(B) = H(A)$. In other words, the entropy of a random variable $A$ can only become lower when conditioned on any other random variable $B$.

Using this relation, we move now to identifying the minimal causal variables. If a minimal causal variable is the empty set, *i.e.*, $s^{\text{var}}(C_i) = \emptyset$, for instance due to not having observed interventions on $C_i$, the lemma is already true by construction since no information must be modeled in $z_{\psi_i}$. Thus, we can focus on cases where $s^{\text{var}}(C_i) \neq \emptyset$, which implies that $C_i^{t+1} \not\perp I_i^{t+1}$. Therefore, the following inequality must strictly hold:

$$H(C_i^{t+1}|C^t, C_{-i}^{t+1}) < H(C_i^{t+1}|C^t, C_{-i}^{t+1}, I_i^{t+1}) \tag{39}$$

for all $i = 1, ..., K$. Additionally, based on the assumption that the observational and interventional distributions share the same support, we know that the intervention posterior, *i.e.*, $p(I^{t+1}|X^{t+1})$, cannot be deterministic for any data point $X^{t+1}$ and intervention target $I_i^{t+1}$. Thus, we cannot derive $I_i^{t+1}$ from the observation $X^{t+1}$. Thirdly, because every latent variable is only conditioned on exactly one intervention target in iCITRIS and there exist no deterministic function between any pair of intervention targets, one cannot identify $I_i^{t+1}$ in any latent variables except $z_{\psi_i}$. Therefore, the only way in iCITRIS to fully exploit the information of the intervention target $I_i^{t+1}$ is to model its dependent information in $z_{\psi_i}$. As this information corresponds to the minimal causal variable, $s^{\text{var}}(C_i)$, any representation function must model the distribution $p(s^{\text{var}}(C_i)|...)$ in $p(z_{\psi_i}|I_i^{t+1}, ...)$ to achieve the maximum likelihood solution. This is independent of the modeled causal graph structure, meaning that if there exist representation functions with different graphs in $\Delta^*$, then all of them must model $s^{\text{var}}(C_i)$ in $z_{\psi_i}$. Finally, using assumption 4 (Appendix D.2.4), we obtain that this distributional relation implies a functional independence of $s^{\text{var}}(C_i)$ in $z_{\psi_i}$ to any other latent variable. Thus, there exists a deterministic map from $z_{\psi_i}$ to $s^{\text{var}}(C_i)$ in any of the maximum likelihood solutions. $\square$

### D.5.3 DISENTANGLING THE MINIMAL CAUSAL VARIABLES

The previous subsection showed that $z_{\psi_i}$ models the minimal causal variable $s^{\text{var}}(C_i)$. This, however, is not necessarily the only information in $z_{\psi_i}$. For instance, for two random variables $A, B \in \mathbb{R}$, the following distributions are identical:

$$p(A) \cdot p(B|A) = p(A) \cdot p(B + A|A) = p(A) \cdot p(B, A|A) \tag{40}$$

The second distribution can add additional information about $A$ arbitrarily to $B$ without changing the likelihoods. This is because the distribution is conditioned on $A$, and the conditional entropy of a random variable to itself is $H(A|A) = H(A, A) - H(A) = H(A) - H(A) = 0$. Hence, for

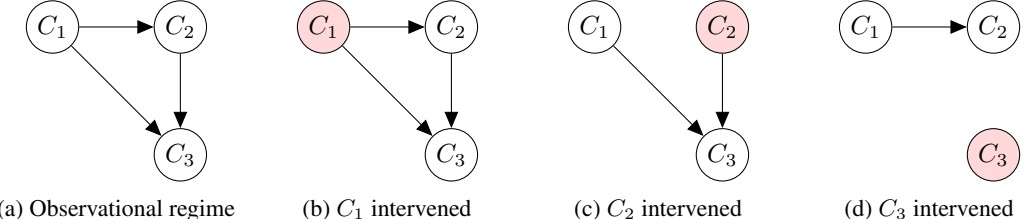

(a) Observational regime     (b) $C_1$ intervened     (c) $C_2$ intervened     (d) $C_3$ intervened

Figure 8: Example instantaneous causal graph between 3 causal variables $C_1, C_2, C_3$, and the augmented graphs under different single-target interventions that remove instantaneous parent dependencies. The augmented graphs have the edges to the intervened variables removed. For readability, the intervened variables are colored in red in the graphs.

arbitrary autoregressive distributions, we cannot identify the variables from each other purely by looking at the likelihoods.

However, in iTRIS, we are given interventions under which variables are strictly independent of their instantaneous parents. With this, we postulate the following lemma:

**Lemma D.5.** *For all representation functions in the class $\Delta^*$, $z_{\psi_i}$ does not contain information about any other minimal causal variable $s^{\text{var}}(C_j), j \neq i$, except $s^{\text{var}}(C_i)$, i.e., $H(z_{\psi_i}|s^{\text{var}}(C_i)) = H(z_{\psi_i}|s^{\text{var}}(C_i), s^{\text{var}}(C_j))$.*

*Proof.* In order to prove this lemma, we consider all augmented graph structures that are induced by the provided interventions on the instantaneous causal graph. Specifically, given a graph $G = (V, E)$ with $V$ being its vertices and $E$ its edges, and a set of binary intervention targets $I = \{I_1, ..., I_{|V|}\}$, we construct an augmented DAG $G' = (V', E')$, where $V' = V$ and $E' = E \setminus \{\{\text{pa}_G(V_i) \to V_i\}|i = 1, ..., |V|, I_i = 1\}$. In other words, the augmented graph $G'$ has all its input edges to intervened variables removed. An example for a graph of three variables and its three single-target interventions is shown in Figure 8.

A representation function in the class $\Delta^*$ must model the optimal likelihood for *all* intervention-augmented graphs of its originally learned graph $\hat{G}$, since it cannot achieve lower likelihood for any of the graphs than the ground truth. For every pair of variables $C_i, C_j$, assumption 2 (Appendix D.2.2) ensures that there exist one out of three possible experiment sets: (1) we observe $I_i^t = 1, I_j^t = 0$ and $I_i^t = 0, I_j^t = 1$, (2) $I_i^t = 0, I_j^t = 0$, $I_i^t = 1, I_j^t = 0$, and $I_i^t = 1, I_j^t = 1$, or (3) $I_i^t = 0, I_j^t = 0$, $I_i^t = 0, I_j^t = 1$, and $I_i^t = 1, I_j^t = 1$. In all cases, there exist at least one augmented graph in which $C_i^t \perp\!\!\!\perp C_j^t$, and hence $z_{\psi_i}^t \perp\!\!\!\perp z_{\psi_j}^t$, must hold since (2) and (3) observe joint interventions on both variables ($I_i^t = 1, I_j^t = 1$). In (1), a constant connection between the two variables would require both edges $C_i \to C_j$ and $C_j \to C_i$ to be present in the graph, which implies a cycle in a graph violating our acyclicity assumption 5. Under the augmented graph, where $z_{\psi_i}^t \perp\!\!\!\perp z_{\psi_j}^t$, the optimal likelihood can only be achieved if the distribution of $z_{\psi_i}^t$ is actually independent of $z_{\psi_j}^t$, thus not containing any information about $s^{\text{var}}(C_j)$. The same holds for $z_{\psi_j}$. Hence, a representation function in the class $\Delta^*$ must identify the minimal causal variables in the latent space. $\qquad\square$

### D.5.4 DISENTANGLING THE REMAINING VARIABLES

In Appendix D.5.2 and Appendix D.5.3, we have shown that for any solution in the class $\Delta^*$, we can ensure that $z_{\psi_i}$ models the minimal causal variable $s^{\text{var}}(C_i)$, and none other. Still, there exist more dimensions that need to be modeled. The causal variables without interventions, the invariant parts of the causal variables, $s^{\text{inv}}(C_i)$, as well as the noise variables $E^t$ are part of the generative model that influence an observation $X^t$. All these variables share the property that they are not instantaneous children of any minimal causal variable, and can only be parents of them. This leads to the situation that any of these variables could be modeled in the latent representation of $z_{\psi_i}$ for an arbitrary $i = 1, ..., K$ as long as $C_i$ is the parent of the same variables. The reason for this is that the distribution modeling of such variables is independent of interventions.

To exclude them from the causal variable modeling, we follow the same strategy as in CITRIS (Lippe et al., 2022b) by taking the representation function that maximizes the entropy of $z_{\psi_0}$:

**Lemma D.6.** *For all representation functions in the class $\Delta^*$ that maximize the information content of $p(z_{\psi_0}|C^t)$ according to LDDP, the latent representation $z_{\psi_i}$ models exclusively the minimal causal variable $s^{\mathrm{var}}(C_i)$ for all causal variables $C_i, i = 1, ..., K$.*

*Proof.* Using Lemma D.4 and Lemma D.5, we know that the only remaining information besides the minimal causal variables are the causal variables without interventions, invariant parts of the causal variables, $s^{\mathrm{inv}}(C_i)$, as well as the noise variables $E^t$. All these variables cannot be children of the observed, intervened variables, as the assumption 1 (Appendix D.2.1) states. Thus, the remaining information $\mathcal{M} = \{s^{\mathrm{inv}}(C_1), ..., s^{\mathrm{inv}}(C_K), E^t\}$ can be optimally modeled by $p(\mathcal{M}|z^t)p(z_{\psi_1}, ..., z_{\psi_K}|\mathcal{M}, z^t, I^{t+1})$. This implies that there exist a solution where $z_{\psi_0} = \mathcal{M}$, which can be found by searching for the solution with the maximum entropy of $p(z_{\psi_0}|C^t)$. In this solution, the latent representation $z_{\psi_1}, ..., z_{\psi_K}$ does not model any subset of $\mathcal{M}$, hence modeling the minimal causal variables exclusively. $\square$

The overall result is that we identify the minimal causal variables in $z_{\psi_1}, ..., z_{\psi_K}$, and all remaining information is modeled in $z_{\psi_0}$. Note that the causal variables without interventions, the noise variables and the invariant part of the causal variables can be arbitrarily entangled in $z_{\psi_0}$. Furthermore, since there exist variables in $z_{\psi_0}$ that may not have any temporal parents (*e.g.*, the noise variables and invariable parts of the intervened causal variables), we cannot rely on assumption 4 (Appendix D.2.4) to ensure functional independence. Hence, while the distribution of $p(z_{\psi_0}|z^t)$ is independent of $z_{\psi_1}, ..., z_{\psi_K}$, there may exist dependencies such that for a single data point, a change in $z_{\psi_i}$ can result in a change of the noise or invariable parts of the causal variables in the observational space.

## D.6    Theorem 3.4 - Proof Step 3: Identifiability of the causal graph

In this step of the proof, we discuss the identifiability of the causal graph under the previous findings. In the first subsection, we discuss what graph we can optimally find under the identification of the minimal causal variables. In the second part, we then show how the maximum likelihood objective is sufficient for identifying the instantaneous causal graph. Finally, we discuss the identifiability of the temporal causal graph.

### D.6.1    Causal graph on minimal causal variables

The identification of the causal graph naturally depends on the learned latent representations of the causal variables. In Appendix D.5, we have shown that one can only guarantee to find the minimal causal variables in iTRIS. Thus, we are limited to finding the causal graph on the minimal causal variables $s^{\mathrm{var}}(C_1), s^{\mathrm{var}}(C_2), ..., s^{\mathrm{var}}(C_K)$ and the additional variables modeled in $z_{\psi_0}$. The graph between the minimal causal variables is not necessarily equal to the ground truth graph. For instance, consider a 2-dimensional position $(x, y)$ and the color of an object as two causal variables. If the $x$-position causes the color, but the minimal causal variable of the position is only $s^{\mathrm{var}}(C_1) = y$, then the color has only $s^{\mathrm{inv}}(C_1)$ as parent, not $s^{\mathrm{var}}(C_1)$. In the learned graph on the latent representation, it would mean that we do not have an edge between $z_{\psi_1}$ and $z_{\psi_2}$, but instead $z_{\psi_0} \to z_{\psi_2}$. Hence, we might have a mismatch between the ground truth graph on the full causal variables, and the graph on the modeled minimal causal variables.

Still, there are patterns and guarantees that one can give for how the optimal, learned graph looks like. Due to the nature of the interventions, the invariable part of a causal variable, $s^{\mathrm{inv}}(C_i)$, cannot have any instantaneous parents. Thus, the instantaneous parents of a minimal causal variable $s^{\mathrm{var}}(C_i)$ are the same ground truth causal variables as in the true graph, *i.e.*, $\mathrm{pa}(C_i) = \mathrm{pa}(s^{\mathrm{var}}(C_i))$. The difference is how the parents are represented. Since each parent $C_j \in \mathrm{pa}(C_i)$ is split into a variable and invariable part, any combination of the two can represent a parent of $s^{\mathrm{var}}(C_i)$. Thus, the learned set of parents for $s^{\mathrm{var}}(C_i)$, *i.e.*, $\mathrm{pa}(z_{\psi_i})$, must be a subset of $\{s^{\mathrm{var}}(C_j)|C_j \in \mathrm{pa}(C_i)\} \cup \{z_{\psi_0}\}$. This implies that if there is no causal edge between two causal variables $C_i$ and $C_j$ in the ground truth causal graph, then there is also no edge between their minimal causal variables $s^{\mathrm{var}}(C_i)$ and $s^{\mathrm{var}}(C_j)$. The causal graph between the true variables and the minimal causal variables therefore shares a lot of similarities, and in practice, is often almost the same.

| | Exp. | $I_1$ | $I_2$ |
|---|---|---|---|
| | $E_0$ | 1 | 0 |
| | $E_1$ | 0 | 1 |

| Exp. | $I_1$ | $I_2$ |
|---|---|---|
| $E_0$ | 0 | 0 |
| $E_1$ | 1 | 0 |
| $E_2$ | 1 | 1 |

| Exp. | $I_1$ | $I_2$ |
|---|---|---|
| $E_0$ | 0 | 0 |
| $E_1$ | 0 | 1 |
| $E_2$ | 1 | 1 |

(a) Causal graph    (b) Experimental setting 1    (c) Experimental setting 2    (d) Experimental setting 3

Figure 9: Identifiability of a causal relation between two variables $C_1, C_2$ under different interventional settings. (a) The causal relation to consider. The discussion is identical in case of the reverse orientation by switching the variable names $C_1$ and $C_2$. (b-d) The tables describe the minimal sets of experiments, *i.e.*, unique combinations of $I_1, I_2$ in the dataset, that guarantee the intervention targets to be unique, *i.e.*, not $\forall t, I_1^t = I_2^t$. Under each of these sets of experiments, we show that the maximum likelihood solution of $p(C_1, C_2|I_1, I_2)$ uniquely identifies the causal orientation.

The additional latent variables $z_{\psi_0}$ summarize all invariable parts of the intervened variables, the remaining causal variables without interventions, and the noise variables. Therefore, $z_{\psi_0}$ cannot be an instantaneous child of any minimal causal variable, and we can predefine the orientation for those edges in the instantaneous graph.

Next, we can discuss the identifiability guarantees for the graph on the minimal causal variables. For simplicity, in the rest of the section, we refer to identifying the causal graph on the minimal causal variables as identifying the graph on $C_1, ..., C_K$.

#### D.6.2 OPTIMIZING THE MAXIMUM LIKELIHOOD OBJECTIVE UNIQUELY IDENTIFIES THE INSTANTANEOUS CAUSAL GRAPH UNDER INTERVENTIONS

Several causal discovery works have shown before that causal graphs can be identified when given sufficient interventions (Brouillard et al., 2020; Eberhardt, 2007; Lippe et al., 2022a; Pearl, 2009). Since the identification of the causal variables already requires interventions that render variables independent of their instantaneous parents, we can exploit these interventions for learning and identifying the graph as well. In assumption 5 (Appendix D.2.5), we have assumed that the causal graph to identify is faithful. This implies that any dependency between two variables, $C_1, C_2$, which have a causal relation among them ($C_1 \rightarrow C_2$ or $C_2 \rightarrow C_1$), cannot be replaced by conditioning $C_1$ and/or $C_2$ on other variables. In other words, in order to optimize the overall likelihood $p(C_1, ..., C_K)$, we require a graph that has a causal edge between two variables if they are causally related. Now, we are interested in whether we can identify the orientation between every pair of causal variables that have a causal relation in the ground truth graph, which leads us to the following lemma:

**Lemma D.7.** *In iTRIS, the orientation of an instantaneous causal effect between two causal variables $C_i, C_j$ can be identified by solely optimizing the likelihood of $p(C_i, C_j|I_i, I_j)$.*

*Proof.* To discuss the identifiability of the causal direction between two variables $C_1, C_2$, we need to consider all possible minimal sets of experiments that fulfill the intervention setup in assumption 2 (Appendix D.2.2). These three sets are shown in Figure 9. For all three sets, we have to show that the maximum likelihood of the conditional distribution $p(C_1, C_2|I_1, I_2)$ can only be achieved by modeling the correct orientation, here $C_1 \rightarrow C_2$. For cases where the true graph is $C_2 \rightarrow C_1$, the same argumentation holds, just with the variables names $C_1$ and $C_2$ swapped. As an overview, Table 3 shows the distribution $p(C_1, C_2|I_1, I_2)$ under all possible experiments and causal graphs.

**Experimental setting 1** (Figure 9b) In the first experimental setting, we are given single target interventions on $C_1$ and $C_2$. In the experiment $E_0$ which represents interventions on $C_1$ and passive observations on $C_2$, the dependency between $C_1$ and $C_2$ persists in the ground truth, *i.e.*, $C_1 \not\perp\!\!\!\perp C_2|I_1 = 1, I_2 = 0$. Hence, only causal graphs that condition $C_2$ on $C_1$ under interventions on $C_1$ can achieve the maximum likelihood in $E_0$. From Table 3, we see that the only causal graph that does this is $C_1 \rightarrow C_2$. Thus, when single-target interventions on $C_1$ are observed, we can uniquely identify the orientation of its outgoing edges.

**Experimental setting 2** (Figure 9c) The second experimental setting provides the observational regime ($E_0$), interventions on $C_1$ with $C_2$ being passively observed ($E_1$), and joint interventions on

Table 3: The probability distribution $p(C_1, C_2|I_1, I_2)$ for all possible causal graphs among the two causal variables $C_1, C_2$ under different experimental settings. Observational distributions are denoted with $p(...)$, and interventional with $\tilde{p}(...)$. Note that under interventions, it is enforced that $\tilde{p}(...)$ is not conditioned on any parents, since we work on the instantaneous graph.

| Interventions | | Causal graph | | |
|---|---|---|---|---|
| $I_1$ | $I_2$ | $C_1 \to C_2$ | $C_2 \to C_1$ | $C_1 \perp\!\!\!\perp C_2$ |
| 0 | 0 | $p(C_1)p(C_2|C_1)$ | $p(C_2)p(C_1|C_2)$ | $p(C_1)p(C_2)$ |
| 1 | 0 | $\tilde{p}(C_1)p(C_2|C_1)$ | $p(C_2)\tilde{p}(C_1)$ | $\tilde{p}(C_1)p(C_2)$ |
| 0 | 1 | $p(C_1)\tilde{p}(C_2)$ | $\tilde{p}(C_2)p(C_1|C_2)$ | $p(C_1)\tilde{p}(C_2)$ |
| 1 | 1 | $\tilde{p}(C_1)\tilde{p}(C_2)$ | $\tilde{p}(C_1)\tilde{p}(C_2)$ | $\tilde{p}(C_1)\tilde{p}(C_2)$ |

$C_1$ and $C_2$ ($E_2$). Since the experiment $E_1$ gives us the same setup as in experimental setting 1, we can directly conclude that the causal orientation $C_1 \to C_2$ is yet again identifiable.

**Experimental setting 3** (Figure 9d) In the final experimental setting, $C_1$ is only observed to be jointly intervened upon with $C_2$, not allowing for the same argument as in the experimental settings 1 and 2. However, the causal graph yet remains identifiable because of the following reasons. Firstly, the experiment $E_0$ with its purely observational regime cannot be optimally modeled by a causal graph without an edge between $C_1$ and $C_2$, reducing the set of possible causal graph to $C_1 \to C_2$ and $C_2 \to C_1$. Under the joint interventions $E_2$, both causal graphs model the same distribution. Still, under the experiment $E_1$ where only $C_2$ has been intervened upon, the two distributions differ. The graph with the anti-causal orientation compared to the true graph, $C_2 \to C_1$, uses the same distribution as in the observational regime to model $C_1$, *i.e.*, $p(C_1|C_2)$. In order for this to achieve the same likelihood as the true orientation, it would need to be conditioned on $I_2$ as the following derivation from the true distribution $p(C_1, C_2|I_1, I_2)$ shows:

$$p(C_1, C_2|I_1, I_2) = p(C_2|I_1, I_2) \cdot p(C_1|C_2, I_1, I_2) \tag{41}$$

$$p(C_1|C_2, I_1, I_2) = \begin{cases} p(C_1|I_1) & \text{if } I_2 = 1 \\ p(C_1|C_2, I_1) & \text{if } I_2 = 0 \end{cases} \tag{42}$$

This derivation shows that $p(C_1|C_2, I_1, I_2)$ strictly depends on $I_2$ if $p(C_1|C_2, I_1, I_2 = 1) \neq p(C_1|C_2, I_1, I_2 = 0)$, which is ensured by $C_1, C_2$ not being conditionally independent in the ground truth graph. As the causal graph $C_2 \to C_1$ models $C_1$ independently of $I_2$, it therefore cannot achieve the maximum likelihood solution in this experimental settings. Hence, the only graph achieving the maximum likelihood solution is $C_1 \to C_2$, such that the orientation can again be uniquely identified.

All other, possible experimental settings must contain one of the three previously discussed experiments as a subset, due to assumption 2 (Appendix D.2.2). Hence, we have shown that for all valid experimental settings, optimizing the maximum likelihood objective uniquely identifies the causal orientations between pairs of variables under interventions. $\square$

Based on these orientations, we can exclude all additional edges that could introduce a cycle in the graph, since we strictly require an acyclic graph. The only remaining non-identified parts of the graph are edges among variables that are independent, conditioned on their parents. In terms of maximum likelihood, these edges do not influence the objective since for two variables $C_1, C_2$ with $C_1 \perp\!\!\!\perp C_2$, $p(C_1) \cdot p(C_2) = p(C_1|C_2) \cdot p(C_2) = p(C_1) \cdot p(C_2|C_1)$. Hence, the equivalence class in terms of maximum likelihood includes all graphs that at least contain the true edges, and are acyclic. By requiring structural minimality, *i.e.*, taking the graph with the least amount of edges that fully describes the distribution, we can therefore identify the full causal graph between $C_1, ..., C_K$.

### D.6.3 IDENTIFYING THE TEMPORAL CAUSAL RELATIONS BY PRUNING EDGES

So far, we have shown that the instantaneous causal relations can be identified between the minimal causal variables. Besides the instantaneous graph, there also exist temporal relations between $C^t$ and $C^{t+1}$, which we also aim to identify:

**Lemma D.8.** *In iTRIS, the temporal causal graph between the minimal causal variables can be*

*identified by removing the edge between any pair of variables $z_{\psi_i}^t, z_{\psi_j}^{t+1}$ with $i, j \in [\![0..K]\!]$, if* $z_{\psi_i}^t \perp\!\!\!\perp z_{\psi_j}^{t+1} | z_{\psi_{-i}}^t, pa^{t+1}(z_{\psi_j}^{t+1})$.

*Proof.* The prior in Equation (2) conditions the latents variables $z^{t+1}$ on all variables of the previous time step, $z^t$. Thus, this corresponds to modeling a fully connected graph from $z_{\psi_0}^t, z_{\psi_1}^t, ..., z_{\psi_K}^t$ to $z_{\psi_0}^{t+1}, z_{\psi_1}^{t+1}, ..., z_{\psi_K}^{t+1}$. Since any temporal edge must be oriented from $z^t$ to $z^{t+1}$, it is clear that the true temporal graph, $G_T$, must be a subset of this graph. Further, since in assumption 5 (Appendix D.2.5), we have stated that the true causal model is faithful, we know that two variables, $z_{\psi_i}^t$ and $z_{\psi_j}^{t+1}$, are only connected by an edge, if they are not conditionally independent of each other: $z_{\psi_i}^t \not\perp\!\!\!\perp z_{\psi_j}^{t+1} | z_{\psi_{-i}}^t, \mathrm{pa}^{t+1}(z_{\psi_j}^{t+1})$. This implies that all redundant edges must be between two, conditionally independent variables with: $z_{\psi_i}^t \perp\!\!\!\perp z_{\psi_j}^{t+1} | \mathrm{pa}^t(z_{\psi_j}^{t+1}), \mathrm{pa}^{t+1}(z_{\psi_j}^{t+1})$ with $\mathrm{pa}^t(z_{\psi_j}^{t+1})$ being a subset of $z_{\psi_{-i}}^t$. Thus, we can find the true temporal graph by iterating through all pairs of variables $z_{\psi_i}^t$ and $z_{\psi_j}^{t+1}$, and remove the edge if both of them are conditionally independent given $z_{\psi_{-i}}^t, \mathrm{pa}^{t+1}(z_{\psi_j}^{t+1})$. $\qquad\square$

## D.7 THEOREM 3.4 - PROOF STEP 4: FINAL IDENTIFIABILITY RESULT

Using the results derived in Appendix D.4, Appendix D.5 and Appendix D.6, we are finally able to derive the full identifiability results. In Appendix D.5, we have shown that any solution that maximizes the likelihood $p_{\phi,\theta,G}(x^{t+1}|x^t, I^{t+1})$ identifies the minimal causal variables of $C_1, ..., C_K$ in $z_{\psi_1}, ..., z_{\psi_K}$. Further, we are able to summarize all remaining variables in $z_{\psi_0}$ by maximizing the entropy (LDDP) of $p_\phi(z_{\psi_0}^{t+1}|z^t)$. In Appendix D.6, we have used this disentanglement condition to show that the causal graph that maximizes the likelihood must have at least the same edges as the ground truth graph on the minimal causal variables. To obtain the full ground truth graph, we need to pick the one with the least edges.

These aspects together can be summarized into the following theorem:

**Theorem D.9.** *In iTRIS, a model $\mathcal{M}^* = \langle \theta^*, \phi^*, \psi^*, G^* \rangle$ identifies a causal system $\mathcal{S} = \langle C, E, h \rangle$ (Definition 3.3) if $\mathcal{M}^*$, under the constraint of maximizing the likelihood $p_{\phi,\theta,G}(X^{t+1}|X^t, I^{t+1})$:*
*(1) maximizes the information content $H(z_{\psi_0}^{t+1}|z^t)$ in terms of the LDDP ([Jaynes, 1957; 1968](#)),*
*(2) minimizes the number of edges in $G^*$, and*
*(3) no intervention variables $I_i^t, I_j^t$ are deterministically related,* i.e., $\forall j \neq i : \neg(\exists f, \forall t : I_i^t = f(I_j^t))$.

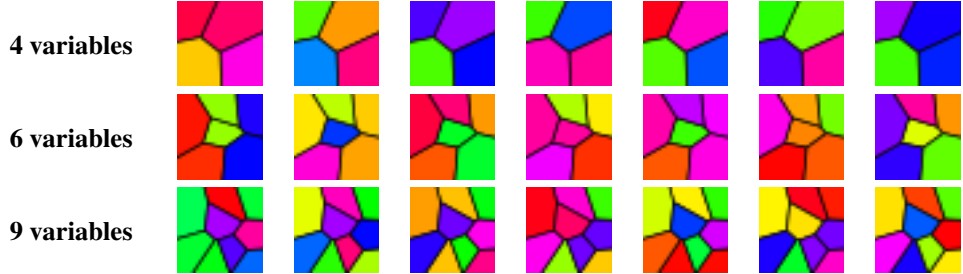

Figure 10: Example sequences of the Voronoi benchmark for the different graph sizes. Each image of $32 \times 32$ is partitioned into $K$ patches. The values of the $K$ true causal variables have been transformed by a two-layer normalizing flow, which result into the hues of the $K$ patches in $\left[-\frac{7}{8}\pi, \frac{7}{8}\pi\right]$. The hues are finally mapped into the RGB space, resulting in the images above.

## E  DATASETS

The following section gives a detailed overview of the dataset and used hyperparameters in all settings. Appendix E.1 contains the description of the Voronoi benchmark, for which the experimental results are shown in Section 5.2. Appendix E.2 discusses the Instantaneous Temporal Causal3DIdent dataset, and Appendix E.3 the Causal Pinball dataset.

### E.1  VORONOI BENCHMARK

The purpose of the Voronoi benchmark is to provide a flexible, synthetic dataset where we can evaluate causal representation learning models on various settings, such as number of variables and graph structure (both instantaneous and temporal). For each dataset, we generate one sequence with 150k time steps, in between which single-target interventions may have been performed. We sample the interventions with $1/(K+2)$ for each variable, and with $2/(K+2)$ a purely observational regime. A visual example of the Voronoi benchmark is shown in Figure 10, and we describe its generation steps below.

#### E.1.1  NETWORK SETUP

In the Voronoi benchmark, we need a data generation mechanism for the conditional distributions $p(C_i^{t+1}|\text{pa}(C_i^{t+1}))$ that support any set of parents. For this, we deploy randomly initialized neural networks which models arbitrary, non-linear relations between any parent set and a causal variable. We visualize the network architecture in Figure 11. As a simplified setup, we use the neural networks to parameterize a Gaussian distribution. Specifically, the neural networks take as input a subset of $C^t, C^{t+1}$ according to the given graph structure (see next subsection for the graph generation), and output a scalar representing the mean of the conditional distribution $\mathcal{N}(C_i^{t+1}|\mu(\text{pa}(C_i^{t+1})), \sigma^2)$ where the standard deviation is set to $\sigma = 0.3$. We have also experimented with having the (log) standard deviation as an additional output of the network. However, we experienced that this leads to the true causal variables to be the optimal solution when modeling $K$ conditionally independent factors. Hence, both iCITRIS and the baselines were able to identify the causal variables well, making the task easier than anticipated. The interventional distribution is thereby set to $\mathcal{N}(0, 1)$ for all causal variables.

On the causal variables, we apply a normalizing flow which consisted of six layers: Activation Normalization, Autoregressive Affine Coupling, Activation Normalization, Invertible 1x1 convolution, Autoregressive Affine Coupling, Activation Normalization. The Activation Normalization (Kingma et al., 2018) layers are initialized once after the Batch Normalizations of the distribution neural networks have been set, and ensure that all outputs roughly have a zero mean and standard deviation of one. The Autoregressive Affine Coupling layers use randomly initialized neural networks, with the average standard deviation of the outputs being 0.2. The coupling layer is volume preserving, *i.e.*, we do not use a scaling term in the affine coupling, to prevent any issues with the image quantization. The Invertible 1x1 convolution (Kingma et al., 2018) is initialized with a random, orthogonal matrix, entangling all causal variables across dimensions. Hence, each output of the normalizing flow is influenced by all causal variables.

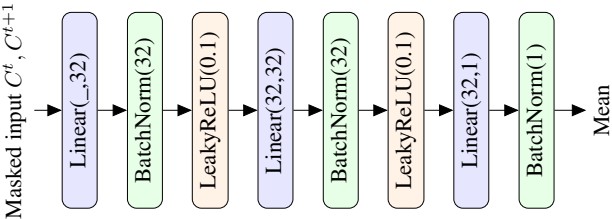

Figure 11: Network architecture of the randomly initialized neural networks in the Voronoi benchmark, modeling the conditional distributions $p(C_i^{t+1}|\text{pa}(C_i^{t+1}), I_i^{t+1} = 0) = \mathcal{N}(C_i^{t+1}|\mu(\text{pa}(C_i^{t+1})), \sigma^2)$ with $\sigma = 0.3$. The BatchNorm layers (Ioffe et al., 2015) are initialized by sequentially sampling 100 batches of the causal variables, using each as the input to the next batch. This ensures that the marginal distribution $p(C_i^{t+1})$ has a mean close to zero and standard deviation of one.

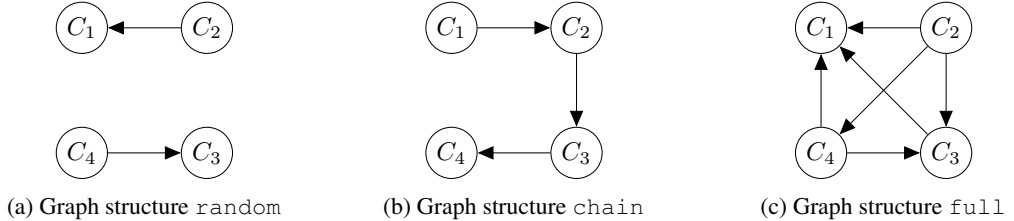

(a) Graph structure `random`         (b) Graph structure `chain`         (c) Graph structure `full`

Figure 12: Example instantaneous causal graphs with four variables for the three graph structures. The causal ordering for the causal variables is randomly sampled for each graph to prevent any structural biases.

Finally, the outputs of the normalizing flow are transformed by the function $f(x) = \frac{7}{8}\pi \cdot \tanh\left(\frac{x}{2}\right)$. This function maps all values to a range of $\left[-\frac{7}{8}\pi, \frac{7}{8}\pi\right]$, which we can use as hues in the patches of the Voronoi diagram. The division by 2 of $x$ is performed to reduce the number of data points in the saturation points of the $\tanh$. The Voronoi diagrams are generated by sampling $K$ points on the image, which have a distance of at least 5 pixels between each other, and are fixed within a dataset. In contrast to just mapping the colors into a grid, the Voronoi diagram is an irregular structure. Hence, the mapping from images to the $K$ color is non-trivial and does not transfer across datasets. Once the Voronoi diagram was created, we have used matplotlib (Hunter, 2007) to visualize the structure as an RGB image.

### E.1.2 GRAPH GENERATIONS

For the instantaneous causal graph, we have considered three graph structures: `random`, `chain`, and `full`. An example of each is visualized in Figure 12.

The `random` graph samples an edge for every possible pair of variables $C_i, C_j, i \neq j$ with a chance of $0.5$. Thereby, we ensure that the graph is acyclic by sampling undirected edges, and directing them according to a randomly sampled ordering of the variables. This way, the average number of edges in the graph is $\frac{K(K-1)}{4}$. For small graphs of size 4, this results in variables to eventually having no incoming or outgoing edges, testing also the model's ability on conditionally independent variables.

The `chain` graph connects the variables in a sequence, where each variable is the parent of the next one in the sequence. This leads to each graph having $K - 1$ edges, *i.e.*, the sparsest, yet continuously connected graph.

The `full` graph represents the densest directed acyclic graph possible. We first sample an ordering of variables, and then add an edge from each variable to all others that follow it in the sequence. Thus, it has the most possible edges in a DAG, namely $\frac{K(K-1)}{2}$.

Finally, the temporal graph is sampled similar to the `random` graph. However, the orientations are pre-determined by the temporal ordering, and no cycles can occur. We therefore sample a directed edge between any pair of variables $C_i^t, C_j^{t+1}$, including $i = j$, with a chance of $0.25$. This leads to an

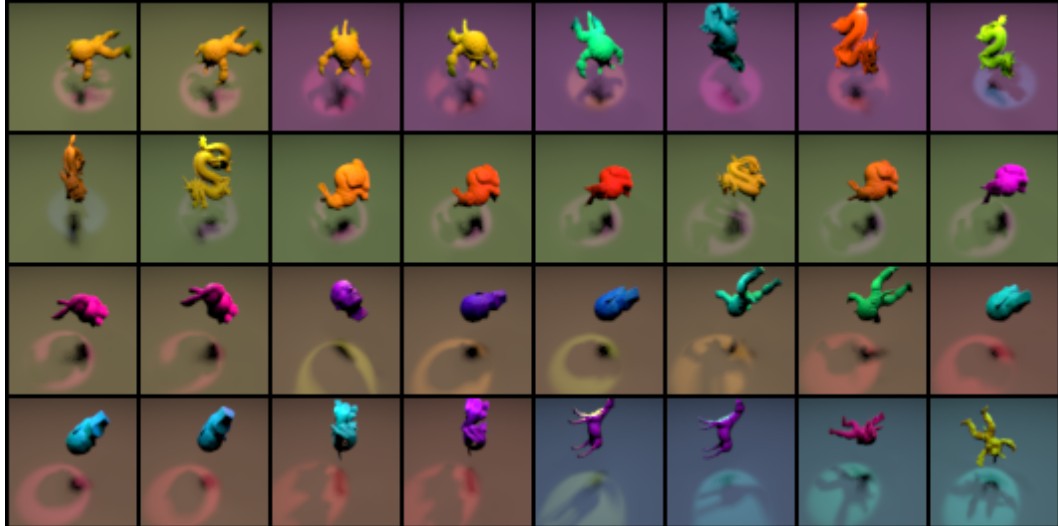

Figure 13: Example sequence from the training set of the Instantaneous Temporal Causal3DIdent dataset (from left to right, top to bottom). Each image is of size $64 \times 64$ pixels. One can see the instantaneous effects of the background influencing the object color, for instance, or the object color again influencing the rotation of the object.

average number of edges of $\frac{K^2}{4}$. Additionally, we ensure that every variable has at least one temporal parent, to prevent variance collapses in the neural network distributions.

## E.2 INSTANTANEOUS TEMPORAL CAUSAL3DIDENT

The creation of the Instantaneous Temporal Causal3DIdent dataset closely followed the setup of Lippe et al. (2022b); von Kügelgen et al. (2021), and we show an example sequence of the dataset in Figure 13. We used the code provided by Zimmermann et al. (2021)[1] to render the images via Blender (Blender Online Community, 2021), and used the following seven object shapes: Cow (Crane, 2021), Head (Rusinkiewicz et al., 2021), Dragon (Curless et al., 1996), Hare (Turk et al., 1994), Armadillo (Krishnamurthy et al., 1996), Horse (Praun et al., 2000), Teapot (Newell, 1975). As a short recap, the seven causal factors are: the object position as multidimensional vector $[x, y, z] \in [-2, 2]^3$; the object rotation with two dimensions $[\alpha, \beta] \in [0, 2\pi)^2$; the hue of the object, background and spotlight in $[0, 2\pi)$; the spotlight's rotation in $[0, 2\pi)$; and the object shape (categorical with seven values). We refer to Lippe et al. (2022b, Appendix C.1) for the full detailed dataset description of Temporal Causal3DIdent, and describe here the steps taken to adapt the datasets towards instantaneous effects.

The original temporal causal graph of the Temporal Causal3DIdent dataset contains 15 edges, of which 8 are between different variables over time. Those relations form an acyclic graph, which we can directly move to instantaneous relations. Thus, the adjacency matrix of the temporal graph is an identity matrix, while the instantaneous causal graph is visualized in Figure 14. The causal mechanisms remain unchanged, except that the inputs may now be instantaneous. For instance, the spotlight rotation is adapted as follows:

$$\text{Previous version: } \text{rot\_s}^{t+1} = f\left(\text{atan2}(\text{pos\_x}^t, \text{pos\_y}^t), \text{rot\_s}^t, \epsilon_{rs}^t\right) \tag{43}$$

$$\text{Instantaneous version: } \text{rot\_s}^{t+1} = f\left(\text{atan2}(\text{pos\_x}^{t+1}, \text{pos\_y}^{t+1}), \text{rot\_s}^t, \epsilon_{rs}^t\right) \tag{44}$$

where $f(a, b, c) = \frac{a-b}{2} + c$. The causal parents of other variables, here pos_x and pos_y, are now instantaneous instead of the previous time step. Hence, an intervention on the position will lead to an instantaneous effect on the rotation of the spotlight.

In the original dataset, all interventions have been perfect with a uniform distribution. To relax these interventions to partially-perfect interventions, we instead use interventions that are centered around

---

[1]https://github.com/brendel-group/cl-ica

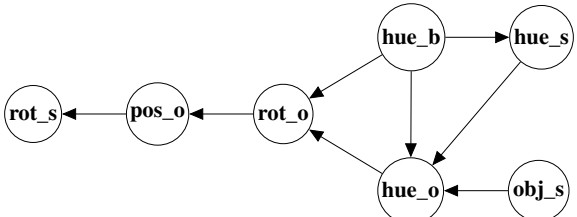

Figure 14: The instantaneous causal graph in the Instantaneous Temporal Causal3DIdent dataset. The graph contains several common sub-structures, such as a chain (rot_o→pos_o→rot_s), a fork (hue_o,hue_b→rot_o), and confounders (hue_b→hue_s,hue_o). The most difficult edges to recover include rot_o→pos_o since the object orientation has a complex, non-linear relation to the observation space which is difficult to model and prone to noise. Further, the edge hue_b,hue_s→hue_o only holds for two object shapes (Hare and Dragon), for which the background and spotlight hue have an influence on the object color. For the other five object shapes, the object color is independent of the other two parents.

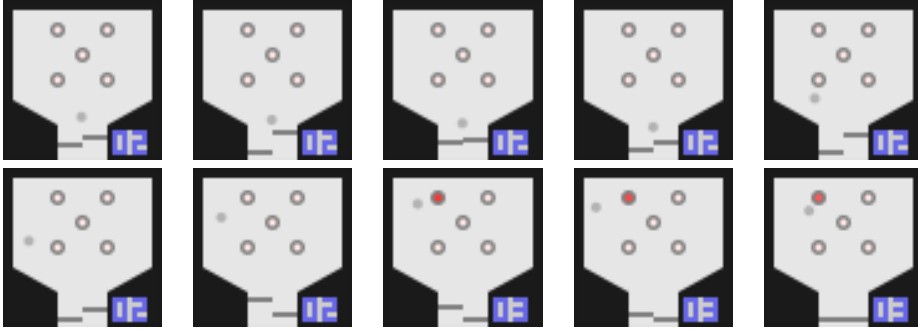

Figure 15: An example sequence of the Pinball dataset, from left to right, top to bottom. The paddles, *i.e.*, the two gray rectangles in the bottom center, are accelerated forwards under interventions such that they make a large jump within an image. For instance, in image 5, the right paddle has been intervened upon and hits the ball (gray circle). It is accelerated immediately, showcasing the instantaneous effect between the two. When no interventions on the paddles are given, they slowly move backwards. In image 8, the ball hits a bumper (5 circle centers with light red filling) which lights up. This represents the scoring of a point, as the instantaneous increase in points shows in image 8 (the digits in the bottom right corner). Note that technically, there is no winning or losing state here since we do not focus on learning a policy, but instead a causal representation of the components. Further, not shown here, there exist a fourth channel representing the ball's velocity.

the previous time step value. Specifically, for circular values, we use $C_i^{t+1} \sim \mathcal{N}(C_i^t, \sigma_i^2)$ with $\sigma_i = 2$. For the position variables, we use $C_i^{t+1} \sim \mathcal{N}^T(0.5 \cdot C_i^t, \sigma_i^2)$ with $\mathcal{N}^T$ denoting a truncated Gaussian at $[-2, 2]$ to prevent objects leaving the canvas, and $\sigma_i = 1.5$. All remaining aspects of the dataset generation are identical to the Temporal Causal3DIdent dataset.

### E.3 CAUSAL PINBALL

The Causal Pinball dataset is a simplified environment of the popular game Pinball, as shown in Figure 15. In Pinball, the user controls two fixed paddles on the bottom of the playing field, and tries to hit the ball such that it collides with various objects for scoring points. There are several versions of Pinball, but for this dataset, we limit it to the essential parts representing the five, multidimensional causal variables:

- The **ball** is defined by four dimensions: the position on the x- and y-axis, and its velocity in x and y. Both are continuous values, with the position being limited to the available spots on the field.
- The **left paddle y-position** (paddle_left) describes the position of the left paddle. Its

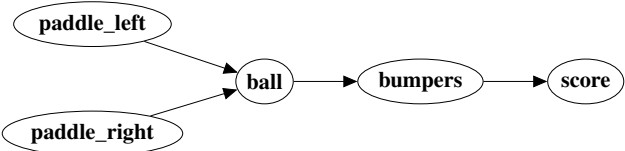

Figure 16: The instantaneous causal graph in the Causal Pinball dataset. An intervention on the paddles can have an immediate effect on the ball by changing its position and velocity. A change in the ball's position again influences the bumpers, whether their light is activated or not. Finally, when the bumpers are activated, the score increases in the same time step.

maximum is close-to the top of the black border next to it (*e.g.*, image 7 in Figure 15), and its minimum is close to the bottom (*e.g.*, image 10 in Figure 15).

- The **right paddle y-position** (paddle_right) is similar to paddle_left, just for the right paddle.

- The **bumpers** represent the activation, *i.e.*, the light, of all 5 bumpers. It is a five-dimensional continuous variable, each dimension being between 0 (light off, *e.g.*, image 1 in Figure 15) and 1 (light fully on, *e.g.*, image 8 in Figure 15).

- The **score** is a categorical variable summarizing the number of points the player has scored. Its value ranges from 0 to a maximum of 20.

The dynamics between these causal factors resembles the standard game dynamics of Pinball, which results in the instantaneous causal graph in Figure 16. The ball can collide with the paddles, borders, and bumpers. When it collides with the borders, it is simply reflected, and we reduce its velocity by 10% (*i.e.*, multiply by 0.9). Under collisions with the paddles, we distinguish between a collision where the paddle has been static or moving backwards, versus a collision where the paddle was moving. When the paddle was static, we use the same collision dynamics as the borders, except that we reduce its $y$-velocity by 70% to reduce oscillations around the paddle position. When the paddle was moving, we instead set the $y$-velocity of the ball to the $y$-velocity of the paddle. Finally, when the ball collides with a bumper, it activates the bumper's light and reflects from it, similar to the borders. When a bumper's light is turned on, we increase the score by one, but include a 5% chance that the score is not increased to introduce some stochastic elements and faulty components in the game. Next to the collisions, the ball is influenced by a gravity towards the bottom, adding a constant every time step to its $y$-velocity, and friction that reduces its velocity by 2% after each time step.

In terms of interventions, we sample the interventions on the five elements independently, but with a chance that would correspond more closely to the game dynamics. Specifically, we intervene on the paddles in 20% of the frames, 10% on the ball, and 5% each the score and bumpers. An intervention of the paddle represents it moving forwards, from its previous position, to a randomly sampled position between the middle and maximum paddle position. Its velocity is set to the difference between the previous position and new position. Since these interventions are usually elements of the standard Pinball game play, we sample them rather often with 20%. An intervention on the ball represents stopping it at the position it is, and give it slightly random velocity towards the bottom. In real-life, this would correspond to a player interfering with the ball by stopping it with their hand. To prevent instantaneous effects from the paddles, we move the ball slightly up if it is in reach of the paddles. An intervention on the bumpers is that we randomly activate a bumper with a 25% chance, while leaving others untouched and maintaining their original dynamics. Finally, an intervention on the score resets it to a random value between 0 and 4.

To render the images, we use matplotlib (Hunter, 2007) and a resolution of $64 \times 64$ pixels. The images are generated by having a single sequence of 150k images.

---

**Algorithm 1** Pseudocode of the training algorithm for the prior and graph learning in iCITRIS with NOTEARS as graph learning method. For efficiency, all for-loops are processed in parallel in the code.

---

**Require:** batch of observation samples and intervention targets: $\mathcal{B} = \{x^t, x^{t+1}, I^{t+1}\}_{n=1}^N$
1: **for** each batch element $x^t, x^{t+1}, I^{t+1}$ **do**
2:     Encode observations into latent space: $z^t = g_\theta(x^t), z^{t+1} = g_\theta(x^{t+1})$
3:     Differentiably sample one graphs $G$: $G_{ij} \sim \text{GumbelSoftmax}(1 - \sigma(\gamma_{ij}), \sigma(\gamma_{ij}))$
4:     Sample latent to causal assignments from $\psi$ for each batch element
5:     **for** each causal variable $C_i$ **do**
6:         Determine parent mask from $G$: $S \in \{0, 1\}^M, S_j = G_{\psi(j),i}$
7:         Calculate $\text{nll}_i = -\log p_\phi \left( z_{\psi_i}^{t+1} | z^t, z^{t+1} \odot S, I_i^{t+1} \right)$
8:     **end for**
9:     Backpropagation loss $\mathcal{L}^n = \sum_{i=1}^K \text{nll}_i$
10: **end for**
11: Acyclicity regularizer: $\mathcal{L}^{\text{cycle}} = \text{tr}\left(\exp(\sigma(\boldsymbol{\gamma}))\right) - K$
12: Sparsity regularizer: $\mathcal{L}^{\text{sparse}} = \frac{1}{K^2} \sum_{i=1}^K \sum_{j=1}^K \sigma(\gamma_{ij})$
13: Update parameters $\phi, \psi, \gamma$ with $\nabla_{\phi,\psi,\gamma} \left[ \lambda_{\text{cycle}} \cdot \mathcal{L}^{\text{cycle}} + \lambda_{\text{sparse}} \cdot \mathcal{L}^{\text{sparse}} + \frac{1}{N} \sum_{n=1}^N \mathcal{L}^n \right]$

---

## F  EXPERIMENTAL DETAILS

In this section, we give further details on implementation details of iCITRIS and hyperparameters that were used for the experiments in Section 5.

### F.1  ICITRIS - MODEL DETAILS

Similar to CITRIS, iCITRIS can be either implemented as a VAE or as a normalizing flow trained on the representation of a pretrained autoencoder. The core elements to implement in iCITRIS are:

- The map $g_\theta$ from observations $x^t$ to latents $z^t$ and back (iCITRIS-VAE: convolutional encoder-decoder of the VAE | iCITRIS-NF: an autoregressive normalizing flow)

- The assignment function $\psi$ of latents to causal variables (matrix of $\mathbb{R}^{M \times (K+1)}$ from which we sample via Gumbel softmax)

- The prior distributions $p_\theta$ (MLPs for conditional Gaussians)

- The continuous-optimization causal discovery method for learning the instantaneous causal graph (ENCO or NOTEARS, see below)

The first three are the same as in CITRIS, with the last being novel in iCITRIS. Thus, we discuss implementation details of this graph learning below, as well as the mutual information estimator, which is only necessary for perfect interventions and extra optimization stability. For the two graph learning methods, we additionally discuss the specific setup used to learn the prior distributions $p_\theta$.

**Graph Learning - NOTEARS** The full training algorithm of iCITRIS with the NOTEARS graph parameterization is shown in Algorithm 1. The adjacency matrix is parameterized by $\boldsymbol{\gamma} \in \mathbb{R}^{(K+1) \times (K+1)}$, where $\sigma(\gamma_{ij})$, with $\sigma$ being the sigmoid function, represents the probability of having the edge $z_{\psi_i} \rightarrow z_{\psi_j}$ in the instantaneous graph. To prevent self-loops, we set $\gamma_{ii} = -\infty, i = 0, ..., K$, and $\gamma_{i0} = -\infty, i = 1, ..., K$ to guarantee an empty instantaneous parent set for $z_{\psi_0}$. At each training iteration, we sample an adjacency matrix per batch element using the Gumbel Softmax trick (Jang et al., 2017). These matrices are used to mask out the inputs to the prior, and therefore obtain gradients by optimizing the likelihood of the prior. Further, NOTEARS requires two regularizers. First, the acyclicity regularizer takes the matrix exponential of the edge probabilities, $\sigma(\boldsymbol{\gamma})$. The trace of this matrix exponential has a minimum of $K$, which is only achieved if the matrix does not contain any cycles. All operations in this regularizer are differentiable, and we weigh this regularizer in the loss by $\lambda_{\text{cycle}}$. This weighting factor follows a scheduling over training, which starts with a value of $\exp(-6) \approx 2.5e - 3$, and reaches a maximum of $\exp(4) \approx 54.6$. In our experiments, this maximum factor ensured the graph to be approximately acyclic. Finally, the second regularizer is a sparsity regularizer, that removes redundant edges and is implemented as a L1 regularizer on the edge probabilities.

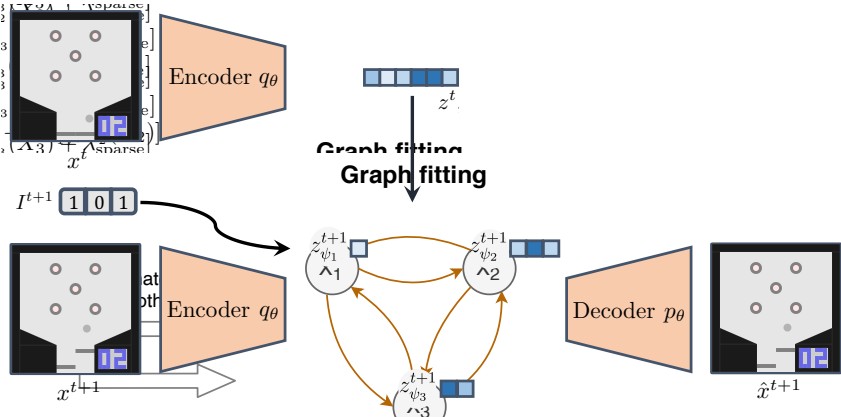

Figure 17: A visualization of iCITRIS as a VAE framework with using ENCO in its prior. Similar to CITRIS, iCITRIS uses an encoder-decoder structure to map images $x^{t+1}$ to latents $z^{t+1}$ and back. The assignment function $\psi$ splits the latent vector $z^{t+1}$ into $K$ parts (here $K = 3$), one per causal variable. Between these, we learn a causal graph with ENCO, and condition the variables additionally on the intervention targets $I^{t+1}$ according to $\psi$, and the previous time step $z^t$.

**Graph Learning - ENCO** The full training algorithm of iCITRIS with the ENCO graph parameterization is shown in Algorithm 2. The adjacency matrix is parameterized by two sets of parameters, with $\gamma \in \mathbb{R}^{(K+1)\times(K+1)}$ representing the edge existence parameters, and $\theta \in \mathbb{R}^{(K+1)\times(K+1)}$ the orientation parameters, with $\theta_{ij} = -\theta_{ji}$. The probability of an edge $z_{\psi_i} \to z_{\psi_j}$ in the instantaneous graph is determined by $\sigma(\gamma_{ij}) \cdot \sigma(\theta_{ij})$. Similar to NOTEARS, we prevent self-loops by setting $\gamma_{ii} = -\infty, i = 0, ..., K$, and fix the orientations of the edges of $z_{\psi_0}$ by setting $\theta_{i0} = -\theta_{0i} = -\infty, i = 1, ..., K$. In contrast to NOTEARS, this parameterization leads to initial edge probabilities of $0.25$. We found it beneficial to initialize the edge probabilities closer to $0.5$, which we implement by initializing $\gamma_{ij} = 4, i \neq j, i, j \in [\![1..K]\!]$ ($\sigma(4) \approx 0.98$). At each training iteration, we sample $L$ graphs from ENCO. For all experiments, we found $L = 8$ to be sufficient. For each of these graphs, we evaluate the negative log likelihood of all variables. Note that in contrast to NOTEARS, this does not need to be differentiable with respect to $\gamma$ and $\theta$. Once all graphs are evaluated, we can determine the average negative log likelihood of a $z_{\psi_j}$ under graphs with the edge $z_{\psi_i} \to z_{\psi_j}$, versus graphs where this edge was missing. We use this to determine the gradients of $\gamma_{ij}$ and $\theta_{ij}$, if $z_{\psi_j}$ has not been intervened upon. For the gradients of $\theta_{ij}$, we further mask out gradients for batch samples in which $z_{\psi_i}$ has not been intervened upon. With these gradients, we can update the graph parameters, while the distribution parameters are updated based on the differentiable negative log likelihood. Note that the sparsity regularizer, $\lambda_{\text{sparse}}$, is integrated in the update of the $\gamma$ parameters.

**Prior networks** Both graph learning algorithms use a prior network of the form $p_\phi\left(z_{\psi_i}^{t+1} | z^t, z_{\psi_i^{\text{pa}}}^{t+1}, I_i^{t+1}\right)$. To implement this efficiently in a neural network setting, we consider for each latent $z_m, m \in 1M$ a 2-layer neural network (hidden size 32 in all experiments), that take as input $z^t, z^{t+1}, I^{t+1}$, and a mask on $z^{t+1}$ and $I^{t+1}$. Therefore, its input size is $M + M + K + M + K = 3M + 2K$. The mask on $I^{t+1}$ depends on which causal variable the latent $z_m$ has been assigned to, i.e., $z_{\psi_i}^{t+1}$ should only depend on $I_i^{t+1}$. The mask on $z^{t+1}$ depends on the graph that was sampled, in combination with the causal variable assignment, i.e., only leave $z_{\psi_i^{\text{pa}}}^{t+1}$ unmasked. Further, we can use an autoregressive prior over the potentially multiple dimensions of $z_{\psi_i}^{t+1}$ by leaving previous latents unmasked that have been assigned to the causal variable $C_i$. We use this autoregressive variant for the Instantaneous Temporal Causal3DIdent and Causal Pinball dataset, since the multiple dimensions in those causal factors may not be independent.

**Mutual information estimator** The full training algorithm of the mutual information estimator is shown in Algorithm 3. The MI estimator is a 2 layer network, that takes as input the latent parents of a causal variable, and its current value, and has a single output value. This value indicates whether the current causal variable and its parents match or not, i.e., is $z_{\psi_i}^{t+1}$ the value of $C_i$ at time step $t + 1$ based on observing the parents $z^t$ and $z_{\psi_i^{\text{pa}}}^{t+1}$, or not. We train this network by a binary classification

---

**Algorithm 2** Pseudocode of the training algorithm for the prior and graph learning in iCITRIS with ENCO as graph learning method. For efficiency, all for-loops are processed in parallel in the code.

---

**Require:** batch of observation samples and intervention targets: $\mathcal{B} = \{x^t, x^{t+1}, I^{t+1}\}_{n=1}^N$
1: **for** each batch element $x^t, x^{t+1}, I^{t+1}$ **do**
2:     Encode observations into latent space: $z^t = g_\theta(x^t), z^{t+1} = g_\theta(x^{t+1})$
3:     Sample $L$ graphs $G^1, ..., G^L$ from $G_{ij}^l \sim \sigma(\theta_{ij})\sigma(\gamma_{ij})$
4:     Sample latent to causal assignments from $\psi$ for each batch element
5:     **for** each graph $G^l$ **do**
6:         **for** each causal variable $C_i$ **do**
7:             Determine parent sets for graph $G^l$: $z_{\psi_i^{\text{pa}}}^{t+1} = \{z_j^{t+1} | j \in [\![1..M]\!], \psi(j) \in \text{pa}_{G^l}(i)\}$
8:             Calculate $\text{nll}_i^l = -\log p_\phi\left(z_{\psi_i}^{t+1} | z^t, z_{\psi_i^{\text{pa}}}^{t+1}, I_i^{t+1}\right)$
9:         **end for**
10:     **end for**
11:     Backpropagation loss $\mathcal{L}^n = \frac{1}{L}\sum_{i=1}^K \sum_{l=1}^L \text{nll}_i^l$
12:     Average nll for $C_i \rightarrow / \nrightarrow C_j$: $\text{pos\_nll}_{ij}^n = \frac{\sum_{l=1}^L G_{ij}^l \text{nll}_j^l}{\sum_{l=1}^L G_{ij}^l}, \text{neg\_nll}_{ij}^n = \frac{\sum_{l=1}^L (1-G_{ij}^l)\text{nll}_j^l}{L - \sum_{l=1}^L G_{ij}^l}$
13: **end for**
14: Theta gradients: $\nabla(\theta_{ij}) = \sigma(\gamma_{ij})\sigma'(\theta_{ij})\left(\frac{1}{N}\sum_{n=1}^N I_i^n(1-I_j^n)(\text{pos\_nll}_{ij}^n - \text{neg\_nll}_{ij}^n)\right)$
15: Gamma gradients: $\nabla(\gamma_{ij}) = \sigma(\theta_{ij})\sigma'(\gamma_{ij})\left(\frac{1}{N}\sum_{n=1}^N (1-I_j^n)\left(\text{pos\_nll}_{ij}^n - \text{neg\_nll}_{ij}^n + \lambda_{\text{sparse}}\right)\right)$
16: Update theta and gamma with the gradients calculated above
17: Update distribution and assignment parameters $\phi, \psi$ with $\nabla_{\phi,\psi}\frac{1}{N}\sum_{n=1}^N \mathcal{L}^n$

---

problem, where the model compares the true set of values, *i.e.*, $z_{\psi_i}^{t+1}, z^t, z_{\psi_i^{\text{pa}}}^{t+1}$, to a randomly picked time step $\tau$, $z_{\psi_i}^{\tau+1}, z^t, z_{\psi_i^{\text{pa}}}^{t+1}$. Since the model does not have the precise time step $t$ or $\tau$ as input, it has to deduce from the values of the causal variables whether they match or not. Under interventions, we know that for the true causal variables, the optimal performance of this binary classifier is 0.5, because $C_i^{t+1}$ is independent of all its parents under perfect interventions. Thus, the gradients of the latents is to move the classifier closer to 0.5, which is equal to trying to increase the misclassification rate of the MI estimator. During training, we need to sample instantaneous graphs $G$ from our graph parameterization. Since especially in the beginning, this graph is close to random, and the true causal variables still depend on their children, for instance, under interventions, it can lead to unstable behavior to train the MI estimator on all parents from the start. Thus, instead, we initially train the MI estimator with an empty instantaneous causal graph and try to make $z_{\psi_i}^{t+1}$ independent of $z^t$, *i.e.*, its temporal parents. Over the progress of training, we introduce the instantaneous parents, similar to the graph learning scheduling, such that at the end of training, the MI estimator is fully trained on both temporal and instantaneous parents.

### F.2 HYPERPARAMETERS

We have summarized an overview of all hyperparameters in Table 4. Additionally, we discuss the main hyperparameter choices for all models here.

**Base VAE architecture** For all VAE-based methods, we have applied the same VAE to have a fair comparison between methods. In particular, we have used a VAE with a normalizing flow prior (Rezende et al., 2015), inspired by the inverse autoregressive flows (Kingma et al., 2016). The encoder outputs the parameters for $M$ independent Gaussian distributions. A sample of these Gaussians is used as input to the decoder to reconstruct the original image, but also as input to a four-layer autoregressive normalizing flow. This flow consists of a sequence of Activation Normalization (Kingma et al., 2018), Invertible $1 \times 1$ Convolutions (Kingma et al., 2018), and autoregressive affine coupling layers. The outputs of the flow are used as input to a prior, which is conditioned on the latents of the previous time step and the intervention targets. For iCITRIS, this prior follows the structure of Equation (2) including causal discovery. For CITRIS, this prior is similar to Equation (2), except that no instantaneous parents are modeled. For the iVAE, the prior is a 3-layer MLP that outputs $M$ independent Gaussian distributions. Finally, for the iVAE-AR, the prior is a 2-layer

---

**Algorithm 3** Pseudocode of the training algorithm for the mutual information estimator in iCITRIS. For efficiency, all for-loops are processed in parallel in the code.

---

**Require:** batch of observation samples and intervention targets: $\mathcal{B} = \{x^t, x^{t+1}, I^{t+1}\}_{n=1}^N$
1: Encode all observations into latent space: $z^t = g_\theta(x^t), z^{t+1} = g_\theta(x^{t+1})$
2: Sample an instantaneous graph $G$ from graph parameterization
3: Sample latent to causal assignments from $\psi$
4: **for** each causal variable $C_i$ **do**
5:     Filter out all batch elements for which $I_i^{t+1} = 0$
6:     **for** each element in the filtered batch **do**
7:         Determine parent sets for graph $G$: $z_{\psi_i^{\mathrm{pa}}}^{t+1} = \{z_j^{t+1} | j \in [\![1..M]\!], \psi(j) \in \mathrm{pa}_{G^l}(i)\}$
8:         Calculate logits of positive pairs: $e_{\mathrm{pos}} = \mathrm{NN}_{\mathrm{MI}}(z_{\psi_i}^{t+1}, z^t, z_{\psi_i^{\mathrm{pa}}}^{t+1})$
9:         For each batch element, sample a different, random time step in the batch, $\tau$
10:        Calculate logits of negative pairs: $e_{\mathrm{neg}} = \mathrm{NN}_{\mathrm{MI}}(z_{\psi_i}^{\tau+1}, z^t, z_{\psi_i^{\mathrm{pa}}}^{t+1})$
11:        Calculate loss for MI estimator: $\mathcal{L}_i^{\mathrm{NNMI}} = -e_{\mathrm{pos}} + \log\left[\exp(e_{\mathrm{pos}}) + \exp(e_{\mathrm{neg}})\right]$
12:        Calculate loss for latents: $\mathcal{L}_i^{\mathrm{zMI}} = -e_{\mathrm{neg}} + \log\left[\exp(e_{\mathrm{pos}}) + \exp(e_{\mathrm{neg}})\right]$
13:     **end for**
14: **end for**
15: Update parameters of $\mathrm{NN}_{\mathrm{MI}}$ according to avg loss $\mathcal{L}_i^{\mathrm{NNMI}}$
16: Backpropagate gradients of latents according to avg loss $\mathcal{L}_i^{\mathrm{zMI}}$

---

autoregressive NN predicting $N$ Gaussian distributions in sequence. The reconstruction loss is based on the Mean-Squared Error (MSE) objective, which provided much better results than learning a flexible distribution over the output images. The specific architecture of the encoder and decoder depends on the dataset, where we used simpler models where possible to reduce computational cost without losing significant performance. For the Voronoi benchamrk, we use a 5-layer CNN. For the Instantaneous Temporal Causal3DIdent dataset and the Causal Pinball dataset, we used a 10-layer CNN for the encoder, and a 5-layer ResNet (He et al., 2016) as decoder.

**Autoencoder + Normalizing flow architecture** For iCITRIS and CITRIS, we use the variation of training a normalizing flow on a pretrained autoencoder for the Instantaneous Temporal Causal3DIdent and Causal Pinball dataset. The autoencoder uses the same encoder and decoder architecture as the VAE, except that we increase the decoder size since it can be trained much faster than the VAE (does not require any temporal dimension), and, in contrast to the VAE, lead to improvements in the reconstruction for the two datasets. The autoencoder is trained on reconstructing the input images, where we add Gaussian noise with a small standard deviation (0.05) to the latents to simulate a distribution. Additionally, we apply a small L2 regularizer on the latents to prevent that the autoencoder counteracts the noise in the latents by artificially scaling up the standard deviation of the latents. For Causal3DIdent, we use a weight of 1e-5 on this regularizer, and 1e-6 for the Causal Pinball since its reconstructions obtain much lower losses. The normalizing flow, applied on it, follows the same architecture as in the VAE.

**Optimizer** For all models, we use the Adam optimizer (Kingma et al., 2015) with a learning rate of 1e-3. Additionally, we warmup the learning rate for the first 100 steps. Afterwards, we follow a cosine annealing learning rate scheduling, that, over the course of the training, decreases the learning rate to 5e-5.

**Frameworks** All models have been implemented and trained using PyTorch v1.10 (Paszke et al., 2019) and PyTorch Lightning v1.6.0 (Falcon et al., 2019).

## F.3 EVALUATION METRICS

For the details on the correlation matrix evaluation, we refer to Lippe et al. (2022b, Appendix C.3.1). The causal graph evaluation is performed for each model in the same way. For each model, we use the checkpoint of the best training loss, and encode all observations to the latent space. Next, we need to separate the latent space into the causal variables. For iCITRIS and CITRIS, we use the learned assignment function $\psi$ to assign latent variables to causal variables. Since the iVAE models do not learn such a latent-to-causal assignment, we instead assign each latent variable to the causal

Table 4: Summary of the hyperparameters for all models evaluated on the Voronoi benchmark, Instantaneous Temporal Causal3DIdent dataset, and the Causal Pinball dataset. For all methods, we performed a hyperparameter search over the individual, most crucial hyperparameters (e.g. KLD factor in iVAE). The smaller networks in the latter two datasets for the iVAE architectures were chosen because they require training the full encoder, decoder, and NF at the same time, and larger networks did not show any noticeable improvements. The graph learning warmup in iCITRIS is equal for all datasets, although deviations to e.g. 5k, 15k or 20k often work equally well. Further, the weighting parameters of target classifier and mutual information estimator for iCITRIS , such that, for instance, equally good results were achieved with smaller (5) or higher (20) weights on the Voronoi benchmark. For the graph sparsity regularizer, we used the same value for all graph structures of the same size.

| Voronoi benchmark | | | | | |
|---|---|---|---|---|---|
| Hyperparameter | iCITRIS-ENCO | iCITRIS-NOTEARS | CITRIS | iVAE | iVAE-AR |
| Learning rate | | —– 1e-3 —– | | | |
| Learning rate warmup | | —- 100 steps —- | | | |
| Optimizer | | —- Adam (Kingma et al., 2015) —- | | | |
| Batch size | | —- 512 —- | | | |
| Number of epochs | | —- 400 —- | | | |
| KLD Factor ($\beta$) | | —- 10 —- | | | |
| Num latents | | —- 2x number of causal variables —- | | | |
| Model variant | | —- VAE with NF prior —- | | | |
| Encoder | | —- 5 layer CNN + 2 linear layers —- | | | |
| NF-based prior | | —- 4 layer, autoregressive affine coupling (Kingma et al., 2016) —- | | | |
| Prior dependencies | | — Independent Gaussians — | | | Autoregressive |
| Decoder | | —- 5 layer (deconv-)CNN + 2 linear layers —- | | | |
| Hidden dimensionality | | —- 32 —- | | | |
| Activation function | | —- Swish (Ramachandran et al., 2017) —- | | | |
| Target classifier weight | | – 10 – | | n.a. | n.a. |
| MI weight | | – 10 – | | n.a. | n.a. |
| Graph learning warmup | | – 10k – | | n.a. | n.a. |
| Graph sparsity reg. | 0.02 (K=4,6) / 0.004 (K=9) | 0.002 (K=4,6) / 0.0004 (K=9) | n.a. | n.a. | n.a. |

| Instantaneous Temporal Causal3DIdent dataset | | | | | |
|---|---|---|---|---|---|
| Hyperparameter | iCITRIS-ENCO | iCITRIS-NOTEARS | CITRIS | iVAE | iVAE-AR |
| Learning rate | | —– 1e-3 —– | | | |
| Learning rate warmup | | —- 100 steps —- | | | |
| Optimizer | | —- Adam (Kingma et al., 2015) —- | | | |
| Batch size | | —- 512 —- | | | |
| Number of epochs | | —- 500 —- | | —- 250 —- | |
| KLD Factor ($\beta$) | | —- 1 —- | | | |
| Num latents | | —- 32 —- | | | |
| Model variant | | — AE + NF — | | – VAE with NF prior – | |
| Encoder | | —- 10-layer CNN (AE) —- | | —- 10-layer CNN (VAE) —- | |
| Num flow layers | | — 6 layers — | | — 4 layers — | |
| Prior dependencies | | — Autoregressive per $z_{\psi_i}$ — | | Independent | Autoregressive |
| Decoder | | —- 10-layer ResNet —- | | —- 5-layer ResNet —- | |
| Hidden dimensionality | | —- 64 (VAE/AE) / 32 (NF) —- | | | |
| Activation function | | —- Swish (Ramachandran et al., 2017) —- | | | |
| Target classifier weight | | – 3 – | 2 | n.a. | n.a. |
| MI weight | | – 2 – | n.a. | n.a. | n.a. |
| Graph learning warmup | | – 10k – | n.a. | n.a. | n.a. |
| Graph sparsity reg. | 0.02 | 0.0004 | n.a. | n.a. | n.a. |

| Causal Pinball dataset | | | | | |
|---|---|---|---|---|---|
| Hyperparameter | iCITRIS-ENCO | iCITRIS-NOTEARS | CITRIS | iVAE | iVAE-AR |
| Learning rate | | —- 1e-3 —- | | | |
| Learning rate warmup | | —- 100 steps —- | | | |
| Optimizer | | —- Adam (Kingma et al., 2015) —- | | | |
| Batch size | | —- 512 —- | | | |
| Number of epochs | | —- 500 —- | | —- 250 —- | |
| KLD Factor ($\beta$) | | —- 1 —- | | | |
| Num latents | | —- 24 —- | | | |
| Model variant | | — AE + NF — | | – VAE with NF prior – | |
| Encoder | | —- 10-layer CNN (AE) —- | | —- 10-layer CNN (VAE) —- | |
| Num flow layers | | — 6 layers — | | — 4 layers — | |
| Prior dependencies | | — Autoregressive per $z_{\psi_i}$ — | | Independent | Autoregressive |
| Decoder | | —- 10-layer ResNet —- | | —- 5-layer ResNet —- | |
| Hidden dimensionality | | —- 64 (VAE/AE) / 32 (NF) —- | | | |
| Activation function | | —- Swish (Ramachandran et al., 2017) —- | | | |
| Target classifier weight | | – 4 – | 2 | n.a. | n.a. |
| MI weight | | – 0 – | n.a. | n.a. | n.a. |
| Graph learning warmup | | – 10k – | n.a. | n.a. | n.a. |
| Graph sparsity reg. | 0.02 | 0.001 | n.a. | n.a. | n.a. |

factor that it has the highest correlation to. This requires using the ground truth values of the causal variables, and hence gives the iVAE an advantage over iCITRIS and CITRIS. With this separation,

we apply ENCO (Lippe et al., 2022a) to learn the temporal and instantaneous graph. Since iCITRIS already learns an instantaneous graph, we reuse the learned orientations of the model, and only relearn the edge existence parameters, $\gamma$, for potential pruning. In general, we found that the graphs predicted by iCITRIS have a few redundant edges between ancestors and descendants, which occur due to correlations in the early training iterations, and can easily be removed in this post-processing step.

As an additional metric to jointly evaluate the disentanglement of the causal variables and the learned causal graph, we use the learned causal graph and distributions by ENCO to sample new data point under novel interventional settings. For each data point in the test dataset, we use the trained model to sample the latents in the next time step, and map them back to the true causal variable space. This mapping is done by a small neural network, trained on the latents of the training dataset. To evaluate how well these samples match the interventional distributions of the true causal model, we train a small discriminator network which tries to distinguish between the true data points in the test set, and the newly generated ones from our model. Only a model that has disentangled the causal variables, *and* learned the correct causal graph, can perform well on this metric. We show the results for this metric on the Voronoi benchmark and the Instantaneous Temporal Causal3DIDent dataset in Appendix G.

Table 5: Experimental results of the large-scale study on the Voronoi benchmark with standard deviations over 5 seeds. We report the $R^2$ correlation ($R^2$ diag / $R^2$ sep), the SHD between the predicted and ground truth graph (instantaneous / temporal), and the accuracy of a discriminator distinguishing between true intervention samples and generated ones from the individual models (optimal 50%).

| Model | #variables | Random $R^2$ | Random SHD | Random Disc. | Chain $R^2$ | Chain SHD | Chain Disc. | Full $R^2$ | Full SHD | Full Disc. |
|---|---|---|---|---|---|---|---|---|---|---|
| iCITRIS-ENCO | | 0.99 / 0.00 (±0.00/±0.00) | 0.00 / 0.00 (±0.00/±0.00) | 55.71% (±1.52) | 0.99 / 0.00 (±0.01/±0.00) | 0.00 / 0.20 (±0.00/±0.45) | 55.61% (±0.25) | 0.97 / 0.00 (±0.01/±0.01) | 0.00 / 0.60 (±0.00/±0.89) | 56.25% (±1.56) |
| iCITRIS-NOTEARS | | 0.96 / 0.01 (±0.01/±0.00) | 0.00 / 0.40 (±0.00/±0.55) | 57.43% (±4.41) | 0.89 / 0.08 (±0.14/±0.12) | 1.40 / 1.60 (±1.67/±2.07) | 65.28% (±16.51) | 0.94 / 0.02 (±0.03/±0.03) | 0.00 / 3.20 (±0.00/±2.39) | 56.42% (±0.40) |
| CITRIS | 4 | 0.86 / 0.09 (±0.09/±0.08) | 1.00 / 2.60 (±1.22/±1.52) | 59.10% (±3.47) | 0.83 / 0.12 (±0.08/±0.08) | 2.00 / 4.20 (±1.58/±2.17) | 60.77% (±2.68) | 0.73 / 0.17 (±0.12/±0.11) | 2.80 / 5.80 (±0.84/±1.64) | 63.06% (±2.95) |
| iVAE | | 0.74 / 0.20 (±0.18/±0.18) | 1.00 / 4.20 (±0.71/±3.63) | 65.41% (±8.14) | 0.75 / 0.19 (±0.14/±0.16) | 2.80 / 6.40 (±1.92/±1.52) | 66.28% (±5.97) | 0.60 / 0.24 (±0.20/±0.13) | 3.60 / 6.80 (±1.34/±1.79) | 68.62% (±6.84) |
| iVAE-AR | | 0.84 / 0.21 (±0.15/±0.18) | 1.80 / 3.00 (±1.64/±2.35) | 69.85% (±6.78) | 0.92 / 0.17 (±0.04/±0.09) | 3.20 / 2.40 (±1.92/±1.67) | 70.76% (±5.06) | 0.86 / 0.32 (±0.07/±0.12) | 4.60 / 3.60 (±1.14/±1.95) | 77.60% (±12.68) |
| iCITRIS-ENCO | | 0.97 / 0.00 (±0.01/±0.00) | 0.20 / 1.00 (±0.45/±1.22) | 59.86% (±3.17) | 0.98 / 0.00 (±0.00/±0.00) | 0.00 / 0.20 (±0.00/±0.45) | 59.31% (±1.84) | 0.93 / 0.01 (±0.03/±0.01) | 0.80 / 4.20 (±1.10/±4.38) | 60.67% (±2.57) |
| iCITRIS-NOTEARS | | 0.95 / 0.01 (±0.03/±0.01) | 0.80 / 3.40 (±1.30/±3.51) | 61.48% (±3.93) | 0.96 / 0.02 (±0.03/±0.03) | 0.80 / 1.00 (±1.30/±2.24) | 65.17% (±12.64) | 0.81 / 0.07 (±0.14/±0.07) | 4.75 / 7.75 (±5.91/±4.57) | 73.34% (±13.32) |
| CITRIS | 6 | 0.80 / 0.13 (±0.04/±0.03) | 5.00 / 11.20 (±3.54/±2.59) | 65.61% (±2.48) | 0.80 / 0.15 (±0.05/±0.05) | 2.80 / 9.80 (±2.05/±2.77) | 65.54% (±2.82) | 0.70 / 0.14 (±0.03/±0.03) | 11.20 / 12.40 (±2.59/±2.70) | 67.63% (±1.18) |
| iVAE | | 0.70 / 0.22 (±0.08/±0.08) | 5.40 / 14.60 (±3.36/±4.10) | 69.99% (±3.11) | 0.70 / 0.23 (±0.04/±0.04) | 3.40 / 13.20 (±1.14/±3.19) | 69.44% (±2.76) | 0.61 / 0.18 (±0.08/±0.04) | 12.00 / 17.20 (±1.87/±2.59) | 70.14% (±3.01) |
| iVAE-AR | | 0.77 / 0.24 (±0.12/±0.11) | 8.20 / 9.20 (±1.30/±3.49) | 79.28% (±2.87) | 0.84 / 0.23 (±0.08/±0.11) | 4.60 / 8.20 (±0.89/±2.86) | 80.84% (±11.49) | 0.75 / 0.24 (±0.09/±0.08) | 12.60 / 9.00 (±0.55/±4.06) | 83.75% (±9.15) |
| iCITRIS-ENCO | | 0.96 / 0.00 (±0.01/±0.00) | 0.20 / 1.20 (±0.45/±1.10) | 63.29% (±0.99) | 0.97 / 0.00 (±0.00/±0.00) | 0.00 / 0.00 (±0.00/±0.00) | 62.91% (±1.17) | 0.89 / 0.02 (±0.03/±0.01) | 1.20 / 9.60 (±2.17/±5.86) | 65.70% (±1.32) |
| iCITRIS-NOTEARS | | 0.88 / 0.05 (±0.06/±0.04) | 1.40 / 2.40 (±2.61/±5.37) | 64.92% (±3.86) | 0.93 / 0.04 (±0.03/±0.02) | 0.40 / 0.20 (±0.89/±0.45) | 62.52% | 0.74 / 0.09 (±0.05/±0.02) | 14.50 / 9.00 (±0.71/±0.00) | 70.71% (±2.49) |
| CITRIS | 9 | 0.71 / 0.17 (±0.10/±0.08) | 14.00 / 18.40 (±4.47/±5.90) | 70.79% (±1.26) | 0.84 / 0.10 (±0.03/±0.02) | 3.00 / 13.00 (±1.22/±1.58) | 67.44% (±1.33) | 0.68 / 0.11 (±0.03/±0.02) | 32.40 / 22.40 (±1.52/±2.41) | 73.62% (±1.07) |
| iVAE | | 0.65 / 0.22 (±0.09/±0.07) | 13.60 / 24.00 (±2.88/±3.81) | 71.78% (±1.27) | 0.71 / 0.21 (±0.04/±0.03) | 4.80 / 27.60 (±1.48/±5.27) | 71.86% (±3.20) | 0.61 / 0.15 (±0.06/±0.05) | 32.40 / 29.20 (±1.14/±5.72) | 74.22% (±1.85) |
| iVAE-AR | | 0.69 / 0.22 (±0.10/±0.03) | 17.60 / 21.00 (±2.97/±7.28) | 90.25% (±8.73) | 0.80 / 0.18 (±0.12/±0.11) | 7.80 / 16.00 (±3.56/±9.25) | 79.12% (±2.51) | 0.70 / 0.21 (±0.05/±0.03) | 34.00 / 16.40 (±1.73/±4.83) | 85.85% (±0.40) |

# G  ADDITIONAL EXPERIMENTAL RESULTS AND ABLATION STUDIES

In this section, we list the detailed results of the experiments in Section 5, including the standard deviations over multiple seeds. We further provide results on the metric for predicting intervention outcomes, as described in Appendix F.3. Moreover, we present ablation studies on the Voronoi benchmark to further investigate the limitations of iCITRIS. Finally, we include a visualization of the predicted graph by all models on the Instantaneous Temporal Causal3DIdent dataset and the Causal Pinball environment.

## G.1  VORONOI BENCHMARK

The full experimental results for the Voronoi benchmark can be found in Table 5. Compared to the results in Figure 2, we also show the results of the discriminator that is trained on newly generated samples from the models. It is apparent that a crucial factor for simulating the true intervention distributions is to have low entanglement across other factors ($R^2$ sep). Both iVAE and especially iVAE-AR have a strong entanglement between factors, and show a significant gap between the true distribution and their modeled ones. For instance, on the `random` graphs of size 9, 90% of the samples can be correctly classified from iVAE-AR, indicating that the distributions do not overlap much. In comparison, iCITRIS achieves close-to optimal scores on the small graphs with $55\%$ accuracy only. Note that $50\%$ is already random performance, *i.e.*, the optimum that could be achieved. Still, with larger graphs, we see that the performance also goes down for iCITRIS, although it still outperforms all baselines.

Furthermore, to show the specific failure types of the different models on graph prediction, we additionally list the recall and precision of the graph prediction (instantaneous - Table 6, temporal - Table 7). A high recall (max 1.00) reflects that the models are able to recover all edges, while a high precision (max 1.00) shows that the model do not overpredict false positive edges. One key characteristic on all models is that they tend to have a higher recall than precision for the temporal

Table 6: Experimental results of the large-scale study on the Voronoi dataset for predicting the **instantaneous** graph, including the recall and precision to highlight false negative and positive predictions.

| Model | #variables | Graph structure | | | | | | | | |
| --- | --- | --- | --- | --- | --- | --- | --- | --- | --- | --- |
| | | | Random | | | Chain | | | Full | |
| | | SHD | recall | precision | SHD | recall | precision | SHD | recall | precision |
| iCITRIS-ENCO | 4 | 0.00 (±0.00) | 1.00 (±0.00) | 1.00 (±0.00) | 0.00 (±0.00) | 1.00 (±0.00) | 1.00 (±0.00) | 0.00 (±0.00) | 1.00 (±0.00) | 1.00 (±0.00) |
| iCITRIS-NOTEARS | | 0.00 (±0.00) | 1.00 (±0.00) | 1.00 (±0.00) | 1.40 (±1.67) | 0.73 (±0.28) | 0.80 (±0.30) | 0.00 (±0.00) | 1.00 (±0.00) | 1.00 (±0.00) |
| CITRIS | | 1.00 (±1.22) | 0.60 (±0.55) | 0.55 (±0.51) | 2.00 (±1.58) | 0.73 (±0.28) | 0.62 (±0.26) | 2.80 (±0.84) | 0.53 (±0.14) | 0.90 (±0.22) |
| iVAE | | 1.00 (±0.71) | 0.57 (±0.43) | 0.75 (±0.43) | 2.80 (±1.92) | 0.47 (±0.45) | 0.38 (±0.41) | 3.60 (±1.34) | 0.40 (±0.22) | 0.95 (±0.11) |
| iVAE-AR | | 1.80 (±1.64) | 0.33 (±0.47) | 0.40 (±0.55) | 3.20 (±1.92) | 0.40 (±0.28) | 0.58 (±0.43) | 4.60 (±1.14) | 0.23 (±0.19) | 0.42 (±0.40) |
| iCITRIS-ENCO | 6 | 0.20 (±0.45) | 0.98 (±0.04) | 1.00 (±0.00) | 0.00 (±0.00) | 1.00 (±0.00) | 1.00 (±0.00) | 0.80 (±1.10) | 0.95 (±0.07) | 1.00 (±0.00) |
| iCITRIS-NOTEARS | | 0.80 (±1.30) | 0.92 (±0.13) | 1.00 (±0.00) | 0.80 (±1.30) | 0.92 (±0.11) | 0.93 (±0.15) | 4.75 (±5.91) | 0.68 (±0.39) | 1.00 (±0.00) |
| CITRIS | | 5.00 (±3.54) | 0.48 (±0.20) | 0.70 (±0.28) | 2.80 (±2.05) | 0.64 (±0.25) | 0.80 (±0.26) | 11.20 (±2.59) | 0.25 (±0.17) | 0.95 (±0.11) |
| iVAE | | 5.40 (±3.36) | 0.36 (±0.15) | 0.70 (±0.30) | 3.40 (±1.14) | 0.56 (±0.30) | 0.66 (±0.25) | 12.00 (±1.87) | 0.20 (±0.12) | 0.85 (±0.15) |
| iVAE-AR | | 8.20 (±1.30) | 0.19 (±0.11) | 0.26 (±0.19) | 4.60 (±0.89) | 0.44 (±0.26) | 0.60 (±0.25) | 12.60 (±0.55) | 0.16 (±0.04) | 0.65 (±0.23) |
| iCITRIS-ENCO | 9 | 0.20 (±0.45) | 1.00 (±0.00) | 0.99 (±0.03) | 0.00 (±0.00) | 1.00 (±0.00) | 1.00 (±0.00) | 1.20 (±2.17) | 0.97 (±0.06) | 1.00 (±0.00) |
| iCITRIS-NOTEARS | | 1.40 (±2.61) | 0.96 (±0.06) | 0.96 (±0.10) | 0.40 (±0.89) | 1.00 (±0.00) | 0.96 (±0.09) | 14.50 (±0.71) | 0.60 (±0.02) | 1.00 (±0.00) |
| CITRIS | | 14.00 (±4.47) | 0.25 (±0.20) | 0.68 (±0.30) | 3.00 (±1.22) | 0.82 (±0.14) | 0.81 (±0.08) | 32.40 (±1.52) | 0.10 (±0.04) | 1.00 (±0.00) |
| iVAE | | 13.60 (±2.88) | 0.27 (±0.13) | 0.68 (±0.22) | 4.80 (±1.48) | 0.55 (±0.19) | 0.66 (±0.20) | 32.40 (±1.14) | 0.10 (±0.03) | 0.96 (±0.09) |
| iVAE-AR | | 17.60 (±2.97) | 0.13 (±0.08) | 0.30 (±0.20) | 7.80 (±3.56) | 0.33 (±0.14) | 0.55 (±0.34) | 34.00 (±1.73) | 0.06 (±0.05) | 0.35 (±0.15) |

Table 7: Experimental results of the large-scale study on the Voronoi dataset for predicting the **temporal** graph, including the recall and precision to highlight false negative and positive predictions.

| Model | #variables | Graph structure | | | | | | | | |
| --- | --- | --- | --- | --- | --- | --- | --- | --- | --- | --- |
| | | | Random | | | Chain | | | Full | |
| | | SHD | recall | precision | SHD | recall | precision | SHD | recall | precision |
| iCITRIS-ENCO | 4 | 0.00 (±0.00) | 1.00 (±0.00) | 1.00 (±0.00) | 0.20 (±0.45) | 1.00 (±0.00) | 0.97 (±0.06) | 0.60 (±0.89) | 1.00 (±0.00) | 0.92 (±0.11) |
| iCITRIS-NOTEARS | | 0.40 (±0.55) | 1.00 (±0.00) | 0.94 (±0.09) | 1.60 (±2.07) | 0.93 (±0.15) | 0.81 (±0.21) | 3.20 (±2.39) | 1.00 (±0.00) | 0.68 (±0.21) |
| CITRIS | | 2.60 (±1.52) | 1.00 (±0.00) | 0.70 (±0.14) | 4.20 (±2.17) | 1.00 (±0.00) | 0.60 (±0.13) | 5.80 (±1.64) | 1.00 (±0.00) | 0.51 (±0.08) |
| iVAE | | 4.20 (±3.63) | 0.87 (±0.22) | 0.63 (±0.24) | 6.40 (±1.52) | 0.85 (±0.20) | 0.48 (±0.10) | 6.80 (±1.79) | 0.63 (±0.41) | 0.39 (±0.23) |
| iVAE-AR | | 3.00 (±2.35) | 0.89 (±0.18) | 0.69 (±0.20) | 2.40 (±1.67) | 1.00 (±0.00) | 0.73 (±0.15) | 3.60 (±1.95) | 1.00 (±0.00) | 0.64 (±0.15) |
| iCITRIS-ENCO | 6 | 1.00 (±1.22) | 1.00 (±0.00) | 0.92 (±0.10) | 0.20 (±0.45) | 1.00 (±0.00) | 0.98 (±0.04) | 4.20 (±4.38) | 0.98 (±0.04) | 0.75 (±0.19) |
| iCITRIS-NOTEARS | | 3.40 (±3.51) | 1.00 (±0.00) | 0.79 (±0.18) | 1.00 (±2.24) | 1.00 (±0.00) | 0.93 (±0.15) | 7.75 (±4.57) | 1.00 (±0.00) | 0.59 (±0.21) |
| CITRIS | | 11.20 (±2.59) | 0.94 (±0.06) | 0.50 (±0.04) | 9.80 (±2.77) | 0.92 (±0.17) | 0.52 (±0.12) | 12.40 (±2.70) | 0.98 (±0.04) | 0.45 (±0.11) |
| iVAE | | 14.60 (±4.10) | 0.79 (±0.21) | 0.42 (±0.09) | 13.20 (±3.19) | 0.91 (±0.13) | 0.44 (±0.13) | 17.20 (±2.59) | 0.87 (±0.19) | 0.36 (±0.07) |
| iVAE-AR | | 9.20 (±3.49) | 0.85 (±0.21) | 0.55 (±0.12) | 8.20 (±2.86) | 0.98 (±0.03) | 0.56 (±0.09) | 9.00 (±4.06) | 0.90 (±0.18) | 0.54 (±0.10) |
| iCITRIS-ENCO | 9 | 1.20 (±1.10) | 1.00 (±0.00) | 0.94 (±0.06) | 0.00 (±0.00) | 1.00 (±0.00) | 1.00 (±0.00) | 9.60 (±5.86) | 1.00 (±0.00) | 0.71 (±0.15) |
| iCITRIS-NOTEARS | | 2.40 (±5.37) | 1.00 (±0.00) | 0.92 (±0.19) | 0.20 (±0.45) | 1.00 (±0.00) | 0.99 (±0.02) | 9.00 (±0.00) | 0.95 (±0.00) | 0.72 (±0.02) |
| CITRIS | | 18.40 (±5.90) | 0.84 (±0.15) | 0.55 (±0.09) | 13.00 (±1.58) | 1.00 (±0.00) | 0.62 (±0.04) | 22.40 (±2.41) | 0.92 (±0.08) | 0.48 (±0.03) |
| iVAE | | 24.00 (±3.81) | 0.82 (±0.15) | 0.48 (±0.08) | 27.60 (±5.27) | 0.99 (±0.02) | 0.44 (±0.07) | 29.20 (±5.72) | 0.86 (±0.05) | 0.41 (±0.05) |
| iVAE-AR | | 21.00 (±7.28) | 0.89 (±0.17) | 0.51 (±0.04) | 16.00 (±9.25) | 0.85 (±0.18) | 0.60 (±0.17) | 16.40 (±4.83) | 0.91 (±0.06) | 0.57 (±0.07) |

graph. In other words, many mistakes are due to predicting too many edges, which easily occurs when causal variables are entangled. However, on the instantaneous graphs, we clearly see that the baselines, CITRIS and iVAE, predict a sparse graph by having a low recall. This also underlines that

Table 8: Results of three ablation studies on the Voronoi benchmark, performed on the `random` graph with 6 variables. **Left (Intervention noise)**: We introduce noise on the intervention targets by introducing 10% false positive cases in the intervention targets, *i.e.*, $I_i^t = 1$ although $C_i$ is sampled from the observation distribution. iCITRIS-ENCO performs almost as well as before. **Middle (No instantaneous)**: We apply all methods on graphs with an empty instantaneous graph. iCITRIS-ENCO obtains almost perfect disentanglement along with CITRIS and iVAE, showing that iCITRIS can be used as a replacement of them under perfect interventions. **Right (No temporal)**: The most difficult setup is when the variables have purely instantaneous relations and samples between time steps are independent. In this case, no method can disentangle the variables well, showing that different optimization strategies than iCITRIS are needed in this setting.

| | **Ablation study** | | | | | |
| **Model** | Intervention noise | | No instantaneous | | No temporal | |
| | $R^2$ | SHD | $R^2$ | SHD | $R^2$ | SHD |
| iCITRIS-ENCO | 0.96 / 0.00 | 0.20 / 4.00 | 0.99 / 0.00 | 0.00 / 0.00 | 0.55 / 0.21 | 6.80 / 0.00 |
| | $(\pm 0.01)/(\pm 0.00)$ | $(\pm 0.45)/(\pm 1.87)$ | $(\pm 0.00)/(\pm 0.00)$ | $(\pm 0.00)/(\pm 0.00)$ | $(\pm 0.08)/(\pm 0.06)$ | $(\pm 2.39)/(\pm 0.00)$ |
| CITRIS | 0.78 / 0.14 | 5.00 / 10.60 | 0.99 / 0.00 | 0.00 / 0.00 | 0.54 / 0.21 | 6.80 / 0.00 |
| | $(\pm 0.09)/(\pm 0.09)$ | $(\pm 4.58)/(\pm 4.51)$ | $(\pm 0.00)/(\pm 0.00)$ | $(\pm 0.00)/(\pm 0.00)$ | $(\pm 0.06)/(\pm 0.06)$ | $(\pm 2.39)/(\pm 0.00)$ |
| iVAE | 0.69 / 0.22 | 5.40 / 15.00 | 0.99 / 0.00 | 0.00 / 0.00 | 0.49 / 0.20 | 7.00 / 0.00 |
| | $(\pm 0.10)/(\pm 0.11)$ | $(\pm 3.36)/(\pm 3.39)$ | $(\pm 0.00)/(\pm 0.00)$ | $(\pm 0.00)/(\pm 0.00)$ | $(\pm 0.05)/(\pm 0.02)$ | $(\pm 2.24)/(\pm 0.00)$ |
| iVAE-AR | 0.84 / 0.22 | 7.40 / 8.60 | 0.79 / 0.23 | 2.80 / 9.00 | 0.54 / 0.28 | 7.80 / 0.00 |
| | $(\pm 0.10)/(\pm 0.06)$ | $(\pm 2.07)/(\pm 3.44)$ | $(\pm 0.13)/(\pm 0.06)$ | $(\pm 0.45)/(\pm 4.42)$ | $(\pm 0.05)/(\pm 0.02)$ | $(\pm 3.27)/(\pm 0.00)$ |

the false positive edges in the temporal graph cannot be simply removed by increasing the sparsity regularizer in the causal discovery method, since otherwise, even more edges would be lost in the instantaneous graph. iVAE-AR, on the other hand, has a low precision *and* recall on the instantaneous graphs; showcasing that it predicts a very different graph with anticausal edges. Meanwhile, only iCITRIS-ENCO obtains a higher recall and precision across the different graph structures and sizes.

Next, we look at ablation studies that investigate the applications and limitations of iCITRIS.

### G.1.1 ABLATION 1: NOISY INTERVENTION TARGETS

In the first ablation study, we focus on the dependency of iCITRIS on accurate intervention targets. In practice, performing perfect interventions is a difficult task, and is prone to noise. While we can easily observe whether we pushed a button or did external actions to influence a dynamical system, we do not know for sure whether the intervention succeeded or not. This corresponds to a case where the intervention targets, $I^t$, are noisy and tend to have false positives, *i.e.*, $I_i^t = 1$ although the intervention did not succeed. How sensitive is iCITRIS to such noise?

To investigate this question, we repeat the experiments of the Voronoi benchmark on the `random` graphs of size 6, but simulate that in 10% of the cases when $I_i^t$, we actually do not intervene on $C_i$ and instead sample the value from its observational distribution. The results are summarized in Table 8 (left two columns), and clearly show that iCITRIS can yet work well in this setting. The variables are almost as well as before disentangled as before. The additional temporal variables are partially also because of noisy interventions in the post-processing causal discovery setting.

### G.1.2 ABLATION 2: EMPTY INSTANTANEOUS GRAPH

The main aspect of iCITRIS in contrast to the baselines is that supports instantaneous effects. However, in practice, we might not know whether instantaneous effects are in the data or not. Thus, this ablation study investigates, whether iCITRIS can yet be used as a replacement of the baselines like CITRIS, when perfect interventions are provided. For this, we repeat the experiments of the Voronoi benchmark on causal models with an empty instantaneous of size 6. As the results in Table 8 show, iCITRIS, CITRIS, and iVAE all are able to identify the causal variables and the graph. This show that iCITRIS can indeed be used as a replacement of CITRIS and iVAE, even in the setting that the variables are independent, conditioned on the previous time step.

Table 9: Experimental results on the Instantaneous Temporal Causal3D dataset, with standard deviations across three seeds.

| Model | $R^2$ | Spearman | Triplets | SHD (Instant) | SHD (Temp) |
|---|---|---|---|---|---|
| iCITRIS-ENCO | 0.96 / 0.07 | 0.96 / 0.12 | 0.11 | 1.67 | 5.67 |
| | $(\pm 0.00)/(\pm 0.01)$ | $(\pm 0.00)/(\pm 0.01)$ | $(\pm 0.01)$ | $(\pm 0.58)$ | $(\pm 1.15)$ |
| iCITRIS-NOTEARS | 0.95 / 0.10 | 0.95 / 0.14 | 0.15 | 4.33 | 6.33 |
| | $(\pm 0.01)/(\pm 0.02)$ | $(\pm 0.01)/(\pm 0.01)$ | $(\pm 0.02)$ | $(\pm 1.15)$ | $(\pm 1.15)$ |
| CITRIS | 0.90 / 0.23 | 0.89 / 0.26 | 0.20 | 5.67 | 12.67 |
| | $(\pm 0.01)/(\pm 0.02)$ | $(\pm 0.01)/(\pm 0.02)$ | $(\pm 0.01)$ | $(\pm 0.58)$ | $(\pm 0.58)$ |
| iVAE | 0.79 / 0.24 | 0.76 / 0.24 | 0.27 | 6.00 | 15.00 |
| | $(\pm 0.02)/(\pm 0.04)$ | $(\pm 0.03)/(\pm 0.02)$ | $(\pm 0.00)$ | $(\pm 1.00)$ | $(\pm 1.00)$ |
| iVAE-AR | 0.74 / 0.29 | 0.72 / 0.36 | 0.31 | 10.67 | 12.33 |
| | $(\pm 0.02)/(\pm 0.03)$ | $(\pm 0.04)/(\pm 0.06)$ | $(\pm 0.00)$ | $(\pm 0.58)$ | $(\pm 4.51)$ |

### G.1.3 ABLATION 3: EMPTY TEMPORAL GRAPH

As a final ablation study, we consider the most difficult setup, namely having no temporal relations at all. In this case, all relations between causal variables are purely instantaneous, and we cannot use any information of the previous time step, *i.e.*, $z^t$, as an initial guidance for disentangling the variables. Once more, we repeat the experiments of the Voronoi benchmark on the `random` graphs of size 6, but with an empty instantaneous graph, and summarize the results in Table 8 (right columns). Due to the difficulty of the task, none of the methods was able to identify the causal variables. Since the probabilities of the edges in iCITRIS are initially around 0.5, the model focuses on finding $K$ independent factors of variations instead of the causal variables. The balance that is crucial for the temporal setup is that the knowledge of the interventions and previous time step is more important than the instantaneous effects for some variables, which, in this case, does not hold. Hence, to overcome this problem, different optimization strategies than the ones discussed in Section 4.3 are needed. Interestingly, even the autoregressive iVAE fails at going beyond finding $K$ independent factors, underlining the difficulty of the task.

## G.2 INSTANTANEOUS TEMPORAL CAUSAL3DIDENT

We report the full experimental results for the Instantaneous Temporal Causal3DIdent, including standard deviations across three seeds for all models, are shown in Table 9. Next to the correlation and graph prediction metrics, we also list the results of the triplet evaluation, following Lippe et al. (2022b). The triplet distance measures how well we can perform combinations of causal factors in latent space without causing correlations among different factors.

### G.2.1 ABLATION STUDY 4: ORIGINAL TEMPORAL CAUSAL3DIDENT WITHOUT INSTANTANEOUS EFFECT

To verify that iCITRIS can be used as a replacement to CITRIS, even in environments where no instantaneous effects are present, we apply iCITRIS to the original Temporal Causal3DIdent dataset. This dataset has the same causal variables and relations, but instead of instantaneous effects, all effects are over temporal time steps. The results are shown in Table 10. iCITRIS achieves almost identical results to CITRIS, verifying that iCITRIS generalizes CITRIS. In terms of the causal graph prediction, iCITRIS occasionally predicted an instantaneous edge between the object shape and the object rotation; an edge that CITRIS incorrectly predicted over time. This is to be expected due to the visual complexity of the dataset.

### G.2.2 ABLATION STUDY 5: PERFECT INTERVENTIONS IN INSTANTANEOUS TEMPORAL CAUSAL3DIDENT

As an ablation to the shown dataset, we conduct an experiment on the Causal3DIdent dataset, where all interventions are perfect. The experimental results for this dataset, including standard deviations across three seeds for all models, are shown in Table 11. We additionally show the discriminator

Table 10: Experimental results on the original Temporal Causal3DIdent dataset. Results for CITRIS and iVAE are taken from Lippe et al. (2022b). iCITRIS performs on par with CITRIS, despite not assuming an empty instantaneous causal graph.

| Model | $R^2$ | Spearman | Triplets |
|-------|-------|----------|----------|
| iCITRIS-ENCO | 0.97 / 0.04 | 0.97 / 0.09 | 0.08 |
| CITRIS | 0.98 / 0.04 | 0.97 / 0.08 | 0.07 |
| iVAE | 0.80 / 0.29 | 0.77 / 0.28 | 0.27 |

Table 11: Experimental results on the Instantaneous Temporal Causal3D dataset, with standard deviations across three seeds. The results for iCITRIS-ENCO are shown with and without using the Mutual Information Estimator.

| Model | $R^2$ | Spearman | Triplets | SHD (Instant) | SHD (Temp) | Disc. Acc. |
|-------|-------|----------|----------|---------------|------------|------------|
| iCITRIS-ENCO | 0.96 / 0.05 | 0.96 / 0.10 | 0.09 | 1.33 | 5.00 | 55.41% |
| | ($\pm$0.00)/($\pm$0.00) | ($\pm$0.00)/($\pm$0.00) | ($\pm$0.00) | ($\pm$1.15) | ($\pm$1.73) | ($\pm$0.87%) |
| iCITRIS-ENCO - No MI | 0.96 / 0.15 | 0.95 / 0.21 | 0.13 | 3.00 | 8.33 | 60.85% |
| | ($\pm$0.00)/($\pm$0.00) | ($\pm$0.01)/($\pm$0.01) | ($\pm$0.00) | ($\pm$1.00) | ($\pm$2.00) | ($\pm$2.34%) |
| iCITRIS-NOTEARS | 0.95 / 0.09 | 0.95 / 0.14 | 0.14 | 4.00 | 5.00 | 61.05% |
| | ($\pm$0.00)/($\pm$0.01) | ($\pm$0.01)/($\pm$0.01) | ($\pm$0.00) | ($\pm$1.00) | ($\pm$1.73) | ($\pm$4.43%) |
| CITRIS | 0.92 / 0.19 | 0.90 / 0.22 | 0.19 | 4.67 | 10.00 | 67.51% |
| | ($\pm$0.01)/($\pm$0.02) | ($\pm$0.01)/($\pm$0.01) | ($\pm$0.01) | ($\pm$0.58) | ($\pm$2.00) | ($\pm$5.90%) |
| iVAE | 0.82 / 0.20 | 0.80 / 0.22 | 0.27 | 6.67 | 15.33 | 86.87% |
| | ($\pm$0.02)/($\pm$0.01) | ($\pm$0.02)/($\pm$0.01) | ($\pm$0.01) | ($\pm$2.52) | ($\pm$1.53) | ($\pm$2.71%) |
| iVAE-AR | 0.79 / 0.29 | 0.78 / 0.33 | 0.27 | 11.00 | 12.67 | 87.07% |
| | ($\pm$0.01)/($\pm$0.03) | ($\pm$0.02)/($\pm$0.00) | ($\pm$0.01) | ($\pm$1.00) | ($\pm$1.53) | ($\pm$0.90%) |

accuracy of distinguishing between true and fake interventional samples. The results indicate that while this makes the task in general a bit easier, iCITRIS-ENCO still performs the best. The small differences in entanglement between iCITRIS-ENCO and iCITRIS-NOTEARS lead to considerable difference for generating new combinations of causal factors, highlighting the importance of strong disentanglement between causal factors. Similarly, the discriminator accuracy shows that iCITRIS-ENCO can accurately model the distribution of the true causal model, while clear differences to the VAE-based baseline, iVAE and iVAE-AR, are visible.

To get an intuition on what graphs the different models identify, we have visualized on example of each model in Figure 18. In general, we see that iCITRIS-ENCO misses only one edge which is sparse anyway, since hue_b affects hue_o only for two shapes. Similarly to the results of Lippe et al. (2022b), we find that the object shape is a false positive parent of the rotation of the object. For CITRIS, we see that it starts to predict incorrect orientations due to correlations among factors. Finally, iVAE and iVAE-AR predict graphs that have little in common with the ground truth one.

To show the importance of the mutual information estimator in iCITRIS, we experiment with iCITRIS but without the MI estimator. The results in Table 11 show that the MI estimator is indeed a crucial performance to reach iCITRIS's strong performance. Without the MI estimator, we experience higher correlations between different latent representations and causal variables. In the end, this also leads to a worsened graph estimation for both instantaneous and temporal effects.

### G.3 CAUSAL PINBALL

Finally, the full experimental results for the Causal Pinball environment can be found in Table 12. Besides the correlation and graph metrics, we again report the triplet evaluation, which shows once more that iCITRIS and CITRIS both work well here.

### G.3.1 ABLATION STUDY 6: PERFECT INTERVENTIONS IN CAUSAL PINBALL

As an ablation study, we simulate a Causal Pinball environment where all partially-perfect interventions are replaced by perfect interventions. With this, we can make use of the Mutual Information es-

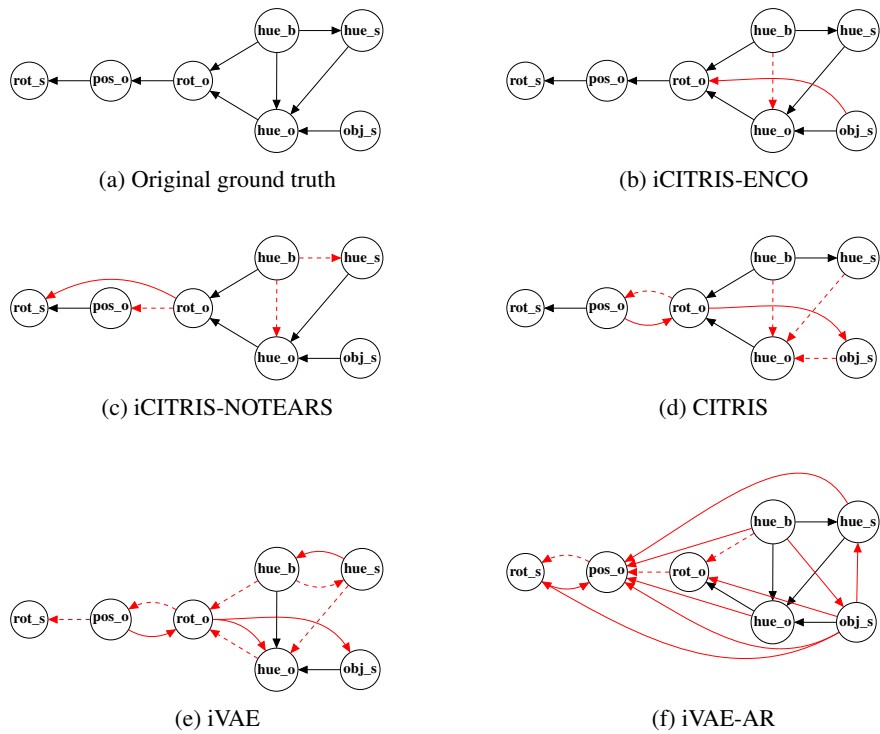

Figure 18: Learned instantaneous graphs in the Instantaneous Temporal Causal3DIdent dataset for all five models for a single seed. Red arrows indicate false positive edges, and dashed red arrows false negatives. (a) The ground truth of the dataset. (b) iCITRIS-ENCO achieves for one score a perfect recovery of the graph, and for the other two graphs, we miss one edge to hue_o since hue_b only affects it for certain object shapes, and have an additional from the object shape to the rotation due to the complexity of the problem. (c) iCITRIS-NOTEARS had more false positive and negative edges than iCITRIS-ENCO. However, all orientations were correct. (d) CITRIS had in general a sparser graph than the true graph, but in contrast to iCITRIS, also obtained wrong orientations several times (e.g. between pos_o and rot_o). (e) The iVAE obtains very different graphs from the ground truth, with many incorrectly edges. (f) Due to the autoregressive prior in iVAE-AR, we observed a significant amount of false positive edges, with occasional incorrect orientation as well.

Table 12: Experimental results on the Causal Pinball dataset over three seeds.

| Model | $R^2$ | Spearman | Triplets | SHD (Instant) | SHD (Temp) |
|---|---|---|---|---|---|
| iCITRIS-ENCO | **0.99** / 0.12 | **0.99** / 0.25 | **0.03** | **0.67** | **3.00** |
| | $(\pm 0.00)/(\pm 0.01)$ | $(\pm 0.00)/(\pm 0.02)$ | $(\pm 0.01)$ | $(\pm 1.15)$ | $(\pm 0.71)$ |
| iCITRIS-NOTEARS | 0.98 / 0.18 | 0.99 / 0.38 | 0.18 | 3.33 | 4.67 |
| | $(\pm 0.00)/(\pm 0.01)$ | $(\pm 0.00)/(\pm 0.03)$ | $(\pm 0.03)$ | $(\pm 1.52)$ | $(\pm 0.58)$ |
| CITRIS | 0.90 / 0.39 | 0.95 / 0.50 | 0.11 | 3.00 | 7.67 |
| | $(\pm 0.06)/(\pm 0.07)$ | $(\pm 0.01)/(\pm 0.08)$ | $(\pm 0.01)$ | $(\pm 1.00)$ | $(\pm 2.31)$ |
| iVAE | 0.44 / **0.05** | 0.47 / **0.05** | 0.61 | 4.33 | 4.67 |
| | $(\pm 0.09)/(\pm 0.00)$ | $(\pm 0.10)/(\pm 0.01)$ | $(\pm 0.04)$ | $(\pm 0.58)$ | $(\pm 0.58)$ |
| iVAE-AR | 0.47 / 0.15 | 0.52 / 0.40 | 0.63 | 8.00 | 3.67 |
| | $(\pm 0.10)/(\pm 0.04)$ | $(\pm 0.11)/(\pm 0.16)$ | $(\pm 0.04)$ | $(\pm 2.00)$ | $(\pm 1.53)$ |

timator in iCITRIS. The results on this dataset are shown in Table 13. Besides identifying the causal variables well, iCITRIS-ENCO identifies the instantaneous causal graph with minor errors. Interestingly, CITRIS obtains a good correlation score as well here. This is likely due to the instantaneous effects being very sparse, and perfect interventions giving a very strong preference towards independent variables in this case. Yet, there is still a gap between iCITRIS-ENCO and CITRIS in the instantaneous SHD, showing the benefit of learning the instantaneous graph jointly with the causal variables.

Table 13: Perfect interventions in the Causal Pinball dataset. Experimental results are shown over three seeds.

| Model | $R^2$ | Spearman | Triplets | SHD (Instant) | SHD (Temp) |
|---|---|---|---|---|---|
| iCITRIS-ENCO | 0.98 / 0.04 | 0.99 / 0.17 | 0.02 | 0.67 | 3.67 |
| | $(\pm 0.00)/(\pm 0.01)$ | $(\pm 0.00)/(\pm 0.03)$ | $(\pm 0.00)$ | $(\pm 0.58)$ | $(\pm 1.15)$ |
| iCITRIS-NOTEARS | 0.98 / 0.06 | 0.99 / 0.19 | 0.02 | 2.33 | 3.67 |
| | $(\pm 0.00)/(\pm 0.04)$ | $(\pm 0.00)/(\pm 0.06)$ | $(\pm 0.00)$ | $(\pm 0.58)$ | $(\pm 0.58)$ |
| CITRIS | 0.98 / 0.04 | 0.99 / 0.18 | 0.02 | 2.67 | 4.00 |
| | $(\pm 0.01)/(\pm 0.01)$ | $(\pm 0.00)/(\pm 0.02)$ | $(\pm 0.00)$ | $(\pm 1.53)$ | $(\pm 1.00)$ |
| iVAE | 0.55 / 0.04 | 0.58 / 0.14 | 0.55 | 2.33 | 4.33 |
| | $(\pm 0.08)/(\pm 0.03)$ | $(\pm 0.09)/(\pm 0.06)$ | $(\pm 0.06)$ | $(\pm 0.58)$ | $(\pm 1.15)$ |
| iVAE-AR | 0.53 / 0.15 | 0.55 / 0.30 | 0.56 | 4.33 | 6.33 |
| | $(\pm 0.08)/(\pm 0.09)$ | $(\pm 0.09)/(\pm 0.08)$ | $(\pm 0.06)$ | $(\pm 1.53)$ | $(\pm 1.53)$ |

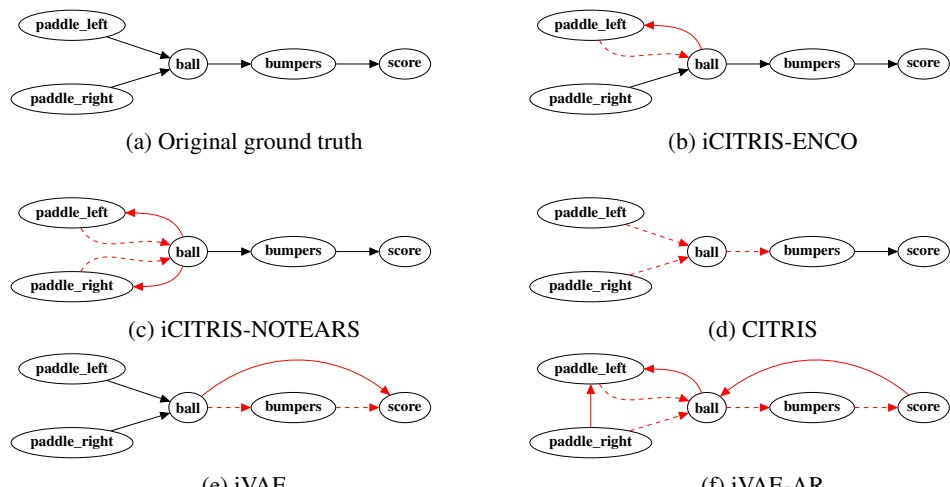

(a) Original ground truth         (b) iCITRIS-ENCO

(c) iCITRIS-NOTEARS         (d) CITRIS

(e) iVAE         (f) iVAE-AR

Figure 19: Learned instantaneous graphs in the Causal Pinball dataset for all five models for a single seed. Red arrows indicate false positive edges, and dashed red arrows false negatives. (a) The ground truth of the dataset. (b) iCITRIS-ENCO recovered the graph for one seed perfectly, and for the other two seeds, incorrectly oriented an edge between the ball and paddles. (c) iCITRIS-NOTEARS commonly has some incorrect orientations between the paddles and the ball. (d) CITRIS, similar to other experiments, tends to have a sparser instantaneous graph. (e) iVAE has a sparser graph, similar to CITRIS, but with additional false positive edges. (f) iVAE-AR predicts a causal graph that has no edge in common with the true graph.

Further, we visualize the predicted causal graphs of the different methods. In general, we found that the most difficult relations are between the paddles and the ball, in particular their orientation. This is due to the deterministic relations between the two factors, such that if the ball has been hit by the paddle, we can already predict it just from the ball position. Further, in many states, the ball and paddle do not affect each other, such that a state where the paddle would have hit the ball, but the ball was intervened upon in the same time step, is extremely rare. Overall, all models suffered from this problem, but iCITRIS showed to handle it.

