# OpenReview forum: "Causal Representation Learning for Instantaneous and Temporal Effects in Interactive Systems"
_ICLR.cc/2023/Conference — ICLR 2023 poster_

### Official Review · Reviewer_1H3n · 2022-10-22

**Confidence:** 4
**Correctness:** 4
**Technical Novelty And Significance:** 3
**Empirical Novelty And Significance:** 3
**Recommendation:** 8

**Clarity, Quality, Novelty And Reproducibility:**

 - This is high quality original work.
 - I would have a really hard time reproducing this from the text alone. That may be because the method is just complex, but I think that the author could probably do a better job of illustrating the various piece that need to be implemented - a good figure would go a long way. They do supply code though.
 - I thought the Preliminaries (Appendix C1) section in the appendix was actually a really nice summary of the method - I found only really understood what you were doing once I read that. I know space is constrained, but augmenting figure 1 to include as much as possible of that information would be helpful to help keep track of all the moving pieces in this.

**Strength And Weaknesses:**

Strengths
 - Instantaneous effects are an important problem to address if we are going to make causal representation learning practical, so I like the clear articulation of the setting and the problems that arise. The light / light switch example is a really nice simple illustrative example, but you could probably strengthen this motivation by finding more settings where this arrises: e.g. it is common in natural science applications where you can only measure a system infrequently (very common in biology); you want to argue that this is very common in practice (I think this is likely), and not just a feature of things that occur at the speed of light.
 - There’s a really nice discussion of the assumptions in the appendix. Space is always an issue, but as much as possible of this should be moved into the text, because it does a better job than the main text of articulating what we’re ruling out with the assumptions.

Weaknesses:
 - The assumption that we know the intervention target is still very strong. This was true of CITRIS and is true here too. I don’t think it’s grounds for rejection, but I do think it severely limits what we can apply this method to.
 - Assumption 5’s symmetry breaking condition seems tricky to verify. The appendix articulates a nice example of a failure case, and gives sufficient conditions for gaussian (interventions need to modify means) - do you have sufficient conditions for any other distributions? You say, “this assumption will likely hold in most practical scenarios” - is that because most practical scenario involve Gaussians? Or some other reason?
 - The assumption that there is a bijection between X and both C and E is strong given the critique of methods that rely on counterfactual information (Locatello et al 2020, Brehmer et al 2022 and Ahjua et al 2022). It implies that the challenges of counterfactuals are reduced to “just” solving disentanglement (of course this isn’t an easy task), because we succeed in disentangling, there is no uncertainty about the noise variables in the abduction step, so counterfactuals are known exactly. I don’t mind this assumption in this context — it’s just saying that we see all the latents rendered visually — I’m having had time deciding whether it really is weaker than the perturbation assumptions.
 - A lot of the detail about the actual implementation of the approach is hidden in a very long appendix - I know it’s just maximum likelihood wrt to the appropriate interventional distributions, but we need a little more detail in the main text to know how that translates to a loss function (e.g. are the distriubiutions over latents conditionally gaussian? Something else? ) and how the various components of the method are parameterized.
 - I think a running example would really help. There are a lot of moving pieces, so if you could have an example to keep referencing, it’d keep track of all the variables and relationships. E.g. C = [light state, switch state, …], X = pixel level image of the light scene, G = … and so on. If you just look at how many variables show up in Definition 4.3, you can see that it’s hard for a reader to track all the pieces.
 - Minimal causal variables were not well explained. I know this is a concept from Lippe et al 2022, but I would have liked to see an explanation here too, in order to make this paper self-contained.


**Summary Of The Paper:**

This paper describes iCITRIS, a generalization of CITRIS [Lippe et al 2022] to allow for “instantaneous effects”: i.e. they allow dependencies between the latent variables within a given time step, as opposed to enforcing conditional independence of the latents at time t given the latents at time t-1. They start by showing that it is impossible to identify latents using only the CITRIS assumptions (known intervention targets & time series observations), because we can’t distinguish between independent latents $P(z_1, z_2) = P(z_1)P(z_2)$ and entangled non-independent latents that have the same joint. However, if you have access to “partially perfect” interventions that break the instantaneous dependencies, then you can recover the same identifiability results as those provided in CITRIS. They support their theory with a series of nice empirical demonstrations that show strong disentanglement on data generating processes with slightly weaker assumptions than their theory.

**Summary Of The Review:**

This is a nice paper that deals with an important problem and has strong experimental results. My main issues are with presentation - there are a lot of elements and the presentation could help the reader keep track of them all better.

---

> ### Author Response · Authors · 2022-11-12
> **Author Response to Reviewer 1H3n [1/3]**
>
> Thank you for your positive feedback and helpful suggestion, which we have used to further improve the paper. Please see our answer below.
>
> **The assumption that we know the intervention target is still very strong. This was true of CITRIS and is true here too. I don’t think it’s grounds for rejection, but I do think it severely limits what we can apply this method to.**
>
> We agree that knowing the intervention target is a limitation of iCITRIS and may not be possible in all environments. The most suitable environments we currently see are interactive settings, where an agent can observe its own actions, and with small prior knowledge, map those actions to interventions on unknown causal factors. Nonetheless, iCITRIS is still one of the first methods to model instantaneous effects in causal representation learning, and requirements of other methods such as access to counterfactuals or supervised labels are often even harder to obtain. Further, we do believe that extending CITRIS to settings of unknown intervention targets is an important future work, and orthogonal to our generalization to instantaneous effects. Furthermore, our insights to instantaneous effects, e.g., the need for partially-perfect interventions and the non-identifiability otherwise, can be extended to other causal representation approaches, providing a larger corpus of possible future work.
>
> **Assumption 5’s symmetry breaking condition seems tricky to verify. The appendix articulates a nice example of a failure case, and gives sufficient conditions for gaussian (interventions need to modify means) - do you have sufficient conditions for any other distributions? You say, “this assumption will likely hold in most practical scenarios” - is that because most practical scenario involve Gaussians? Or some other reason?**
>
> The sufficient conditions for Gaussians actually apply to a much larger class of functions, and we have used Gaussians just as an example. All continuous distributions that only have symmetries around their mean or another fixed point fulfill the symmetry breaking condition if their mean/fixed point depends on interventions and/or temporal parents. This includes distributions like the Logistic, Laplace, Cauchy (which does not have a mean but symmetry around its mode), and more, if their parameter $\mu$/$x_0$ is influenced by the interventions/temporal parents.
>
> These distributions, including Gaussians, are to our knowledge among the most common distributions to model noise in physical dynamical systems. Furthermore, since in most time series, the value of a causal variable will depend on its previous value (e.g. previous position, velocity, state of the light bulb, etc.), its mean will similarly depend on its previous values and/or interventions. Thus, the symmetry breaking assumption is fulfilled for such cases.

---

> > ### Author Response · Authors · 2022-11-12
> > **Author Response to Reviewer 1H3n [2/3]**
> >
> > **The assumption that there is a bijection between X and both C and E is strong given the critique of methods that rely on counterfactual information (Locatello et al 2020, Brehmer et al 2022 and Ahjua et al 2022). It implies that the challenges of counterfactuals are reduced to “just” solving disentanglement (of course this isn’t an easy task), because we succeed in disentangling, there is no uncertainty about the noise variables in the abduction step, so counterfactuals are known exactly. I don’t mind this assumption in this context — it’s just saying that we see all the latents rendered visually — I’m having had time deciding whether it really is weaker than the perturbation assumptions.**
> >
> > We clarify that $E$ in our setup represent *observation* noise variables that may influence the high-dimensional observation, e.g. image, but are *not* related to the noise variables in the SCM of the individual causal variables or influence them in any way. The variable $E$ is only used to allow for extra noise in the observations, and can be the empty set for noise-free observations. In other words, in Causal Pinball, the observation noise variables $E$ could model observation effects like global color changes or minor camera offsets, which however do not affect the underlying causal variables. Hence, the observation $X$ only shows the values of the causal variables, not the counterfactual noise. In turn, identifying the causal representation in iTRIS does not identify counterfactuals of the causal variables, since the counterfactual noise remains unknown. We have clarified the explanation of $E$ in Section 3.1.
> >
> > If we consider $E=\emptyset$, i.e., a simplified version without any observation noise, the bijection between $X$ (observation) and $C$ (all causal variables) implies that the values of all causal variables can be identified from the images. This assumption of bijection between the values of the causal variables and the observation image is the same as in Brehmer et al. [2022] (Definition 1) and Locatello et al. [2020] (Theorem 1), and follows closely to Ahuja et al. [2022] by their injectiveness assumption. Without this assumption, further challenges such as occlusion or noisy observations may need to be tackled, which we consider orthogonal to our discussion on instantaneous effects.
> >
> > ### Presentation
> >
> > **The light / light switch example is a really nice simple illustrative example, but you could probably strengthen this motivation by finding more settings where this arises**
> >
> > Thank you for the suggestion. We have added an example from biology on protein-protein interactions to the introduction. Similarly to other biological processes you mentioned, protein-protein interactions happen at different timescales, and it is difficult to appropriately model some of these interactions, which can also be nearly-instantaenous, e.g. in the case of transient protein-protein interactions [Acuner Ozbabacan et al, 2011]. Understanding and modelling these (nearly-)instantaneous effects is crucial to the understanding of many important biological processes, e.g. signaling cascades.
> >
> > **[Reproducibility] I know it’s just maximum likelihood wrt to the appropriate interventional distributions, but we need a little more detail in the main text to know how that translates to a loss function (e.g. are the distriubiutions over latents conditionally gaussian? Something else? ) and how the various components of the method are parameterized.**
> >
> > In the revised version, we have added a new Section 5.1 which discusses the original CITRIS model. This includes a discussion on the distributions over latents (MLP per latent that predicts the mean and std of the next time step), the general setup (e.g. convolutional encoder and decoder of VAE), and how the maximum likelihood translates to a loss function. We further clarify that iCITRIS builds upon all these elements of CITRIS, and the described modifications that follow are only for the prior. The full architectural details, including pseudocode algorithms for the full loss function of ENCO- and NOTEARS-based iCITRIS, are shown in Appendix E.
> >
> > **[...] illustrating the various piece that need to be implemented - a good figure would go a long way. They do supply code though.**
> >
> > Besides the new Section 5.1, we have added an explicit list of elements that need to be implemented for iCITRIS to the implementation details in Appendix E.1. Further, we have added an illustration of iCITRIS-ENCO (Figure 17) to give an overview of the key elements. We hope to fit it into the main paper for a camera ready version if space permits.

---

> > > ### Author Response · Authors · 2022-11-12
> > > **Author Response to Reviewer 1H3n [3/3]**
> > >
> > > **I think a running example would really help.**
> > >
> > > Thank you for pointing this out. To make the notation more intuitive, we have added the example of the light switch throughout Section 3.1, in which we introduce most notation, as well as Section 4.2 (Definition 4.3) to ease the understanding of the identifiability class.
> > >
> > > **Minimal causal variables were not well explained. I know this is a concept from Lippe et al 2022, but I would have liked to see an explanation here too, in order to make this paper self-contained.**
> > >
> > > In Section 3.2, we give a formal definition of the minimal causal variables, in particular for the theoretical results in Section 4. To make the minimal causal variables more intuitive, we have added an example in the explanation in Section 3.2. Due to space limitations, a longer explanation and discussion on minimal causal variables can be found in Appendix C.5.1.
> > >
> > > **There’s a really nice discussion of the assumptions in the appendix. Space is always an issue, but as much as possible of this should be moved into the text, because it does a better job than the main text of articulating what we’re ruling out with the assumptions.**
> > >
> > > Thank you for the suggestion. Most assumptions come from the original CITRIS [Lippe et al., 2022b] setting, which we introduce in Section 3.1. The main new assumption on partially-perfect interventions is detailed in Section 4.1. To further promote the discussion on the assumption in the revised version, we explicitly point to the Appendix C.2 in Section 4.2. Further, for the camera-ready version, we will try to move more discussion from Appendix C.2 in the main text where space permits.
> > >
> > > **I thought the Preliminaries (Appendix C1) section in the appendix was actually a really nice summary of the method - I found only really understood what you were doing once I read that. I know space is constrained, but augmenting figure 1 to include as much as possible of that information would be helpful to help keep track of all the moving pieces in this.**
> > >
> > > Based on your suggestion, we have extended Figure 1 to include the observation noise $E^t$. Further, we also add the observation function $h$ on the causal relations between $C^t,E^t$ and $X^t$, to visualize the relation between causal variables, observation noise, and observation image. We additionally clarified the introduction of the learnable pieces in CITRIS (Section 3.2, $g_{\theta}, \psi$) with further discussions from Appendix C.1.

---

### Official Review · Reviewer_4Ntv · 2022-10-24

**Confidence:** 3
**Correctness:** 4
**Technical Novelty And Significance:** 3
**Empirical Novelty And Significance:** 3
**Recommendation:** 8

**Clarity, Quality, Novelty And Reproducibility:**

From the definition of the problem, and discussion to the proposed model, the paper is organized in a clear style.

The paper investigates a new problem setting, which is a novelty. Following the assumptions, the authors offer a solid solution for that problem.

The proposed model is based on CITRIS, whose code has been released on GitHub. So it is not difficult to reproduce the experimental results reported in the paper.

**Strength And Weaknesses:**

Strength of the paper:
1. The paper investigates a new problem setting in which the causal variables may contain instantaneous causal relations.
2. The paper offers a detailed explanation of the motivation and method for modeling the instantaneous effects in causal variables.
3. Sufficient experiments are conducted to evaluate the effectiveness of the proposed model in modeling the instantaneous effects of causal variables.
4. A detailed SUPPLEMENTARY MATERIAL is attached after the main paper, in which some proofs for assumptions and theories are also contained.


Weakness of the paper:

There is little description of related models, such as CITRIS, however, that is not a big issue.

**Summary Of The Paper:**

The authors deeply dived into CAUSAL REPRESENTATION LEARNING on temporal sequences of observations and argue that temporal sequences of observations may still contain instantaneous causal relations in practice.
In order to model the causal structure of the temporal sequences of observations with  INSTANTANEOUS AND TEMPORAL EFFECTS, the authors propose a new method iCITRIS based on CITRIS.

**Summary Of The Review:**

In the paper, the authors investigate a new problem setting, which is a novelty.
Meanwhile, a detailed discussion is stated and an effective model for causal variables identification is proposed.

---

> ### Author Response · Authors · 2022-11-12
> **Author Response to Reviewer 4Ntv**
>
> Thank you for your positive feedback, highlighting the effectiveness and reproducibility of our method. Please see our answers below.
>
> **There is little description of related models, such as CITRIS, however, that is not a big issue.**
>
> In the revised version, we have introduced a new Section 5.1 which gives a more detailed description of the CITRIS architecture. This section also helps in separating the architecture-based contributions of iCITRIS. Further, the hyperparameter details for all baselines in our experiments are listed in Appendix E.

---

### Official Review · Reviewer_R1KQ · 2022-10-25

**Confidence:** 4
**Correctness:** 3
**Technical Novelty And Significance:** 4
**Empirical Novelty And Significance:** 4
**Recommendation:** 6

**Clarity, Quality, Novelty And Reproducibility:**

Clarity:  This paper is almost clear and well-written.  It is better to add a separate section to introduce CITRIS to help understand the difference and contributions.
Novelty: The problem and method are novel.  Just concerned with the rationality of the assumptions.
Reproducibility: code is attached.


**Strength And Weaknesses:**

Strength:
1) The task to identify the latent variables are very important to understand the temporal dynamic process.
2) The problem of the instantaneous effect is challenging and ubiquitous, which has the potential for improving time-series representation.
3) Both theoretical and experimental evidence shows that the proposed method is effective to solve the instantaneous effect problem.

Weaknesses:
1) This method assumes that the intervention targets are observed.  However, these interventions act on the latent variables.  How can we intervene the latent variables? Is it possible for the real-world application to observe the intervention targets?
2) All experiments are conducted on the simulation environment, such as Temporal Causal3DIdent and Pinball. These datasets are far away from real-world videos where the latent variables are unseen and uncontrollable.  Could this method is used for real-world video benchmarks, such as traffic event (Sutd-trafficqa[1]) or human actions (Oops[2]) . These datasets are more important to evaluate causal representation learning.
3) The contribution of this paper should be further highlighted. It is suggested to separate the original CITRIS method and this extension to avoid overlap and confusing contribution, e.g. adding a baseline method section to introduce CITRIS.

[1]Xu L, Huang H, Liu J. Sutd-trafficqa: A question answering benchmark and an efficient network for video reasoning over traffic events[C]//Proceedings of the IEEE/CVF Conference on Computer Vision and Pattern Recognition. 2021: 9878-9888.

[2]Epstein D, Chen B, Vondrick C. Oops! predicting unintentional action in video[C]//Proceedings of the IEEE/CVF conference on computer vision and pattern recognition. 2020: 919-929.


**Summary Of The Paper:**

This paper proposes a causal representation learning method, iCITRIS, to solve the instantaneous effects of time series data via latent intervention. Specifically, iCITRIS learn a temporal process using VAE, where the latent variables can be identified by the construction. Then the structure learning methods such as NoTears and Enco are employed to learn the relations of latent variables.
Beyond that, to solve the instantaneous effects, it assumes that the intervention targets are observed. Overall, as an extension of CITRIS, this method shows the ability to allow instantaneous effect in both theoretical and experimental perspectives.


**Summary Of The Review:**

Overall, I almost like this paper and think it focuses on an important challenge. However, the rationality of the assumptions that the intervention of latent variables can be accessed seems an issue.
Balancing the positive and negative points, I think this paper should be further polished. More experiments using real-world data are needed.


Most concerns are addressed in the responses. I would like to raise my score accordingly.

---

> ### Author Response · Authors · 2022-11-12
> **Author Response to Reviewer R1KQ [1/2]**
>
> Thank you for your constructive and helpful review that supports us to further strengthen the paper. Please see our answers below.
>
> **This method assumes that the intervention targets are observed. However, these interventions act on the latent variables. How can we intervene the latent variables? Is it possible for the real-world application to observe the intervention targets?**
>
> We agree that the need for knowing the intervention targets is a limitation of our work and may not be always possible to obtain, even though we point out that the specific values of the intervened variables does not need to be known. Nonetheless, we would like to clarify that in this work, we consider interactive environments where an agent or an expert can interact with the environment, as common in reinforcement learning settings. Further, the 'latent' causal variables are not explicitly hidden, i.e. unobserved, but rather we observe them in an image and can interact with them in the dynamical system. What is 'latent' about these variables is their definition and how the image translates to these causal variables. In such RL-like setups, an agent usually has actions which have direct influences on the causal variables, giving the agent the ability to interact with the environment. Moreover, we commonly are able to observe the agent's actions, which, with little pre-knowledge, can be used as intervention targets. For instance, in Pinball, a typical agent has actions such as press paddle on the left, the paddle on the right, reset the game, and eventually a table shake for freeing the ball (note that to the agent's knowledge, it is only 'action 1', 'action 2', 'action 3'). These actions have direct influences on the causal variables, giving the agent the ability to perform interventions. Therefore, although the paddles, the ball and the score are all 'latent' variables to the agent, since it only observes pixel-level images, it yet has the ability to intervene on them.
>
> A second example for observing intervention targets is when we are given a dataset by an expert, which has the ability to intervene on the system. For example, we may record a video of a human playing Pinball in real-world. In the end, we want an agent to learn the causal factors from the videos without the need to annotate the values of every single causal variable. In this case, giving the intervention targets as in iCITRIS is significantly easier to achieve than annotating the full causal system in every frame.
>
> Finally, as pointed out by all reviewers, this work is one of the first solutions to instantaneous effects in causal representation learning. Other works in this domain require, for example, counterfactual information [Ahuja et al., 2022; Brehmer et al., 2022; Locatello et al., 2020a] or supervised labels of the causal factors [Yang et al., 2021]; both of which are arguably at least as or even more difficult to obtain. Similarly, reviewer 1H3n argues that observing intervention targets is a limitation, but not "ground for rejection". Therefore, despite its assumptions, we believe that our work can form a stepping stone towards practical algorithms in this setting, and its insights on instantaneous effects can be used to extend future works in causal representation learning that go beyond these assumptions.
>
> **All experiments are conducted on the simulation environment[...] Could this method used for real-world video benchmarks, such as traffic event (Sutd-trafficqa[1]) or human actions (Oops[2])?**
>
> Going towards real-world videos is indeed an important future step for the whole field of causal representation learning. However, as mentioned in the previous response, we focus in this paper on finding the complete causal structure of environments where interactions/interventions by an agent or an expert are possible, such as in RL. While our simulation environments not being actual real-world, they yet test a broad range of real-world elements. For example, our Causal Pinball environment models the real-world dynamics of a classical RL environment, while the Causal3DIdent dataset test visually complexity.
>
> The two suggested video benchmarks are an interesting challenge, but go beyond the capabilities of iCITRIS and other provable causal representation learning methods we are aware of. Since every short video depicts a different system with new causal variables, a few-shot system is needed for predicting new causal structures from very few frames. Further, the benchmarks consider very high-level actions and intentions of humans/agents, while we focus with iCITRIS often on lower-dimensional causal variables (e.g. position, shape, etc.). Nonetheless, we thank the reviewer for these suggestions, as these benchmarks would indeed be worthy targets of some future method that might build on top of ours to solve the instantaneous relations in these datasets.

---

> > ### Author Response · Authors · 2022-11-12
> > **Author Response to Reviewer R1KQ [2/2]**
> >
> > **It is suggested to separate the original CITRIS method and this extension to avoid overlap and confusing contribution, e.g. adding a baseline method section to introduce CITRIS.**
> >
> > Based on your suggestion, we have introduced a new Section 5.1 which gives a more detailed description of CITRIS as a model/architecture. We further give more details on the elements which are shared with iCITRIS, e.g., implementing the latent distributions $p_{\phi}$ as conditional Gaussians.

---

> > ### Comment · Reviewer_R1KQ · 2022-11-19
> > **Response to authors**
> >
> > Dear Authors,
> >
> > I appreciate the author's detailed point-to-point responses. I have read these responses. It seems that my main concerns are not addressed well.
> >
> > 1) If knowing the intervention targets is difficult in the real-world data (we all agree that) but possible in the RL system,  may the paper highlight this in the title and abstract?  It is better to show the boundary of the method so that readers can know when they can use this method and avoid misleading. I know that it is not allowed to modify the paper. We can discuss the details here.
> >
> > 2) As noticed that "we may record a video of a human playing Pinball in real-world. In the end, we want an agent to learn the causal factors from the videos without the need to annotate the values of every single causal variable. In this case, giving the intervention targets as in iCITRIS is significantly easier to achieve than annotating the full causal system in every frame", it is suggested to add an experiment with this data (if it is a natural situation we would encounter in real life). When conclusions are claimed, experimental support is needed.  It is an alternative if the real applications are too difficult.
> >
> > I will calibrate my evaluation after further feedback.
> >
> > Many thanks,
> >
> > Reviewer R1KQ

---

> > > ### Author Response · Authors · 2022-11-21
> > > **Follow-up Response to Reviewer R1KQ [1/2]**
> > >
> > > Dear Reviewer R1KQ,
> > >
> > > Thank you for your answer and additional suggestions. Please see our response below.
> > >
> > > **1. If knowing the intervention targets is difficult in the real-world data (we all agree that) but possible in the RL system, may the paper highlight this in the title and abstract?**
> > >
> > > The main difference between natural real-world video data and the environments we consider in this paper, as mentioned in the previous response, is that they have an *interactive* component to them, e.g., influencing the environment by actions of an agent in an RL system. While our method does not have to perform these interactions by itself, it requires an environment in which some system, e.g. an expert, can interact with it. Based on your suggestion and to highlight these aspects further in the paper, we plan to do the following edits to the title and abstract (edits highlighted in bold):
> > >
> > > *New title*: Causal Representation Learning for Instantaneous and Temporal Effects **in Interactive Systems**
> > >
> > > *New abstract*: Causal representation learning is the task of identifying the underlying causal variables and their relations from high-dimensional observations, such as images. Recent work has shown that one can reconstruct the causal variables from temporal sequences of observations under the assumption that there are no instantaneous causal relations between them. In practical applications, however, our measurement or frame rate might be slower than many of the causal effects. This effectively creates ``instantaneous'' effects and invalidates previous identifiability results. To address this issue, we propose iCITRIS, a causal representation learning method that allows for instantaneous effects in **intervened** temporal sequences **when intervention targets can be observed, e.g., as actions of an agent**. iCITRIS identifies the potentially multidimensional causal variables from temporal observations, while simultaneously using a differentiable causal discovery method to learn their causal graph. In experiments on three datasets **of interactive systems**, iCITRIS accurately identifies the causal variables and their causal graph.
> > >
> > > **2. "we may record a video of a human playing Pinball in real-world." [...] it is suggested to add an experiment with this data (if it is a natural situation we would encounter in real life)**
> > >
> > > Thank you for the suggestion. Given the time constraints for the rebuttal and your suggestion, we perform the following experiment. The original Causal Pinball environment deploys a random policy that controls the paddles, i.e. activates the two paddles independent of the environment state. However, a human in real-world may select the interventions based on the environment state to achieve a higher score. Thus, to bring the environment closer to real-world, we extend Causal Pinball by replacing the random policy by a human-like policy. This policy activates the paddles with higher probability when they can hit the ball, and only rarely intervenes on the paddles otherwise. We report the results on this new Pinball setup for our best method, iCITRIS-ENCO, and the best baseline, CITRIS, over 3 seeds below. iCITRIS-ENCO identifies the causal variables and the graph up to similar accuracy for a human-like policy, while it makes the task even harder for baseline CITRIS. For the camera-ready version, we will add these results in Section 6.4 along with the results of the remaining baseline methods.
> > >
> > > | Model + Dataset                  |  $R^2$ (diag $\uparrow$ / sep $\downarrow$) | SHD (instant $\downarrow$ / temp $\downarrow$) |
> > > |----------------------------------|:-------------------------------------------:|:-------------------------------------------------------:|
> > > | iCITRIS-ENCO (random policy)     | 0.99 ($\pm 0.00$) / 0.12 ($\pm 0.01$)       | 0.67 ($\pm 1.15$) / 3.00 ($\pm 0.71$)                   |
> > > | iCITRIS-ENCO (human-like policy) | 0.98 ($\pm 0.00$) / 0.11 ($\pm 0.03$)       | 1.00 ($\pm 1.00$) / 3.67 ($\pm 0.58$)                   |
> > > | CITRIS (random policy)           | 0.90 ($\pm 0.06$) / 0.39 ($\pm 0.07$)       | 3.00 ($\pm 1.00$) / 7.67 ($\pm 2.31$)                   |
> > > | CITRIS (human-like policy)       | 0.87 ($\pm 0.06$) / 0.56 ($\pm 0.09$)       | 4.33 ($\pm 1.15$) / 8.00 ($\pm 1.00$)                   |

---

> > > > ### Author Response · Authors · 2022-11-21
> > > > **Follow-up Response to Reviewer R1KQ [2/2]**
> > > >
> > > > Other differences of a real-world setting compared to what we have implemented currently include (a) visual complexity and (b) higher noise level. In terms of visual complexity, we refer to our experiments on the Temporal Causal3DIdent dataset modeling various object shapes in 3d. For the second point, the two critical sources of noise in real-world are the temporal process, e.g., the ball is not moving as expected due to external influences like sand, and the accuracy of the intervention targets, i.e., how often do interventions not succeed. We show in the experiments on the Voronoi benchmark in Section 6.2 that iCITRIS works well for environments with a high degree of noise in the temporal process. On the other hand, noise on intervention targets occur, for example, when a human presses a paddle button in Pinball but due to a mechanical failure, the paddle does not move forward as expected. In Appendix F.1.1, we report results on environments with such noisy intervention targets, showing that iCITRIS is robust against this type of noise as well.
> > > >
> > > > ---
> > > >
> > > > Thank you again for your time in reviewing and discussing the rebuttal. Please let us know if further concerns or suggestions remain.

---

> > > > > ### Comment · Reviewer_R1KQ · 2022-11-21
> > > > > **Response to Authors**
> > > > >
> > > > > Thanks very much for the detailed responses.
> > > > >
> > > > > Most of my concerns were addressed.  Though it is difficult and time-limited, experiments with real-world data are expected in the final version.
> > > > > Thanks again for the author's effort to solve the concerns. I would like to raise my score accordingly.
> > > > >
> > > > > With best regards,
> > > > >
> > > > > Reviewer R1KQ

---

### Official Review · Reviewer_4Gh8 · 2022-10-27

**Confidence:** 3
**Correctness:** 3
**Technical Novelty And Significance:** 3
**Empirical Novelty And Significance:** 2
**Recommendation:** 5

**Clarity, Quality, Novelty And Reproducibility:**

(-) The paper is a bit difficult to read and it is not clear in the beginning what and why are causal variables and causal representation.

(-) The problem is important; whereas, the proposed solution relies on access to the intervention of each causal variable. It becomes less significant and interesting with the proposal. (I do reckon the effort on the proof.) Maybe try to justify how reasonable the setting is.

(-) It is not clear to me how the causal variables are determined. How the number of latent variables is determined and how the invertible function is constructed when the dimensions of the observation and the latent spaces are different?


**Strength And Weaknesses:**

The strength:

(+) The work spends a lot of effort on the proof of identifiability.

(+) The problem is important: the generalization of CITRIS (Lippe et al., 2022b) in the presence of instantaneous effects.


(-) The identifiability relies on the access to the interventions on each causal variable, which can not be easy to get in practice. This in turn limits the contribution of the work by the setting of the solution.

(-) The paper claims to identify the causal representation; however, the experiments didn't show what is the learned causal representation. Perhaps, besides showing the performance of causal discovery in the setting, it can be helpful to explain the causal representations. If there is any in the appendix, it would be good to include part of the analysis in the main content. Moreover, it would be more convincing to include a

**Summary Of The Paper:**

The paper aims at learning the causal representation from time-series data with the access of interventions. The formulation is a state-space model, which the causal process is unobserved and the observation is a funtion of the unobserved causal process. It generalizes CITRIS (Lippe et al., 2022b) to include instantaneous effects in the underlying causal graph. To identify the unique representation, the paper requires the access to the intervensions on each latent variable. The causal graph and the causal representation (or variables) can be identified with the access of the interventions by maximizing likelihood, maximizing the information content, and minimizing the edges of a causal graph.


**Summary Of The Review:**

My main concern is the assumption of access to the interventions and the significance of the contributions can be limited. Moreover, it would be more convincing with real-world data experiments.

---

> ### Author Response · Authors · 2022-11-12
> **Author Response to Reviewer 4Gh8 [1/2]**
>
> Thank you for your helpful and valuable review, which has allowed us to further improve the paper. Please find our answers below.
>
> **The identifiability relies on the access to the interventions on each causal variable, which can not be easy to get in practice. This in turn limits the contribution of the work by the setting of the solution.**
>
> While we agree that sometimes we cannot get access to all the interventions that we might need to identify all the causal variables, we want to stress a few points: (1) the inherent difficulty of our setting, (2) the fact that we do not require single-target interventions on each causal variable, but only $\lfloor \log_2(K) \rfloor + 2$ interventions to identify all $K$ causal variables, (3) the definition of \emph{minimal causal variables}, which allows us to formally describe what can be identified if we do not have all these interventions.
>
> Causal representation learning is a naturally much more difficult task than causal discovery, since we also need to recover the causal variables from high-dimensional data, e.g. images.
> Because of that, having access to interventions on all variables is a common assumption in related works on causal representation learning, including [Ahuja et al., 2022; Brehmer et al., 2022; Lippe et al., 2022b; Locatello et al., 2020]. One example for the need for interventions is that if we have two variables following a Gaussian distribution with the same std, then those variables cannot be identified without interventions or similar distribution changes in general. The problem is that any rotation of the two variable axes around the Gaussians mean can represent the data generation process equally well. However, under interventions, these two variables become identifiable.
>
> In our setting, instantaneous effects introduce additional challenges (and therefore requirements in terms of background knowledge or interventions) in this setting. For instance, in Section 4.1, we prove that without interventions on all variables, we cannot identify all causal variables in our setting iTRIS (Instantaneous TempoRal Intervened Sequences). This underlines the explicit need for interventions to resolve instantaneous effects, without making any additional (parametric) assumption.
>
> Furthermore, we want to point out that we do not need an intervention on each causal variable, or in other words $K$ single-target interventions. The interventions we consider do not need to be single-target. Instead, multiple variables can be intervened upon at the same time. As shown by [Lippe et al., 2022c] and similar to what shown by [Eberhardt, 2007] for causal discovery, CITRIS requires at most $\lfloor \log_2(K) \rfloor + 2$ experiments, i.e. sets of simultaneous interventions, which generalizes to iCITRIS due to sharing the same interventional assumptions.
>
> Moreover, with the concept of the minimal causal variables, we do formally describe what can be identified for variables that have not been intervened. The only restriction for those variables is that they cannot be instantaneous children of any other variable (Lemma 4.2). In iCITRIS, such variables are modelled in $z\_{\psi\_0}$. If multiple variables without interventions exist, they may be entangled in $z\_{\psi\_0}$, similar as in CITRIS [Lippe et al., 2022b].
>
> [Lippe et al., 2022c] Phillip Lippe, Sara Magliacane, Sindy Löwe, Yuki M. Asano, Taco Cohen, and Efstratios Gavves. Intervention Design for Causal Representation Learning. First Workshop on Causal Representation Learning (CRL 2022), UAI 2022.
>
> **The paper claims to identify the causal representation; however, the experiments didn't show what is the learned causal representation.**
>
> In our experimental evaluation, we show both the causal representation (in terms of the causal variables that we learn), as well as the causal graph. The experiments in Section 6 evaluate the learned causal representations of a model by the $R^2$ correlation scores, similar to previous works [von Kügelegen et al., 2021; Lachapelle et al., 2022a; Lippe et al., 2022b]. The correlations are reported between the true causal variables and the learned latent variables, and measure how well the model has identified and separated the individual causal variables. In other words, it shows how well a model has learned a *causal representation* according to the identifiability class of Definition 4.3. The causal graph predicted by the model is then evaluated by the SHD. We further show example causal graphs predicted by all models and additional causal representation metrics (e.g. Spearman correlation and triplet evaluation following [Lippe et al., 2022b]) in Appendix F.
>
> **Moreover, it would be more convincing to include a**
>
> It seems that the review accidentally cut off at the end of the "Strengths and Weaknesses" section. We would appreciate a clarification by the reviewer if further questions remain.

---

> > ### Author Response · Authors · 2022-11-12
> > **Author Response to Reviewer 4Gh8 [2/2]**
> >
> > ### Presentation
> >
> > **The paper is a bit difficult to read and it is not clear in the beginning what and why are causal variables and causal representation.**
> >
> > Thanks for the suggestion, we have improved our manuscript. To make the overall setting more accessible, we have added in the revised version a running example of the light switch in the definitions of Section 3.1 and 4.2. To paraphrase Section 3.1, in this example, the state of the light switch and the light bulb are the two *causal variables*, i.e. random variables between which causal relations may exist. These causal variables can be potentially multidimensional variables, e.g., multiple light bulbs controlled by the same switch or the 3d position of the light bulb. However, we only observe a high-dimensional observation of this system, e.g. an image of the light switch and the light bulb.
> >
> > A causal representation, on the other hand, is a space/set of values that separately models the causal variables of the system. In other words, in a VAE, we call the latent space a *causal representation* if different latent variables model different causal factors, and the learned causal graph corresponds to the true one. In terms of a light switch example, one latent variable may model the state of the switch, while another models the state of the bulb. Note that we allow for invertible transformations between latent and causal variables. For example, the latent variable of the switch might model it in the opposite direction, i.e. 1 for off and 0 for on.
> >
> > We state a full formal definition of the identifiability class in Definition 4.3, and it aligns with the related work on causal representation learning.
> >
> > **It is not clear to me how the causal variables are determined. How the number of latent variables is determined and how the invertible function is constructed when the dimensions of the observation and the latent spaces are different?**
> >
> > In the revised version, we have introduced a new Section 5.1 which gives a higher level overview of the original CITRIS architecture on which iCITRIS builds upon. We additionally provide answers for your questions below:
> >
> > *Determining causal variables*: In practice, iCITRIS learns a latent space like a variational autoencoder (VAE). The major difference is, however, that it enforces the prior structure described in Section 4.2. Each latent variable learns which causal variable it belongs to (assignment function $\psi$). After training, we therefore determine a causal variable by the group of latents that are assigned to it.
> >
> > *Number of latent variables*: We assume that we can overestimate the true number of causal variables. The latent space must be at least the dimensionality of the causal variables to model the whole system dynamics, but can be arbitrarily many times larger. In experiments, we observe that the model is not sensitive to the extent of additional latent dimensions, and we choose them based on having a good reconstruction by the (V)AE, i.e. as we would do for any other VAE. For the Voronoi benchmark, this results in 8, 12, 18 latents for 4, 6, 9 causal variables respectively. We use 32 latents for Causal3DIdent, and 24 for Causal Pinball. All hyperparameters, including the number of latent variables, can be found in Appendix E.2 (Table 4).
> >
> > *Invertible function for dimensionality mismatch*: In all experiments of Section 6, the observation space is much higher dimensional than the latent space. In iCITRIS, the invertible map $g_{\theta}$ is approximated by learning an encoder and decoder of a (variational) autoencoder (see Section 5.1 and Appendix E.2 for the architecture hyperparameters). Training such a model to have a low reconstruction loss results in an approximate invertible function between the image and latent space. Similar approaches were taken by previous works [Khemakhem et al., 2020a; Locatello et al., 2020; Brehmer et al., 2022].
> >
> > ### Summary of The Review
> >
> > **My main concern is the assumption of access to the interventions and the significance of the contributions can be limited. Moreover, it would be more convincing with real-world data experiments.**
> >
> > As mentioned before, causal representation learning is a nascent field that deals with a very challenging setting and the assumption on the access to interventions is based on the inherent difficulty of the task. We do believe that this line of work could be applied in the future to settings in which interventions are common, e.g. in reinforcement learning. In such RL-like setups, an agent usually has actions which have direct influences on the causal variables, giving the agent the ability to interact with the environment. We showcase these types of environment in our Causal Pinball setup.
> > At the current stage of our work and other works in causal representation learning, there are no real-world experiments yet that could be easily applied.

---

### Author Response · Authors · 2022-11-12
**General Response**

We would like to thank all reviewers for their insightful and valuable feedback.

We are happy to see that all reviewers highlight the importance and novelty of the problem (instantaneous effects in time series) as well as the effectiveness of our presented approach. For example, reviewer 1H3n mentions that "this is high quality original work" and "deals with an important problem and has strong experimental results". Reviewer 4Ntv describes the approach as "an effective model for causal variables identification", and reviewer R1KQ states "the problem and method are novel". Finally, reviewer 4Gh8 highlights that "the problem is important" and "the work spends a lot of effort on the proof of identifiability".

We welcome the opportunity to clarify all points raised. Based on the reviews, we have made the following main edits to the paper (edits marked in red in PDF):

* We have introduced a new Section 5.1 which separates the CITRIS [Lippe et al., 2022b] architecture from the proposed extension to instantaneous effects (iCITRIS). This gives more details in the specific implementation of the model, and further highlights the architectural contributions of this work.
* We have made the notation and definitions in Section 3 and 4.2 more accessible by, for example, adding the running example of the light-switch environment.
* We have extended Figure 1 to include more notation elements of the main text, e.g., the observation noise $E^t$ and the observation function $h$.
* To further promote reproducibility, we have added an illustration of the method in Figure 17 (Appendix E.1), and stated an explicit list of elements that need to be implemented in iCITRIS.

---

### Author Response · Authors · 2022-11-18
**Thank you for your reviews**

Dear Reviewers, dear AC,

We are entering the last day of the discussion period. We would like to thank you again for all your time and suggestions, which have definitely helped us improve the clarity and presentation of our paper.

We are confident that we have addressed all points raised in both responses and the revised version. If there happen to be any remaining concerns, we would be happy to address these and incorporate further changes in the camera-ready.

Thanks!
The authors

---

### Decision · Program_Chairs · 2023-01-20

**Decision:**

Accept: poster

**Justification For Why Not Higher Score:**

The paper is interesting and worth publishing, but it's not very novel, insightful, or surprising.

**Justification For Why Not Lower Score:**

N/A

**Metareview: Summary, Strengths And Weaknesses:**

This paper studies the problem of causal representation learning from high-dimensional observations in the presence of instantaneous effects. Overall, reviewers considered the work interesting and novel. Therefore, I recommend the acceptance of the paper.

**Note From Pc:**

if the above contains the word "oral" or "spotlight" please see: "oral" presentation means -> notable-top-5% and "spotlight" means -> notable-top-25%. As stated in our emails, we are disassociating presentation type from AC recommendations